# Spatiotemporal ITCZ dynamics during the last three millennia in Northeastern Brazil and related impacts in modern human history

Giselle Utida[1]*, Francisco W. Cruz[1], Mathias Vuille[2], Angela Ampuero[1], Valdir F. Novello[3], Jelena Maksic[4], Gilvan Sampaio[5], Hai Cheng[6,7,8], Haiwei Zhang[6]; Fabio Ramos Dias de Andrade[1], R. Lawrence Edwards[9]

[1]Instittuto de Geociências, Universidade de São Paulo, Rua do Lago, 562, Cidade Universitária, São Paulo-SP, 05508-090, Brazil

[2]Department of Atmospheric and Environmental Sciences, University at Albany, SUNY, Albany, NY, USA

[3]Geo- and Environmental Research Center, University of Tübingen, Tübingen, Germany

[4]Division of Impacts, Adaptation and Vulnerabilities (DIIAV), National Institute for Space Research (INPE), São Jose dos Campos-SP, Brazil

[5]General Coordination of Earth Science (CGCT), National Institute for Space Research (INPE), São Jose dos Campos-SP, Brazil

[6]Institute of Global Environmental Change, Xi'an Jiaotong University, Xi'an, China

[7]State Key Laboratory of Loess and Quaternary Geology, Institute of Earth Environment, Chinese Academy of Sciences, Xi'an, China

[8]Key Laboratory of Karst Dynamics, MLR, Institute of Karst Geology, CAGS, China

[9]Department of Earth Sciences, University of Minnesota, Minneapolis, MN, USA

*Corresponding author: giselleutida@hotmail.com

*Abstract*
Changes in tropical precipitation over the past millennia have usually been associated with
latitudinal displacements of the Intertropical Convergence Zone (ITCZ). Recent studies
provide new evidence that contraction and expansion of the tropical rainbelt may also have
contributed to ITCZ variability on centennial time scales. Over tropical South America few
records point to a similar interpretation, which prevents a clear diagnosis of ITCZ changes
in the region. In order to improve our understanding of the equatorial rainbelt variability,
our study presents a reconstruction of precipitation for the last 3200 years from the
Northeast Brazil (NEB) region, an area solely influenced by ITCZ precipitation. We analyze
oxygen isotopes in speleothems that serve as a faithful proxy for the past location of the
southern margin of the ITCZ. Our results, in comparison with other ITCZ proxies, indicate
that the range of seasonal migration, contraction and expansion of the ITCZ was not
symmetrical around the equator on secular and multidecadal timescale. A new NEB ITCZ
pattern emerges based on the comparison between two distinct proxies that characterize
the ITCZ behavior during the last 2500 years, with an ITCZ zonal pattern between NEB
and the eastern Amazon. In NEB, the period related to the Medieval Climate Anomaly
(MCA – 950 to 1250 CE) was characterized by an abrupt transition from wet to dry
conditions. These drier conditions persisted until the onset of the period corresponding to
the Little Ice Age (LIA) in 1560 CE, representing the longest dry period over the last 3200
years in NEB. The ITCZ was apparently forced by teleconnections between Atlantic and
Pacific that controlled the position, intensity and extent of the Walker cell over South
America, changing the zonal ITCZ characteristics, while sea surface temperature changes
in both the Pacific and Atlantic, stretched/weakened the ITCZ-related rainfall meridionally
over NEB. Wetter conditions started around 1500 CE in NEB. During the last 500 years,
our speleothems document the occurrence of some of the strongest drought events over
the last centuries, which drastically affected population and environment of NEB during the

Portuguese colonial period. The historical droughts were able to affect the karst system, and led to significant impacts over the entire NEB region.

*Keywords*: Holocene, stalagmites, stable isotopes, droughts, Portuguese colony

*1. Introduction*

Northeastern Brazil (NEB) is one of the areas in South America (SA) most vulnerable to the impacts of climate change. The semi-arid conditions in NEB are strongly affected by precipitation variability, and since the 18th century the region has experienced more frequent drought events (Marengo and Bernasconi, 2015; Lima and Magalhães, 2018). Today the frequent droughts put ~57 million people, ~27% of the Brazilian population, at risk of experiencing water scarcity (Marengo and Bernasconi, 2015; Lima and Magalhães, 2018). Aside from native people, the region has been occupied since the Portuguese colonization in the 16th century, and the ensuing intense agricultural activity has been responsible for a large-scale degradation of the Caatinga biome, the typical vegetation of NEB's semi-arid areas. This land mismanagement and the increasing frequency of regional droughts has put some of these areas at great risk of desertification (Marengo and Bernasconi, 2015; Sampaio et al., 2020). Advancing our knowledge about NEB's climate and recurrence of extreme events in a long-term context is therefore of great importance to better anticipate the impacts of these intense and abrupt drought events.

The Intertropical Convergence Zone (ITCZ) is one of the key elements responsible for precipitation over NEB, which also indirectly affects the South American Summer Monsoon (SASM). When the ITCZ is in its southernmost position during austral autumn, northern areas of NEB experience increased precipitation (Schneider et al., 2014), while the precipitation in the southern areas of NEB occurs mainly during austral summer in response to climatic conditions in the tropical South Atlantic (Vera et al., 2006; Vuille et al.,

2012). Although these systems are independent and arise in different seasons, the
position of the ITCZ affects SASM intensity and its development through moisture influx to
the continent (Vuille et al., 2012; Schneider et al., 2014).
On orbital to centennial timescales, weakened precipitation in NEB has been
associated with enhanced subsidence over NEB during intense SASM periods (Cruz et al.,
2009; Orrison et al., 2022), giving rise to a zonal dipole between the western Amazon and
NEB (Cruz et al., 2009; Novello et al., 2018). This mode also operates today on
interannual and seasonal time scales (Lenters and Cook, 1997; Sulca et al., 2016).
More recent studies suggested that these variations on millennial and centennial
timescales in NEB may also have been caused by contraction or expansion of the tropical
rainbelt affecting the precipitation over South America (Utida et al., 2019; Chiessi et al.,
2021). These ITCZ dynamics would be forced by changes in tropical Atlantic and Pacific
sea surface temperature (SST) and related atmospheric circulation changes (e.g.,
Lechleitner et al., 2017; Utida et al., 2019; Chiessi et al., 2021; Steinman et al., 2022).
These results suggest complex ITCZ dynamics operating over NEB; a region where the
lack of studies complicates the paleoclimate interpretations for the last millennia.
In comparison with the ITCZ, the SASM has received more attention from recent
studies, mainly due to its larger area of influence in SA, extending from the tropical Andes
to the Amazon and southeastern SA (e.g., Apaéstegui et al., 2018; Azevedo et al., 2019;
Della Libera et al., 2022). Rainfall variability over Southern Northeast Brazil (S-NEB) is
also determined by the dynamics of the South Atlantic Convergence Zone (SACZ), a
component of the SASM (Novello et al., 2018; Zilli et al., 2019; Wong et al., 2021). The
spatiotemporal precipitation variability over tropical SA during the Common Era (CE) was
evaluated based on a network of high-resolution proxy records (Novello et al., 2018;
Campos et al., 2019; Orrison et al., 2022). These studies point to an association between
SASM variability and the latitudinal displacement of the ITCZ and SACZ, although
changes in the latitude of the ITCZ during the last millennia are not well established.
Previous studies based on oxygen and hydrogen isotopes from paleorecords
obtained in NEB have served as useful proxies for ITCZ precipitation in the region (Cruz et
al., 2009; Novello et al., 2012; Utida et al., 2019), while carbon isotopes have been used to
interpret soil erosion/production and vegetation cover in different biomes of Brazil (Utida et
al., 2020; Azevedo et al., 2021; Novello et al., 2021).
For the past 4200 years, NEB has experienced semi-arid conditions (Cruz et al.,
2009; Utida et al., 2020) that were imprinted on the oxygen isotope signals recorded in
stalagmites. These drier conditions in NEB could have resulted in a seasonal bias toward
the $\delta^{18}O$ rainfall of recharge periods or an evaporative fractionation of stored karst water
(Baker et al., 2019). In addition, isotopic fractionation processes associated with different
karst architectures can affect the stalagmites $\delta^{18}O$ signals (Treble et al., 2022).
Unfortunately, cave monitoring in northern NEB is not available due to the scarcity of
dripping water, probably as a result of increasing droughts in the region in the last decades
(Marengo and Bernasconi, 2015). Because of this, the interpretation of oxygen isotopes in
the region has been challenging.
Although the hydrological processes occurring in the epikarst may affect the
fractionation of oxygen isotope values in the dripping water and thus control $\delta^{18}O$ recorded
in stalagmites on a global scale, previous studies mentioned above, suggest a strong
relationship with rainfall amount based on model results and comparison with other
regional and global records.
Building on these recent advances, we present an ITCZ precipitation reconstruction
based on stalagmite records from the state of Rio Grande do Norte (RN), located at the
modern southernmost limit of the ITCZ in eastern South America (Fig. 1). By using oxygen
isotopes obtained from these stalagmites we reconstruct precipitation, based on field

correlations between precipitation amount and oxygen isotopic composition of modern

rainfall, and by using carbon isotopes to reconstruct vegetation/soil cover over the last

3200 years over NEB. These data are essential to fill the gap of high-resolution records in

NEB and to improve the interpretation of ITCZ dynamics over SA and how they are related

to SASM variability during the CE.

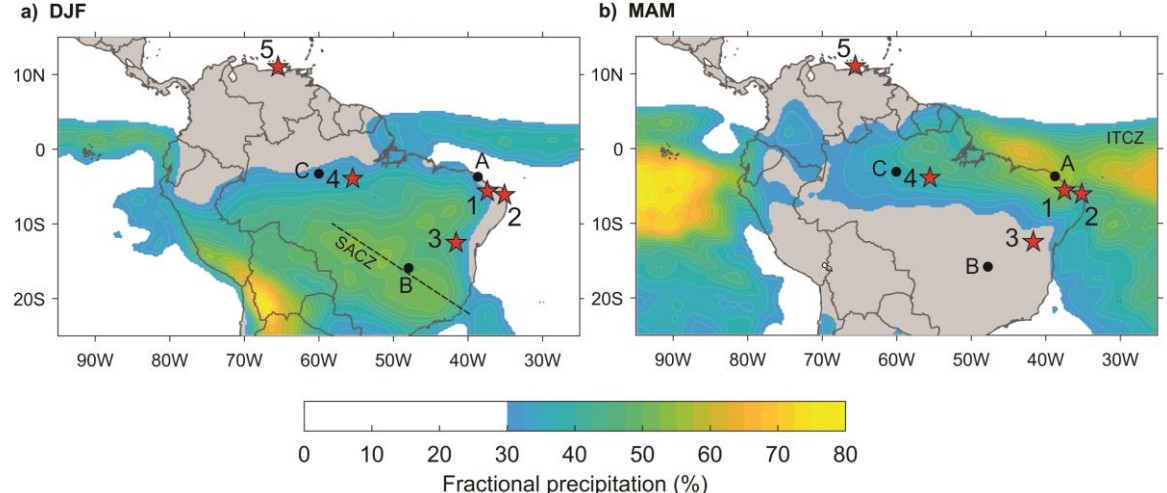

Figure 1 – Location and precipitation climatology of study sites during the austral

summer (DJF - December to February) and autumn (MAM - March to May). Color shading

indicates percentage of the annual precipitation total that is received during either DJF or

MAM and highlights the extent of (a) SASM over the continent and (b) the ITCZ over the

ocean. Precipitation data from the Global Precipitation Measurement (GPM) mission, with

averages calculated over the period 2001–2020. 1) Trapiá and Furna Nova Cave, Pedra

das Abelhas Station (this study), 2) Boqueirão Lake (Utida et al., 2019), 3) Diva de Maura

Cave (Novello et al., 2012), 4) Paraíso Cave (Wang et al., 2017), 5) Cariaco Basin (Haug

et al., 2001). GNIP stations: A) Fortaleza, B) Brasília, C) Manaus.

*2. Regional settings*

*2.1. Study area*

We study stalagmites from two caves located in the Rio Grande do Norte State, in northern NEB (Fig. 1), Trapiá and Furna Nova Cave. The caves were developed in the Cretaceous carbonate rocks of the Jandaíra Formation, Potiguar Basin, close to the Apodi River valley in a region of exposed karst pavements (Pessoa-Neto, 2003; Melo et al., 2016; Silva et al., 2017). We collected speleothems in Trapiá and Furna Nova caves. Trapiá Cave (5°33'45.43"S, 37°37'15.92"W) is a 2330 m long cave with 29 m of bedrock above the cave cavity. This cave is located 90 km from the Atlantic coast and ~ 50 m above sea level, with temperature and relative humidity of 28.5°C and 100%, respectively, in the chamber. Furna Nova Cave (5°2'3.22"S, 37°34'16"W) is located 60 km north of Trapiá Cave, 45 km from the Atlantic coast and ~95 m above sea level. The cave is 239.3 m long, with 29.8 m of bedrock above the cave cavity. Its temperature and relative humidity in the speleothem chamber are 25°C and 95.0%, respectively.

The annual mean temperature in the region is around 28°C (INMET - National Institute of Meteorology – Instituto Nacional de Meteorologia – data from 1961-1990) and the average precipitation is approximately 730 mm/year, concentrated in the period between March and May, during the southernmost position of the ITCZ (Agência Nacional de Águas – ANA - National Agency of Waters, 2013; Ziese et al., 2018). Caatinga dry forest is the typical vegetation of the region. It is adapted to short rainy seasons of 3 to 4 months in length and tolerates large interannual variations in precipitation. It is characterized by sparse dry forest, dominated by arboreal deciduous shrubland (Erasmi et al., 2009).

*2.2. Climatology*

The drylands of NEB extend from 2.5°S to 16.1°S, and from 34.8°W to 46°W, with an area of about 1,542,000 km$^2$, representing 18.26% of the Brazilian territory (Marengo

and Bernasconi, 2015). Although the whole area is classified as semi-arid and has faced
intense droughts, especially influenced by El Niño, there are significant differences in
climatic systems between the northern and southern sectors of NEB. Furthermore, the
NEB eastern coastal sector is characterized by a different rainfall seasonality, receiving
more rainfall across the year, as the climate in this region is modulated by the sea breeze
circulation and easterly wave disturbances during June and July (Gomes et al., 2015;
Marengo and Bernasconi, 2015; Utida et al., 2019). Northern NEB (N-NEB), where the
studied caves are located, receives most of its precipitation from March to May, when the
seasonal migration of the ITCZ reaches its southernmost position around 2°N (Schneider
et al., 2014; Utida et al., 2020), and ITCZ-related precipitation extends across the equator
southward to NEB (Fig. 1). In Southern-NEB (S-NEB), the precipitation occurs mainly
during summer, from December to February influenced by the margins of the SACZ (Fig.
1a).

*3. Materials and Methods*
The rainfall patterns over the study area were evaluated by analyzing monthly
rainfall data from the Pedra das Abelhas National Agency of Water (ANA) Station – RN,
located ~ 1 km from the Trapiá Cave (Fig. 1), using data from 1911 to 2015 (n=103). In
order to exclude possible extreme events with a known forcing, we excluded the 39 El
Niño - Southern Oscillation (ENSO) years that most drastically changed the precipitation
amount in NEB, following the methodology of Araújo et al. (2013).
In order to identify spatial patterns of rainfall associated with the oxygen isotope
signal in northeast and central Brazil, we produced maps showing the Pearson's
correlation scores between GPCC gridded precipitation anomalies (Schneider et al.,
2011), based on the period 1961-1990 for December to February (DJF) and March to May
(MAM) (Ziese et al., 2018); and $\delta^{18}O$ values for IAEA-GNIP stations (International Atomic

Energy Agency - Global Network of Isotopes in Precipitation, IAEA-WMO, 2021) for Northern NEB (Pedra das Abelhas ANA and Fortaleza GNIP Station); Southern NEB (Andaraí ANA and Brasília GNIP stations) and the Eastern Amazon (Belterra ANA and Manaus GNIP stations). The IAEA stations were chosen based on their closest proximity to sites discussed in the study: 1) Trapiá Cave and Furna Nova Cave (this study), 2) Boqueirão Lake (Utida et al., 2019), 3) Diva de Maura Cave (Novello et al., 2012) and 4) Paraíso Cave (Wang et al., 2017). Sites 1 and 2 are located in in N-NEB, 3 in S-NEB and 4 in the Eastern Amazon. Four stalagmites were collected in N-NEB caves, two at Trapiá Cave, TRA5 and TRA7 that are 178 and 270 mm long, respectively (Fig. S1), and two at Furna Nova, FN1 and FN2, with a length of 202 and 95 mm, respectively (Fig. S2). The stalagmite FN1 was previously studied by Cruz et al. (2009) for chronology and oxygen isotopes. Utida et al. (2020) also studied TRA7 for chronology and carbon isotopes.

Chronological studies on speleothems were based on U-Th geochronology performed at the Laboratories of the Department of Earth and Environmental Sciences, College of Science and Engineering, University of Minnesota (USA), and at the Isotope Laboratory of the Institute of Global Environmental Change, Xi'an Jiaotong University (China), according to Cheng et al. (2013). Subsamples of ~100 mg were obtained in clear layers, close to the growth axis trying to keep a maximum thickness of 1.5 mm, 10 mm wide and no more than 3 mm depth. The powder samples were dissolved in 14 N $HNO_3$ and spiked with a mixed solution of known $^{233}U$ (0.78646 ± 0.0002 pmol/g) and $^{229}Th$ (0.21686 ± 0.0001 pmol/g) concentration. Th and U were co-precipitated with FeCl and separated with Spectra/Gel® Ion Exchange 1x8 resin column with 6N HCl and super clear water, respectively. Th and U were counted in an inductively coupled plasma-mass spectrometry (MC-ICP-MS Thermo-Finnigan NEPTUNE PLUS) and the results calculated in a standard spreadsheet based on Edwards et al. (1987) and Richards and Dorale (2003) using the isotopic ratios measured, machine parameters and corrections factors to

eliminate effects of contamination by detrital Th to finally obtain the age of each sample.
The decay constants used are: $\lambda_{238}$ 1.55125 x $10^{-10}$ (Jaffey et al., 1971), $\lambda_{234}$ 2.82206 x $10^{-}$
$^6$ and $\lambda_{230}$ = 9.1705 x $10^{-6}$ (Cheng et al., 2013). Corrected $^{230}$Th ages assume the initial
$^{230}$Th/$^{232}$Th atomic ratio of 4.4 ± 2.2 x $10^{-6}$. Those are the values for a material at secular
equilibrium, with the bulk earth $^{232}$Th/$^{238}$U value of 3.8 (McDonough and Sun, 1995). The
ages are reported in BP (Before Present, defined as the year 1950 A.D.) and also
converted to Common Years (CE) and age uncertainties are 2 $\sigma$. We analyzed a large
number of U/Th ages to improve the age model and reduce the errors associated with
detrital Th and recrystallization.

Age models of speleothem TRA5 and FN2 were based on 12 and 10 U/Th dates,

respectively (Table S1 and S2). The FN1 chronology is based on 10 previously published
U/Th results obtained by Cruz et al. (2009) plus 8 additional new dates obtained for this
study (Table S1). Speleothem TRA7 has 27 U/Th ages that were presented in Utida et al.
(2020). The individual age models for all speleothems were constructed by the software
COPRA (Breitenbach et al., 2012) through a set of 2000 Monte Carlo simulations, where a
random age within the ±1 $\sigma$ age interval was chosen each time.

For oxygen and carbon isotope analysis of the speleothems, around 200 $\mu$g of

powder was drilled for each sample, consecutively at intervals of 0.1 mm (TRA5), 0.3 mm
(TRA7) and 0.15 mm (FN2), with a Micromill micro-sampling device. These samples were
prepared using an online automated carbonate preparation system and analyzed by a
GasBench interfaced to a Thermo Finnigan Delta V Advantage at the Laboratory of Stable
Isotopes (LES) at the Geoscience Institute of the University of São Paulo. Isotopes are
reported in delta notation ($\delta^{18}$O and $\delta^{13}$C) relative to the Vienna Pee Dee Belemnite
(VPDB) standard, with uncertainties in the reproducibility of standard materials < 0.1 ‰.
The isotopic profiles of TRA5, TRA7, FN1 and FN2 stalagmites consist of 443, 885, 1215
and 651 isotope samples, respectively. These datasets provide an average resolution of
~1 year per sample for TRA5 and ~ 4 years for the other speleothem records. TRA7 $\delta^{13}$C
results were presented by Utida et al. (2020) and FN1 $\delta^{18}$O results by Cruz et al. (2009)
using the same methods. Cruz et al. (2009) do not provide FN1 $\delta^{13}$C results, which were
not included in this study.
Different textural characteristics of speleothem TRA5 and FN2 were identified in
intervals which were analyzed for mineralogical composition based on approximately 20
mg samples with X-ray powder diffraction in a Bruker D8 diffractometer (Cu Ka, 40 kV, 40
mA, step 0.02°, 153 s/step, scanning from 3 to 105° 2θ) at the NAP Geoanalítica
Laboratory of the University of São Paulo. Qualitative and quantitative mineralogical
analyses were performed with *Match!* and *FullProf* software, using the Crystallographic
Open Database (Grazulis et al., 2009). Crystallographic data for the mineral phases were
taken from Pokroy et al. (1989) for aragonite and from Paquette and Reeder (1990) for
calcite. Mineralogical results of TRA7 and FN1 were obtained by Utida et al. (2020) using
the same method. All results are presented in weight proportion (wt %). The $\delta^{18}$O results of
speleothems were calibrated according to the percentage of calcite identified for the
interval applying the aragonite–calcite fractionation offset of 0.85 ‰ ± 0.29 ‰ (Zhang et
al., 2014). The $\delta^{13}$C results were not corrected because the original aragonite–secondary
calcite fractionation factor is negligible (~0.1-0.2 ‰) (Zhang et al., 2014). Even considering
the original aragonite-original calcite mean fractionation factor of 1.1 ‰ (Zhang et al.,
2015), the range of $\delta^{13}$C RN stalagmites is very large (>8 ‰) and the correction would not
affect the main interpretation.
The intra-site correlation model (*iscam*) was used to construct a composite record
(Fohlmeister, 2012). It combined the climate records to obtain a unique age model and
oxygen isotopic record, corrected only for mineralogical composition of speleothems from
Rio Grande do Norte, which here is referred to as the RN Composite. The age-depth
modeling software was adjusted to calculate 1000 Monte-Carlo simulations on absolute
age determinations to find the best correlation between oxygen isotope records from
Trapiá and Furna Nova speleothems, reproducing adjacent archives. The results estimate
the error of the age-depth model by indicating the 68 %, 95 % and 99 % confidence
intervals obtained from evaluation of a set of 2000 first order autoregressive processes
(AR1) for each record (Table S3). This method allows significantly reducing the age
uncertainty within the overlapping periods and it can be tested if the signal of interest is
indeed similar in all the records (Fohlmeister, 2012). The age data were assumed to have
a Gaussian distribution and were calculated pointwise. The composite result was
detrended and normalized, according to the *iscam* method. The performance of the *iscam*
results is affected by low quality of chronological control, low resolution and hiatuses.
Therefore, the following intervals were removed from the stalagmite records before
constructing the RN Composite: FN1 0-12 mm and 187-202 mm, FN2 0-6 mm, TRA5 0-37
mm and TRA7 222-227 mm. In addition, the FN1 record was divided into two portions:
FN1a 12.14-136.99 mm and FN1b 140.15-186.87 mm that are separated by a hiatus. The
chronological age-depth relationship in the overlapping parts of the individual stalagmites
was modified and improved according to the *iscam* results of the composite record. The
composite calculation rearranges the proxies in order to obtain the optimal calculated age
and then calculates the average of the proxy data after normalizing the records. The RN
record only contains overlapping segments between two stalagmites per period. Hence the
RN composite proxy error can be quantified as the difference between the $\delta^{18}$O of the
stalagmites combined for any given point in time (Fig. S6).

*4. Results*
*4.1. Modern climatology and $\delta^{18}$O rainfall distribution*
The data from Pedra das Abelhas Station reveal that in the majority of years
(normal years - interquartile range) the rainy season persists from February to April, with
precipitation varying from 100 to 180 mm/month, and minor contributions occurring in
January and May (50-70 mm/month) (Fig. 2). During the drier years (lower quartile),
February has a reduced precipitation amount, similar to the amount in January during
normal years, as described above. The maximum precipitation of 90 mm/month occurs
between March and April. For wetter years (upper quartile), the rainy season starts in
January with more than 100 mm/month and lasts until May with almost 150 mm/month,
reaching values higher than 250 mm around March. These data show that wetter years
are characterized by increased precipitation amounts and a longer rainy season starting in
January and ending in May, while the precipitation deficit during drought years is a result
of decreased precipitation amount and a shorter rainy season, with a peak in precipitation
between March and April. The anomalous length of the rainy season during dry and wet
years is attributed to variations in the meridional SST gradient in the tropical Atlantic that
results in a shift of the ITCZ to the north or south of its climatological position (e.g.,
Andreoli et al., 2011; Marengo and Bernasconi, 2015; Alvalá et al., 2019).
In S-NEB, the precipitation occurs mainly during summer, from December to
February (Fig. 1a and 3b). This regional seasonality difference with N-NEB is evident in
the spatial correlation map between GPCC precipitation anomalies and $\delta^{18}O$ anomalies
obtained from IAEA-GNIP for Fortaleza and Brasília stations (Fig. 3). The reddish areas on
the map indicate significant negative correlations during the austral summer (DJF) and
autumn (MAM) between the local precipitation $\delta^{18}O$ signals and the regional precipitation
amount. Overall, the spatial correlations indicate that in both areas the amount effect is the
dominant effect on the isotopic composition of rainfall (Dansgaard, 1964). However, the
isotopic signal varies seasonally and as a function of the two different circulation systems.
The negative spatial correlation observed over N-NEB (Fig. 3a) suggests precipitation is
dominated by ITCZ dynamics, similar to the conditions over Fortaleza, while the negative
spatial correlation over S-NEB (Fig. 3b) is a result of the rainfall influenced by the SASM
(Fig. 1) (Vera et al., 2006), such as in Brasília City, in central Brazil. Therefore,
precipitation and the associated isotopic signal are the result of ITCZ dynamics in N-NEB,
while they are influenced by the SASM in the S-NEB. Accordingly, their rainfall seasonality
is also different (Fig. 3), with a NDJFM peak in the south (Brasília, Fig. 3b) and a MAM
rainfall peak in the north (Fortaleza, Fig. 3a).

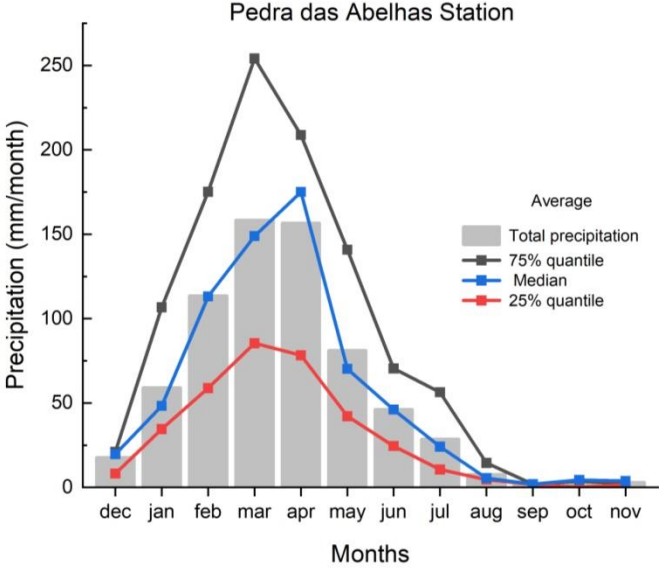


Figure 2 - Pedra das Abelhas ANA Station precipitation analyzed from 1911 to

2015 (n= 103), excluding the strongest ENSO years (39 years), according to Araújo et al.

(2013).


Another important region in SA affected by the ITCZ behavior is the eastern

Amazon, west of the NEB (Fig. 1 and Fig. 3c). This region is characterized by increased
precipitation during DJFMAM and a peak in rainfall and $\delta^{18}$O minimum in MAM (Fig. 3c) as
a result of precipitation received from the ITCZ in both summer and autumn. It can be
depicted by the negative correlation between $\delta^{18}$O at the Manaus GNIP station and rainfall
over the upstream equatorial region under direct ITCZ influence. In addition, there is only a
minor influence through water recycling over the Amazon Basin, due to its proximity to the
coast (Wang et al., 2017).

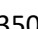


Figure 3 – Monthly mean observed precipitation amount collected at ANA and $\delta^{18}O$
values for GNIP stations (IAEA-WMO, 2021) (black dots) and correlation maps between
gridded precipitation and $\delta^{18}O$ anomalies from the same stations (black dots) for: (a)
Northern NEB, Fortaleza and Pedra das Abelhas stations (star 1), (b) Southern NEB,
Brasília and Andaraí stations (star 3), c) Eastern Amazon, Manaus and Belterra stations
(star 4). The maps show the spatial correlation between $\delta^{18}$O anomalies at GNIP stations
and GPCC gridded precipitation anomalies based on the period 1961-1990 for December
to February (DJF) and March to May (MAM) for Fortaleza, Brasília and Manaus stations
(Ziese et al., 2018). The $\delta^{18}$O values (left *y* axis) and precipitation (right *y* axis) for each
station were obtained from GNIP IAEA/WMO database. Stars indicate the site locations: 1)
Trapiá Cave, Furna Nova Cave and Pedra das Abelhas ANA Station (reference period
1910-2019), 2) Boqueirão Lake (Utida et al., 2019), 3) Diva de Maura Cave (Novello et al.,
2012) and Andaraí ANA Station (reference period 1960-1986), 4) Paraíso Cave (Wang et
al., 2017) and Belterra ANA Station (reference period 1975-2007), 5) Cariaco Basin (Haug
et al., 2001).

*4.2. Chronology and mineralogy*
The RN record covers the last 5000 years, four stalagmites cover the last 3250
years, and two of these stalagmites cover partially the time period between 3000 and 1260
Before Common Era (BCE), with the exception of one hiatus at 2100 -1720 years BCE
(Fig. 4, Table S1 and S2).
Stalagmite TRA7 from Trapiá Cave was deposited from 3000 to 2180 BCE (Fig.
S3) with a low deposition rate (DR) of approximately 0.05 mm/yr. After a hiatus of 1880
years, it resumed deposition from 300 BCE until 1940 CE with a DR of 0.18 mm/yr. The
TRA5 stalagmite deposition occurred continuously from 1490 to 1906 CE (Fig. S3) with a
DR of 0.33 mm/yr.
Stalagmite FN1 from Furna Nova was deposited over the last 3600 years, with a
hiatus from 125 to 345 BCE and another one of approximately 100 years between 1525
and 1662 CE (Fig. S3), with an average DR of 0.09 mm/yr. The ages from the FN1
stalagmite are all in chronological order and contain low errors and were therefore all kept
in the age model. The FN2 stalagmite deposited continuously from 1226 BCE to 7 CE,
except for a hiatus between 189 and 45 BCE (Fig. S3) with a DR of 0.20 mm/yr.

The mineralogy of the stalagmites from Trapiá Cave is formed by layers of crystals

with mosaic and columnar fabrics, composed exclusively of calcite, except for the base
portion of TRA7 from 173 to 270 mm (3000 BCE to 130 CE), which is described as an
interbedded needle-like crystals texture, composed of 87.1 to 99% of aragonite (Fig. S1,
Table S4). The same needle-like morphology is present in most of the Furna Nova Cave
stalagmites, composed of aragonite with a weight proportion greater than 85 % in FN1,
extending from 0 to 83 mm (160 to 1340 CE) and from 128 to 183 mm (1730 BCE to 80
CE). In the FN2 sample this weight proportion is greater than 93.4 % (1265 BCE to 35
CE). The only interval composed of 100 % calcite is from 95 to 125 mm in FN1 (Fig. S2,
Table S4). These speleothem samples show no sign of dissolution or recrystallization.

*4.3. Stalagmite $\delta^{18}O$ and $\delta^{13}C$*

The oxygen isotope ratios of the RN record vary from 0.6 ‰ to -4.5 ‰, with $\delta^{18}O$

mean values for each speleothem of -2.8 ‰ for TRA7, -3.5 ‰ for TRA5, -2.4 ‰ for FN1
and -1.5 ‰ for FN2. Similarities among the stalagmites are evident, especially around
1500 CE when $\delta^{18}O$ values abruptly decrease in TRA7 and TRA5, while in FN2 this period
features a hiatus (Fig. S4).

The $\delta^{18}O$ correction due to mineralogy for the stalagmites from Furna Nova Cave

resulted in changes of less than 0.1 ‰ of their mean values. The mean correction for
TRA7 equals an enrichment of 0.5 ‰ during the period spanning 130 BCE to 1940 CE.
Values from TRA5 were corrected along the entire sample by adding 0.85 ‰, as it is
composed of 100 % calcite. Therefore, the mean values increased from -3.5 ‰ to -2.7 ‰
(Fig. S4).

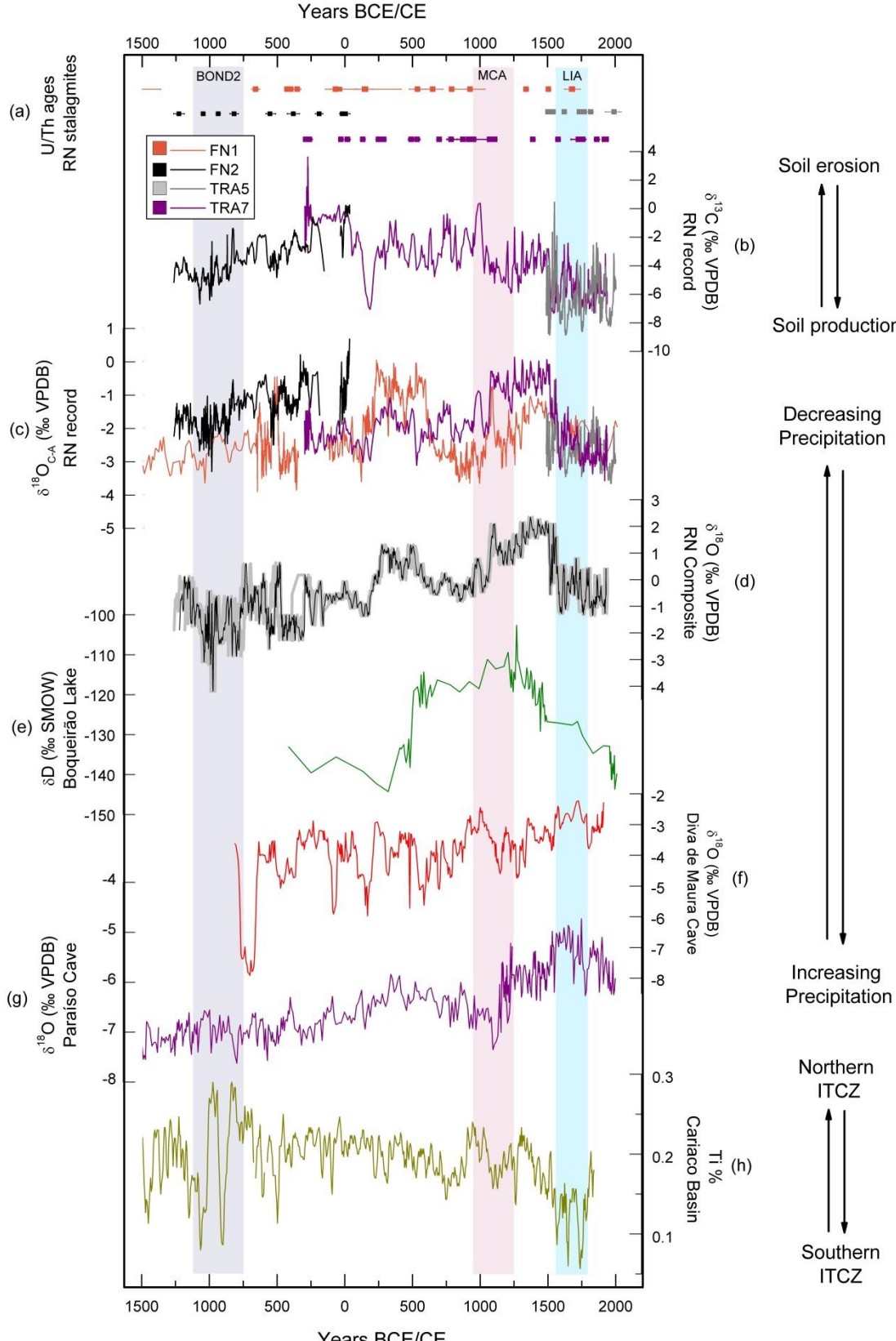

Figure 4 – Rio Grande do Norte stalagmite isotope records and comparisons with
other records from South America. a) U/Th ages from each stalagmite studied. b) Raw
data of $\delta^{13}$C. c) Oxygen isotope results corrected for calcite-aragonite fractionation ($\delta^{18}$O$_{C-}$
$_A$), according to weight proportion of mineralogical results. d) $\delta^{18}$O RN Composite
constructed using stalagmite records from NEB (black line). Grey shaded area denotes the
99 % confidence interval of the age model. Blue shaded area refers to LIA (Little Ice Age),
pink shaded area refers to MCA (Medieval Climate Anomaly), light grey shaded area
refers to Bond 2 event. e) Boqueirão Lake $\delta$D record (Utida et al., 2019). f) DV2
speleothem oxygen isotope record from Diva de Maura cave, southern NEB (Novello et
al., 2012). g) PAR01 and PAR03 $\delta^{18}$O records from Paraíso cave, eastern Amazon (Wang
et al., 2017).  h) Ti record of Cariaco Basin (Haug et al., 2001).

Four main phases describe the $\delta^{13}$C dataset (Fig. 4b). The oldest phase from 3000
to 2160 BCE is characterized by $\delta^{13}$C values close to zero. After a hiatus (2170-1270 BCE)
there is a short interval of stability with $\delta^{13}$C values around -4 ‰ that lasts from 1270 to
840 BCE and is followed by a $\delta^{13}$C enrichment that reaches a value of zero at 30 CE.
Between 30 and 1500 CE there is a trend toward more negative $\delta^{13}$C values, varying from
0 to -8.8 ‰. This interval is marked by a valley at 190 CE with $\delta^{13}$C values of -7.2 ‰ and a
peak at 1000 CE with $\delta^{13}$C values of 0.22 ‰. The youngest period, from 1500 to 1930 CE
is more stable than the previous one, with $\delta^{13}$C values averaging around - 6.4 ‰.

*4.4. Composite*
Combining the $\delta^{18}$O results from the four RN stalagmites allows establishing a
continuous record covering the last ~3200 years, the RN Composite (Fig. 4d). The
correlation coefficient (r) between each measured $\delta^{18}$O stalagmite time series is >0.59,
significant at the 95 % level (Fig. S5). The composite provides an average temporal
resolution of ~2 years. The entire stable isotope time series is composed of 2495 $\delta^{18}$O
measurements, corrected according to mineralogical composition.

*5. Discussion*
*5.1. U/Th chronology and RN Composite*
The high values of $^{232}$Th and low $^{230}$Th/$^{232}$Th ratio suggest incorporation of detrital
Th transported by the seepage solution to the speleothems, which lead to a higher
uncertainty of the age values. Recrystallization of aragonite into calcite might also reduce
the U content and given older ages for carbonates (Lachniet et al., 2012). We assume that
these are the main reasons for age inversions along speleothems from Northeast Brazil.
Because FN1 is mostly composed of aragonite and presents low U concentration in
some samples of the first 127 cm and high $^{232}$Th amounts, we considered the association
of low $^{230}$Th/$^{232}$Th and low U content the most important factor affecting the age errors and
inversions in the FN1 stalagmite. In contrast, the FN2 stalagmite has a more precise
chronology due to the predominant aragonite composition, with high $^{238}$U content and
higher $^{230}$Th/$^{232}$Th ratio than FN1. Although the TRA5 stalagmite is entirely composed of
calcite, the $^{238}$U content is relatively high compared to other stalagmites, which improves
the confidence in its age results. The high $^{232}$Th contamination of TRA5 samples is the
main factor attributed to cause age inversions and increased errors. According to age
results produced by Utida et al. (2020), most of the TRA7 ages are in chronological order
and the inversions seem to not have a direct relationship with $^{238}$U amount, and the high
$^{232}$Th content is similar to other ages from TRA7. Most of the TRA7 stalagmite used in our
composite is composed of calcite and might not affect the main trends of $\delta^{18}$O.
The age uncertainties caused by high $^{232}$Th concentration and calcite
recrystallization in stalagmites might affect the age model. However the strong coherence
between the $\delta^{18}$O curves from different stalagmites argues in favor of the good quality of
our chronology. This is evident when FN2, which is composed 100 % of aragonite, is
compared with other samples. There is a different amplitude range in its $\delta^{18}$O values, but
when the curve is superposed on other $\delta^{18}$O records the variability is similar. This
amplitude range is corrected when the $\delta^{18}$O results are submitted to the *iscam* composite
construction, since it normalizes the results (Fig. S6).

Although the $\delta^{18}$O results present a different range of values between FN2 and

FN1, the mineralogical correction did not significantly change the main curves (Fig. S4).
TRA7 and FN1 underwent substantial changes due to mineralogical corrections between
80 to 1500 CE (Table S4). However the $\delta^{18}$O trends were not modified. The mineralogical
correction for the last 500 years, adjusts the $\delta^{18}$O values over the same range for TRA5,
TRA7 and FN1 (Fig. S4). Some of this $\delta^{18}$O variability might also be attributed to karst
fractionation effects. However, no cave monitoring in northern NEB is available that could
quantify the extent of these processes.

These differences in mineralogical corrections and possible $\delta^{18}$O fractionations did

not alter the general shape of the RN Composite. Before merging the results, *iscam*
normalizes the $\delta^{18}$O and different range values are adjusted to the same scale, resulting in
significant reduction in the difference between stalagmite records (Fig. S6). The largest
error occurs between 250 and 580 CE, when the maximum and minimum values of FN1
and TRA7 are 2.4 ‰ and -1.50 ‰ after normalization, respectively (Fig. S6). This is a
period when FN1 registers high $\delta^{18}$O values; an anomaly that is not evident in TRA7. The
period extending from 500 to 570 CE, is characterized by an anti-phased signal between
FN1 and TRA7, and hence the RN Composite shows a smoothed signal during this time.

*5.2. Paleoclimate interpretation*

The variability of the global $\delta^{18}$O values for speleothems originating from the same

cave is ~ 0.37 ‰, which can be attributed to karst fractionation effects and not directly to
hydroclimate, host rock geology, cave depth or cave microclimate instability (Treble et al.,
2022). Some intervals in coeval RN stalagmites from the same cave are above this limit,
however, we demonstrated based on the composite treatment associated with
mineralogical corrections that the $\delta^{18}$O variability from the RN record is similar for
stalagmites from the same cave and between the two studied caves throughout the period
analyzed, further reinforcing the notion applied by previous studies that these records can
be interpreted in a paleoclimatic context (Cruz et al., 2009; Utida et al., 2020). In addition,
we consider the RN composite as representative of a precipitation $\delta^{18}$O signal, since the
differences between stalagmite records are significantly reduced after age rearrangements
and isotope normalization.

The $\delta^{18}$O RN Composite allowed us to reconstruct precipitation changes influenced

by the ITCZ position in N-NEB and its convective intensity. This interpretation is based on
the spatial correlation between $\delta^{18}$O at GNIP stations and GPCC precipitation (Fig. 3).
Highest precipitation amounts occur between March and May and they coincide with more
depleted $\delta^{18}$O precipitation signals, consistent with the amount effect (Dansgaard, 1964).
Hence, the most negative $\delta^{18}$O values in RN stalagmites reflect an increased rainfall
amount, as a consequence of an ITCZ position close to N-NEB (Cruz et al., 2009; Utida et
al., 2019).

A generally drier climate prevailed in NEB after the 4.2 ky BP (kiloyear Before

Present) event in the Mid-Holocene (Cruz et al., 2009). This led to the development of the
Caatinga, a sparse vegetation cover which has persisted in NEB to the present (De
Oliveira et al., 1999; Utida et al., 2020; Chiessi et al., 2021). These drier conditions favored
soil erosion during rainfall events and reduced soil thickness (Utida et al., 2020). When
erosion events remove most of the soil cover, there is an increase in the carbon
contribution from local bedrock (mean $\delta^{13}$C of 0.5 ‰), which leads to higher $\delta^{13}$C values in
the NEB stalagmites from RN. On the other hand, more negative $\delta^{13}$C values in
stalagmites are associated with increased soil coverage and soil production (Utida et al.,
2020). In NEB soils have a $\delta^{13}$C average around -25 ‰, which suggests a dominant
influence from C3 plants with $\delta^{13}$C values ranging between -32 ‰ and -20 ‰ (Pessenda et
al., 2010). Therefore, the $\delta^{13}$C stalagmite results are interpreted as changes in soil
production/erosion and the density of vegetation coverage (e.g., Utida et al., 2020;
Azevedo et al., 2021; Novello et al., 2021).

The oldest period covered by the RN Composite, from 1200 to 500 BCE, is

characterized by successive dry and wet multidecadal periods, with increased precipitation
in N-NEB from 1060 to 750 BCE and from 460 to 290 BCE, as suggested by the negative
departures seen in the $\delta^{18}$O values. During this last period, there is also a tendency from
lower to higher $\delta^{13}$C values, suggesting progressive surface soil erosion related to rainfall
variability (Fig. 4), as interpreted by Utida et al. (2020). This period ends up in a stable
interval, lasting from 300 BCE to 0 CE, with little fluctuation in $\delta^{18}$O values and $\delta^{13}$C values
close to the bedrock signature at about -1 ‰ to +1 ‰, indicating a lack of soil above the
cave. After an abrupt reduction of both isotopes around 200 CE, there was a brief time of
increased precipitation and vegetation development. Between 200 CE and 1500 CE,
decreased $\delta^{13}$C values, reaching approximately -2 ‰, suggest a vegetation development
above the cave. However, $\delta^{18}$O values indicate significant variability with two main periods
of dry conditions, from 270 to 530 CE and 1060 to 1500 CE. From 1500 CE to the present,
more negative values of $\delta^{18}$O represent wetter climatic conditions. The more negative $\delta^{13}$C
during this period can be related to denser vegetation that favored both soil production and
stability above the cave. Due to the high range of $\delta^{13}$C results (more than 11 ‰), we
assume that the Prior Calcite Precipitation effect is negligible in our results. In addition, a
more positive $\delta^{13}$C signal occurs around 280 BCE when the climate conditions were not
the driest in the last 5000 years, thus probably representing a local environmental change.

During the last 2500 years, the RN Composite shows similar characteristics as the

lower-resolution $\delta$D lipids record (n-C28 alkanoic acid from leaf waxes) obtained in
Boqueirão Lake sediments (N-NEB) (Figs. 1 and 4). Both records show a more stable
climatic signal between 400 BCE and 350 CE. From 500 to 1500 CE, enriched $\delta$D lipids
obtained in Boqueirão Lake were interpreted as the beginning of a long dry phase (Utida et
al., 2019), although the beginning of the dry period is slightly delayed when compared with
the RN speleothem isotope record. This inconsistency might be related to different
chronological controls between lake and stalagmite records and possibly also by the
location of Boqueirão Lake that is affected by the ITCZ and winter breezes as it is located
in the eastern coastal sector of NEB (Zular et al., 2018; Utida et al., 2019).

It is important to note that the RN record exhibits a climatic signal that is distinctly

different from the from DV2 speleothem record from Diva de Maura Cave in S-NEB
(Novello et al., 2012). Although both regions are affected by the same mesoscale
atmospheric circulation, the RN site receives its precipitation directly from the ITCZ. At the
S-NEB site, on the other hand, the primary source of precipitation is associated with the
monsoon, as it is located too far inland to be affected directly by the ITCZ, as
demonstrated by the correlations maps (Fig. 3). The general trend toward more positive
values, as a result from insolation forcing, occurs from 150 to 1500 CE in the RN
Composite, but from 600 to 1900 CE in the DV2 sample (Cruz et al., 2009; Novello et al.,
2012). This trend is a result of the persistent dry conditions in the entire NEB region that
suggests an ITCZ contraction in an orbital timescale, resulting in drier conditions over NEB
during periods of maximum austral summer insolation (Cruz et al., 2009; Chiessi et al.,
2021; Campos et al., 2022). However, the DV2 record does not document the same
multidecadal and centennial-scale climate variability as recorded in the RN speleothem
record, nor the less dry interval from 600 to 1060 CE seen in the RN Composite (Fig. 4).
As demonstrated by the spatial correlation maps between $\delta^{18}O$ values and regional
precipitation (Fig. 3), the S-NEB and N-NEB regions are influenced by distinct rainfall
regimes whose peaks of precipitation arise during the summer monsoon season and the
autumn ITCZ, respectively. Our data provide evidence for a spatial and temporal
distinction of NEB climate patterns for the past that can be interpreted as differences in
seasonality during the last millennia. Furthermore, contemporaneous dry or wet events in
both N-NEB and S-NEB suggest the occurrence of larger regional climate changes with
higher environmental impacts.
When comparing N-NEB and eastern Amazon conditions, it is evident that the RN
Composite shares some similarities with the Paraíso stalagmite record (Wang et al.,
2017), due to the contribution of ITCZ precipitation in both places. But there are also
important differences (Fig. 4). The RN Composite shows lower $\delta^{18}O$ values between 500
and 1000 CE, compared to the earlier period, while Paraíso shows gradually decreasing
values around the same period, suggesting a slight increase in precipitation in both areas.
From 1160 to 1500 CE, abrupt increases in $\delta^{18}O$ values are seen in both records, which
indicate abrupt and prolonged drought conditions due to a northward ITCZ migration.
However, around 1100 CE, centered in the MCA, and the period from 1500 to 1750 CE,
Paraíso is antiphased with the RN Composite and in phase with the Cariaco Basin (Haug
et al., 2001), which is inconsistent with the notion of an ITCZ-induced regional precipitation
change. Instead, a zonally-oriented precipitation change within the ITCZ domain over
Brazil is required to explain the anti-phased behavior between precipitation in N-NEB and
the eastern Amazon, and similarities between Cariaco and the eastern Amazon.

We investigate the potential relationship between $\delta^{18}O$ values in our RN

speleothems and an ITCZ displacement toward the warmer hemisphere to explain
paleoclimate variability observed in N-NEB. In order to test this hypothesis, the RN
Composite was compared with a reconstruction of Atlantic Multidecadal Variability (AMV)
(Lapointe et al., 2020) (Fig. 5). Some studies suggest that the warm phase of the AMV
(when the North Atlantic presents warm SST) forces the mean ITCZ to shift to the north of
its climatological position, thereby causing a reduction in NEB rainfall (Knight et al., 2006;
Levine et al., 2018), while a recent study suggests that the warm phase of the AMV would
cause a weakening of the ITCZ from February to July (Maksic et al., 2022). The driest
periods from 750 to 500 BCE, 200 to 580 CE and 1100 and 1500 CE occurred during long,
relatively warm AMV anomalies. The warm average temperature of 22.19° C for the
period, would force a northward ITCZ displacement or an ITCZ weakening, and in both
cases the result is low precipitation over NEB. The lowest AMV temperature (cold phase)
around 1500 CE might be related to the abrupt dry conditions seen in the RN Composite
and suggests an increased equatorial Atlantic SST, and consequently increased
precipitation over N-NEB (Fig. 5). Opposite conditions between the RN Composite and the
AMV can be observed during the Current Warm Period, which requires further
investigation. The relationship between North Atlantic temperature and ITCZ location can
also explain the Bond 2 Event recorded in the RN Composite. It is marked by increased
precipitation around 1000 BCE, when the ITCZ was displaced toward the south. This
southerly ITCZ displacement might be attributed to persistently lower temperatures in the
North Atlantic (Bond et al., 2001; Broccoli et al., 2006) caused by the slowdown of the
Atlantic Meridional Overturning Circulation (Jackson et al., 2015).

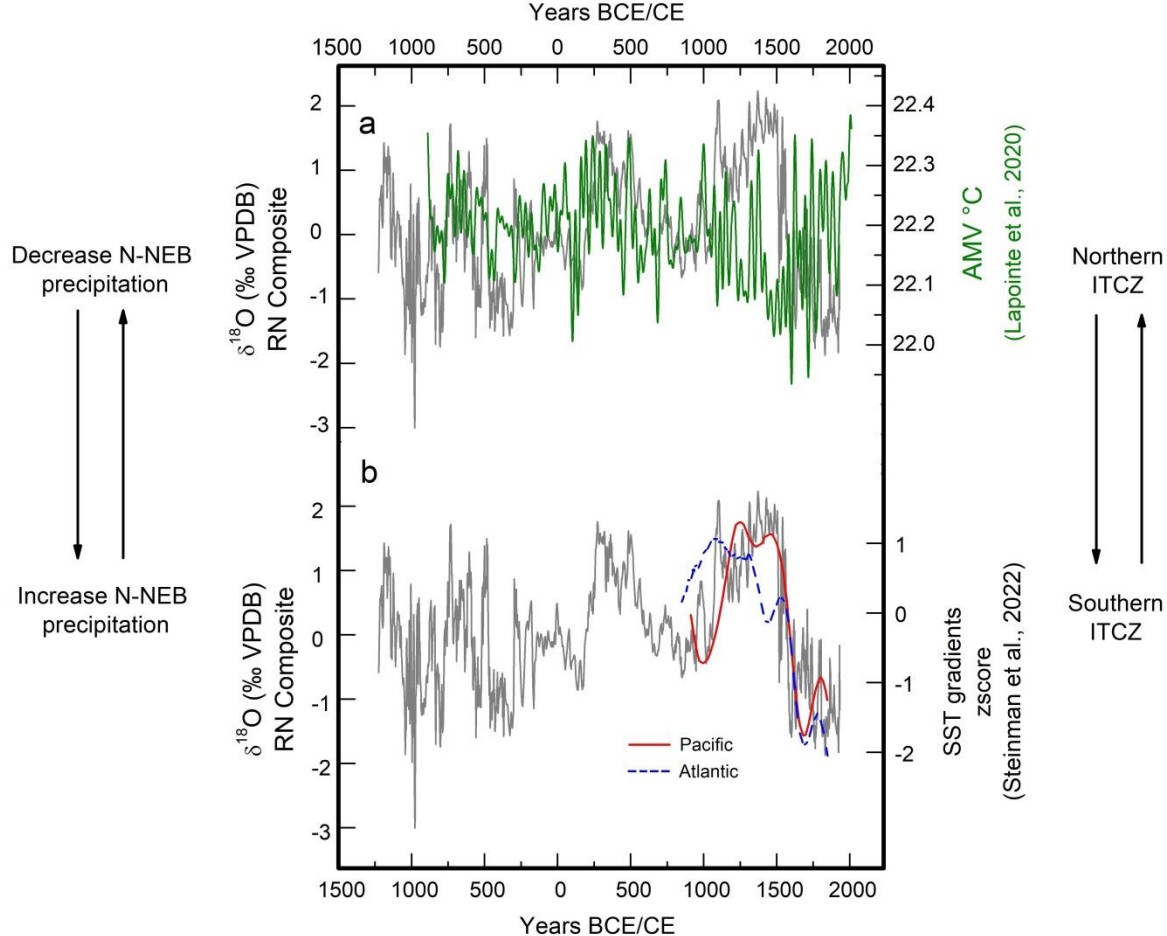


Figure 5 - $\delta^{18}$O RN Composite compared with (a) Atlantic Multidecadal Variability

(Lapointe et al., 2020) and (b) Pacific and Atlantic Sea Surface Temperature gradients

calculated (z-score) according to Steinman et al. (2022). Atlantic: 2 σ range of 1000

realizations of the Atlantic meridional SST gradient (north – south). Pacific: median of 1000

realizations of the Pacific zonal SST gradient (west – east).

Steinmann et al. (2022) suggested a southward displacement of the ITCZ during

the Common Era toward the southern hemisphere in response to changes in the Pacific

and Atlantic meridional SST gradients. Indeed, our RN Composite is dynamically

consistent with these SST gradient changes and in agreement with the hypothesis of a

north-south oscillation of the latitudinal ITCZ position in the tropical Atlantic during the last

millennia, modulating precipitation over N-NEB. When the tropical South Atlantic and
tropical eastern Pacific are anomalously warm – negative z-score (cold - positive z-score)
(Fig. 5) the ITCZ is displaced to the south (north), resulting in increased (decreased)
precipitation over NEB. The abrupt changes in N-NEB precipitation around 1100 and 1500
CE occur approximately synchronous with the SST gradient changes, confirming how
sensitive the RN speleothems respond to changes in the ITCZ latitudinal position (Fig. 5).
The same is observed during the period equivalent to the LIA, between 1560 and 1800 CE
considering N-NEB, S-NEB and eastern Amazon records, when both Pacific and South
Atlantic became warmer (Fig. 5). According to Steinmann et al. (2022), during the LIA
period warm SST in the eastern tropical Pacific and in the tropical South Atlantic would
promote a southward displacement of the ITCZ. This is supported by other records from
the western Amazon and the tropical Andes that document an intensified SASM during the
LIA, fueled by the southern location of the ITCZ (e.g., Vuille et al., 2012; Apaéstegui et al.,
2018), which is also very well recorded in other archives around the tropics (Leichleitner et
al., 2017; Campos et al., 2019; Orrison et al., 2022; Steinmann et al., 2022).
According to Kayano et al. (2020, 2022), during the last century, dry conditions
over N-NEB and the eastern Amazon are present when AMV and Pacific Decadal
Variability (PDV) are both in their warm phases, or when the AMV is in a cold phase and
the PDV in its warm phase. On the other hand, when AMV and PDV are both in their cold
phase, precipitation over the Amazon is anti-phased with NEB, resulting in decreased
precipitation over the Amazon and increased precipitation over NEB. This zonally aligned
precipitation signal over eastern tropical South America is the result of joint perturbations
of both the regional Walker and Hadley Cell's produced by teleconnection between the two
ocean basins (He et al., 2021). This joint interaction between the two basins can help
explain the results seen during the cold AMV phase between 1500 and 1750 CE (Fig. 5),
when precipitation over N-NEB increased, but the eastern Amazon saw a decrease in
precipitation (Fig. 4).

*5.3. TRA5 $\delta^{18}O$ stalagmite and extreme drought events*
The last 500 years were the wettest of the last two millennia and the onset of this
period was forced by Atlantic and Pacific SST, according to our results (Figs. 4 and 5).
Superimposed on these long-term negative $\delta^{18}O$ anomalies, distinct peaks are recorded in
the TRA5 $\delta^{18}O$ record from 1500 to 1850 CE (Fig. 6). These drought events are visible in
this record thanks to its higher deposition rate (faster growth) and thus higher temporal
resolution of the $\delta^{18}O$ record when compared to other stalagmites used in our study. No
preferred periodicity of these events is apparent in our record, preventing comparison with
ENSO events, for example. There exist no precipitation reconstructions or observations
from this region between 1500 and 1850 CE, aside from historical drought records.

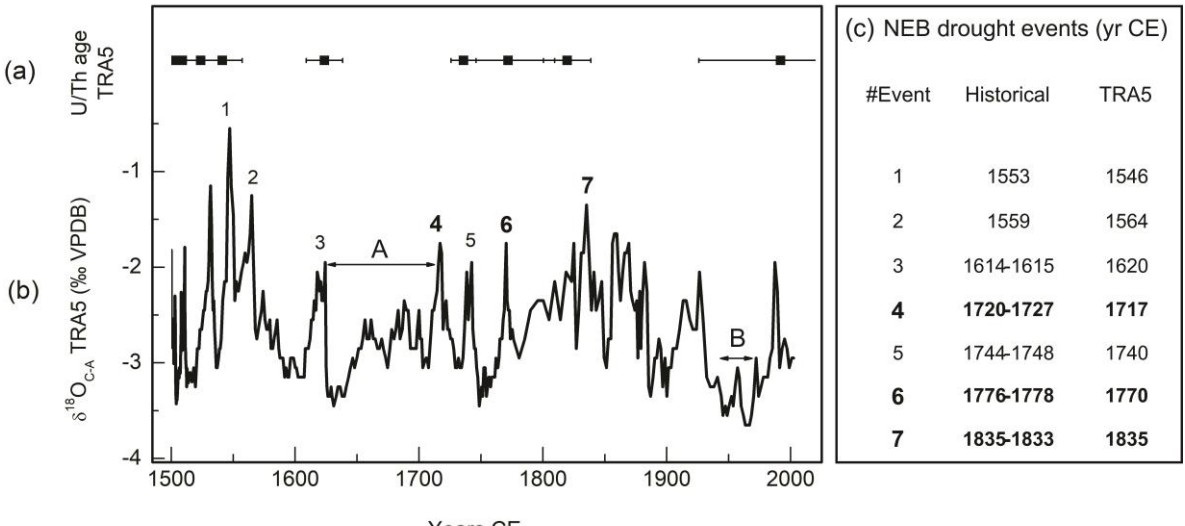


Figure 6 – TRA5 record and equivalent historical record. (a) U/Th age is
represented by black dots and horizontal lines indicate age uncertainty. (b) $\delta^{18}O_{C\text{-}A}$ record,
numbers represent the peak of a drought event. Bold numbers represent the most severe
drought events. A - Few drought events interval from 1620 to 1970s period. B - 1940s to
1970s period. (c) Occurrence of historical drought years compiled from Lima and
Magalhães (2018).

Although the age model errors of TRA5 are larger and could limit our ability to

attribute $\delta^{18}$O peaks to specific single-year events, it still allows for a comparison between
these abrupt events with historical records to demonstrate the long-term context of abrupt
drought events in modern human history. We thus consider our speleothem-based record
as a first attempt to reconstruct precipitation in Northeast Brazil that would allow for a
comparison with historical droughts. If our speleothem records regional hydroclimate, it
should retain a signal of the most intense droughts over NEB that are known to have
struck the region based on the available historical literature of Brazil.

The highest peaks correspond to extreme drought events, such as the ones

centered around 1546 and 1564 CE (points 1 and 2 of Fig. 6). They can be associated
with observed historical droughts that took place in 1553 and 1559 CE. These were the
first two events recorded in Brazil by the Portuguese Jesuits that led to a reported
reduction in riverflow in the tributaries of the main rivers of NEB (Serafim Leite, 1938; Hue
et al., 2006; Lima and Magalhães, 2018).

Another relevant drought according to TRA5 is centered around 1620 CE (point 3

of Fig. 6). This drought is recorded in historical documents and lasted from 1614 to 1615
CE, although it did not have the same socioeconomic impact as the two prior droughts
(Lima and Magalhães, 2018). In fact, between the 16th and 17th century there are few
historical drought records (period A in Fig 6). One hypothesis to explain this hiatus is the
low population density of the NEB territory, resulting in poor historical documentation of
such events. However, according to the TRA5 record, between the event 2 ~1564 CE and
event 4 ~1717 CE (Fig. 6), the only drought peak occurs in 1620 CE, confirming an almost
150-year long period of relative climate stability with prevailing wet conditions in NEB.
These favorable conditions certainly helped with the initial population establishment at the
beginning of 16th century, and led to the peak era of sugar cane production in NEB around
1650 CE along coastal areas (Taylor, 1970).

During the 18th century NEB experienced a significant increase in rural population,

characterized by the establishment of large cattle farms (Fausto, 2006). In this period,
three droughts are documented in the TRA5 record (Fig. 6). The $\delta^{18}$O excursion around
1717 CE (point 4 in Fig. 6) can be associated with the drought that lasted from 1720 to
1727 CE; the first big drought in NEB, which according to historical documents, caused the
mortality of wildlife and cattle, and affected the agricultural productivity. Entire Indigenous
tribes died of starvation as a consequence of this drought and a concurrent smallpox
(variola) epidemic, which also killed other ethnic groups, especially the native population
and black people enslaved during that period (Alves, 1929).

The following event around 1740 CE (point 5 in Fig. 6) was also recorded in

historical documents, but did not seem to be associated with major impacts. However, all
of these droughts were probably responsible for a drop in sugar-cane exports to Europe
during the first half of the 18th century (Galloway, 1975).

Another drought occurred from 1776 to 1778 CE, and is imprinted in our record

around 1770 CE (point 6 in Fig. 6). This event was again accompanied by a variola
outbreak probably spread by a lowering in the sanitary conditions and increased people
agglomeration. The association between this disease and droughts might explain the
economic and health crisis, since people started to migrate to the cities looking for
treatment and food, leading the Brazilian Governor to transfer infected people to isolated
lands, resulting in thousands of deaths (Rosado, 1981). Finally, the most recent peak in
our data displays an event around 1835 CE (point 7 of Fig. 6), associated with a drought
that lasted from 1833 to 1835 CE, reaching the northernmost areas of NEB, and leading to
the largest human migration to other Brazilian regions (Lima and Magalhães, 2018). The
droughts centered around 1770 and 1835 CE had a huge impact on society according the
historical records (Lima and Magalhães, 2018).
Although the precision of the TRA5 speleothem chronology is reduced during the
last ~150 years, we observe that the wet period from the 1940s to the 1970s (line B in Fig.
6) is coincident with the mid-20th century break in global warming that has been discussed
as being forced by aerosol emissions (e.g., Booth et al., 2012; Undorf et al., 2018). Our
data suggest an increased precipitation in this period that is supported by a trend in
decreasing values of $\delta^{18}O$ in corals from the northeast coast of N-NEB, equally interpreted
as an ITCZ southward displacement caused by a decreasing SST gradient between the
North and South Atlantic (Pereira et al., 2022).
Our TRA5 stalagmite data record some of the most important droughts that
occurred in NEB between the 16th and the 18th centuries, demonstrating the potential of
stalagmite studies in monitoring abrupt and extreme climate events through time.
However, the speleothems do not record all documented historical dry events, as some
droughts may not have affected the Trapiá Cave region, or they were not strong or long
enough to affect the isotopic signal of the groundwater storage in the epikarst.
Furthermore, the period between 1620 and 1717 CE is devoid of any abrupt drought
events in the TRA5 stalagmite, which is again consistent with the historical records. It is
also important to mention that Lima and Magalhães (2018) report all drought events in
NEB and do not indicate their location. We suggest that progressive changes in the mean
ITCZ position along the last 500 years might be responsible for historical droughts that
affected the seasonality of N-NEB and caused abrupt and strong drought events.
Additional drought-sensitive high-resolution records will be required to improve our
understanding of these historical droughts events in NEB.

*6. Conclusions*


We present the first high-resolution record for the ITCZ in N-NEB that covers the
last 3200 years and also records the major historical droughts that took place in NEB
during the last 500 years. Based on stalagmite oxygen isotopes, we describe the regions'
paleoclimate variability for the last 2500 years and its connections to remote forcing
mechanisms such as the AMV and changes in Pacific and Atlantic SST gradients.
The N-NEB record presents a trend toward drier conditions from 1000 BCE to 1500
CE as is also being observed in the Diva de Maura Cave in S-NEB, interpreted as an ITCZ
contraction and SASM weakening on an orbital timescale, respectively. Although the two
records are influenced by distinctly different climate systems with different precipitation
seasonality, ITCZ and SASM dynamics are known to be closely linked (Vuille et al., 2012).
During the last millennia, ITCZ dynamics in the tropical Atlantic – South America
sector cannot be explained solely by north-south ITCZ migrations or one single forcing
mechanism. We propose a zonally non-uniform behavior of the ITCZ during the event
centered around 1100 CE and the drought events between 1500 and 1750 CE, when the
RN record is anti-phased with the Paraíso cave record from the eastern Amazon. This
zonal behavior would be forced by the interactions between AMV and PDV modes that
changed the regional Walker cell position and ITCZ intensity/extent and thus affecting
precipitation variability between the eastern Amazon and N-NEB.
The historical droughts discussed are the longest drought events in Northeast
Brazil that occurred within the zone of influence of the ITCZ, and are thus probably the
most likely to be recorded by stalagmites, according to our interpretation. The northern and
southern NEB are influenced by different climatic systems, the ITCZ and SASM,
respectively, and this can explain, in part, the differences between historical and
stalagmite records of Rio Grande do Norte. These historical droughts recorded in the RN
stalagmite suggest that much of the socioeconomic development of the NEB, which
occurred after 1500 CE, benefitted from conditions that were unusually humid in a long-
term context. During the last 500 years the technological development, infrastructure,
civilization and population growth relied on more abundant resources. On the other hand,
our data also shows how short, abrupt drought events significantly affected human
population and other life forms, especially when associated with anthropogenic changes in
the environment. These droughts induced an environment favorable for spreading of
disease, starvation, lack of water, environmental degradation and crowding of people
seeking help, among other problems. These events demonstrate the social and
environmental impacts associated with extreme events in this vulnerable environment and
our speleothem work documents the enormous potential of these archives to reconstruct
the drought history in this region.

*Acknowledgments*
We thank Alyne Barros M. Lopes, Osmar Antunes and Christian Millo (LES-IGc-
USP, Brazil) for their support during the analyses. We thank M.E.D.-L.G, J.C.R., E.A.S.B,
V.A. and W.D. for their support in U/Th analysis. We are grateful to Dr. Cristiano Chiessi
for comments on the draft, and Leda Zogbi and Diego de Medeiros Bento for the Trapiá
cave and Furna Nova maps. We thank Jocy Brandão Cruz, Diego de Medeiros Bento,
José Iatagan Mendes de Freitas, Darcy José dos Santos, Uilson Paulo Campos
(CECAV/RN), Antônio Idaelson do Nascimento and Geilson Góes Fernandes for all their
support during the field trip, information and data about the caves. This work was
supported by the São Paulo Research Foundation (FAPESP), Brazil through PIRE NSF-
FAPESP [2017/50085-3 to F.W.C], as well as the fellowships to G.U. [2020/02737-4;
2021/12860-0; 2022/14915-0], V.F.N [2016/15807-5], J.M. [2018/23522-6] and A.A.
[2020/09258-4]. The United States NSF support through grants [AGS-1303828 and OISE-
1743738] to MV and 1103403 to R.L.E and H.C. is acknowledged. The NSFC, China
support through grant [NSFC 41888101] to H.C. and [NSFC 42261144753] to H.Z. is
acknowledged, G.U. is grateful to CAPES for the PhD and PostDoc fellowships through
the Programa de Pós-Graduação em Geoquímica e Geotectônica at Universidade de São
Paulo, Brazil.

*Data availability*

The dataset generated as part of this study will be available in the PANGAEA

website.

*Author contribution*

G.U. and F.W.C designed the experiment, performed isotopic analysis and

prepared the manuscript with help from the coauthors; F.W.C. directed the project and
revised all versions of manuscript; M.V. helped with the interpretation and revision of the
manuscript; A.A. contributed with statistical analysis and interpretation; V.F.N. contributed
with the paleoclimate interpretations and revision of the manuscript; G.S. and J.M. helped
with interpretation and revision of the manuscript; F.R.D.A. provided and interpreted the
mineralogical analysis; H.Z. helped with U/Th analysis and revision of the manuscript, and
H.C. and R.L.E. coordinated the laboratory procedures for U/Th analysis.

*Competing interests*

The authors declare that they have no known competing financial interests or

personal relationships that could have appeared to influence the work reported in this
paper.

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
