# Peer review of "Spatiotemporal ITCZ dynamics during the last three millennia in"

_Climate of the Past, 2023_

## Referee Comment (RC1)

"Spatiotemporal ITCZ dynamics during the last three millennia in northeastern Brazil and related impacts in modern history" presents a new composite speleothem $\delta^{18}O$ record (using new data and previously published data) as well as a new $\delta^{13}C$ record used to characterise precipitation and vegetation/soil cover over northeast Brazil for the Late Holocene. The authors make clear links to the necessity for this research in South America, and frame it within the context of the increased proportion of the Brazillians who experience water scarcity in modern times. By analysing samples taken from sites at the southernmost extent of the ITCZ, they are able to link periods of changed precipitation to the movement of the ITCZ.

**Strengths**

This is relevant research with tangible outcomes for policy. Combining multiple stalagmite proxies can overcome some of the drawbacks encountered by single-proxy studies. It is great to see the continued use of already-published data, supplemented by new data. I really enjoyed the links between the proxy record and historical climate events – finding historical climate information is non-trivial, well done to the authors for their persistence. The introduction and study set-up is good.

**Weaknesses**

The main weakness of the manuscript is that there is no consideration of the impact of hydrological processes on speleothem $\delta^{18}O$, the primary proxy of the study. Treble et al. (2022) showed in a global analysis of coeval calcite and dripwater samples that karst hydrology exerts a control on speleothem $\delta^{18}O$, and that the variability of $\delta^{18}O_c$ can exceed that which can be attributed to rainfall $\delta^{18}O$. In the absence of cave monitoring data in the paper, the authors should add some discussion of how the karst processes at each site impact their results (or could impact their results) and how the composite handles this variability. The introduction/literature review should do also do a more thorough job of what controls $\delta^{18}O$ in NEB. The RN composite appears to only have uncertainty in the time domain, while other composites (e.g. Kaufman et al., 2020) include uncertainty in the composited proxy value.

**Specific comments and questions**

1. Figure 1

Please shade either the land or the ocean to differentiate them. Please choose an accessible colour palette – the rainbow colour palette is not useful for colour blind readers.

2. Line 163: please clarify whether you analysed the precipitation data as annual (or hydrological year), monthly, or daily totals.
3. Figure 2

Please change green dots to another colour (black?). Please also change the green line in the top panel to a different colour.

Consider changing the red-blue colour palette – in maps this palette is often used to show temperature variability, and so I find it slightly misleading here.

Please change the legend in the top panel to 'Site precipitation – GNIP' and 'Site precipitation – ANA' to be consistent with 'Site $\delta^{18}O$ – GNIP'.

The caption suggests that the correlation map correlates observed precipitation against observed $\delta^{18}O$ – suggested rephrase: "Figure 2 – monthly mean observed precipitation amount for ANA stations and $\delta^{18}O$ values for GNIP stations (IAEA-WMO, 2021) (green dots), with correlation maps between gridded precipitation anomalies and GNIP $\delta^{18}O$ anomalies...." And then carry on from (a) with the rest of your caption, while also adding (star 1) at line 201 for Pedra das Abelhas station.

Please clarify what correlation was used.

The difference between GNIP rainfall amount and ANA rainfall amount is really large between Fortaleza and Pedras de Abelhas. These sites are so close, have you double checked that that is correct?

4. Line 184: add reference to Fig 2.
5. Line 190: add ref to Fig 2C
6. Line 208: why 1960 – 2016 as a reference period?  The WMO uses 1961-1990 for long-term monitoring, or the 3 decades prior to the most recent year ending in 0 (e.g. 1991 – 2020) for short term changes. Could you please justify your choice or change to a standard ref. period.
7. Line 216: Figure 2C.
8. Line 272: typo, please correct to 'would not affect'
9. The $\delta^{18}O$ data are of different resolutions – can you please clarify how the iscam handles differently-sampled data
10. Line 331: please change 'first 1800 years' to 'the period spanning 1940 CE to 130 BCE' for less ambiguity.

More detail is needed about the C-A correction and how it was calculated (this could go in the Supplement. Could you please add the initial mean and corrected mean $\delta^{18}O$ values for each interval to your Table S3. Something like the below?

Table S3 – Speleothem intervals according to texture and mineral weight proportion (wt). Texture description: A - crystals with mosaic and columnar fabrics; B - interbedded needle-like crystals. *Obtained by Utida et al. (2020).

| Speleothem Mineralogy | | | | | |
|---|---|---|---|---|---|
| Sample | Interval (mm) | Age (yr BCE/CE) | Texture | Aragonite (wt %) | Calcite (wt %) |
| TRA5 | 30-54 | 1855 to 1745 CE | A | 0.0 | 100.0 |
| | 54-87 | 1745 to 1640 CE | A | 0.0 | 100.0 |
| | 87-108 | 1640 to 1565 CE | A | 0.0 | 100.0 |
| | 108-178 | 1565 to 1490 CE | A | 0.0 | 100.0 |
| TRA7* | 0-173 | 1940 CE to 130 BCE | A | 0.0 | 100.0 |
| | 173-215 | 130 to 290 BCE | B | 99.0 | 1.0 |
| | 215-270 | 290 to 3000 BCE | B | 87.1 | 12.9 |
| FN1* | 0-27 | 1790 to 1170 CE | B | 85.2 | 14.9 |
| | 27-83 | 1170 to 610 CE | B | 90.6 | 9.4 |
| | 83-128 | 610 to 80 CE | A | 0.0 | 100.0 |
| | 128-202 | 80 CE to 1730 BCE | B | 94.5 | 5.5 |
| FN2 | 6-31 | 189 to 660 BCE | B | 94.7 | 5.3 |
| | 31-56 | 660 to 960 BCE | B | 94.8 | 5.2 |
| | 56-63 | 960 to 1,005 BCE | B | 94.8 | 5.2 |
| | 63-95 | 1,005 to 1,265 BCE | B | 93.4 | 6.6 |

11. Can you please move Figure 3 earlier in the manuscript.

12. Line 362-368: I suggest you reword this to demphasise the 4.2 ka event (which your record mostly postdates). Something like "A generally drier climate prevailed in NEB after the 4.2 ky BP (Before Present) event in the Mid-Holocene (ref). This led to the development of the Caatinga, a sparse vegetation cover which has persisted in NEB to the present (ref). These drier conditions …."

13. Line 368-9: it is unclear if this is statement 'more negative $\delta^{13}C$ values in stalagmites are associated with…' refers to NEB samples or is a general statement. If general, please add impact of temperature and PCP (see Fohlmeister et al. 2020), and perhaps relocate this to the literature review.

14. Figure 3

As for other figures, please change the colour scheme.

Please make the lines in the legend thicker so that the colours are easier to see.

Please update the 99% confidence interval to a shaded band – the two cyan lines are hard to see (assuming there are 2? In some places it seems like the black line is outside of the bounds of the 99% confidence interval? E.g. see ~1100 CE).

The U-Th data should have a label (i.e. a) to be consistent with the other data presented here.

Can this figure be combine with Figure 4? There is a lot of overlap.

Are the older TRA7 $\delta^{13}C$ data needed – suggest removing them if they are not referred to in the paper.

15. Figure 4

As for Figure 3 re. colour palette, composite, and U-Th data.

Have you quantified the difference in $\delta^{13}$C between samples? From ~1500 CE onwards they don't appear to covary closely.

16. Line 417: can you please expand on why DV2 and the RN record differ? "The general trend towards more positive values" – please add over what time period this trend occurs, as I don't think it persists over the whole records.
17. Line 421: please change 4.2 ka BP, or whatever convention you choose and be consistent throughout.
18. Line 452: please explain why you think AMV and RN decoupled after ~0 CE.
19. Figure 5

I think you have accidentally plotted the Lapointe AMV backwards.

20. Line 503: please move Figure 6 up to about here.
21. Line 520: please capitalise 'Indigenous'
22. Line 521 – "Entire Indigenous tribes died of starvation as a consequence of this drought and a related smallpox epidemic" – this suggests the smallpox outbreak was caused by the drought – is that correct? Suggest rewording to "Entire Indigenous tribes died of starvation as a consequence of this drought and a concurrent smallpox epidemic"
23. Line 529: what is the age error at 1770 CE – adding the uncertainty might bolster your point that this event is the 1776-1778 drought
24. Line 535: as per above please add age uncertainty.
25. Line 544: suggest reword to "Although the TRA5 speleothem chronology precision is reduced during the last ~150 years..."
26. Figure 6: as for earlier figs, add a, b... label for U-Th data
27. Line 567: "these data suggest a trend toward increased aridity over NEB from 3000 BP to present..." Please be consistent with use of BP vs BCE. At line 495 you say the last 500 years were the wettest of the last 2 millenia, which contradicts the above statement.
28. Line 572: "drought period between 1500 and 1750" – Is this referring to the drought events in TRA5? The wording suggests it is linked to the RN composite, which shows abrupt change at~1500 CE to wetter conditions. Could you please clarify. Throughout, I suggest you make sure you are consistent with naming conventions between samples and between the composite record and the individual samples. Perhaps consider adding sub-headings to differentiate the longer composite record and the more recent drought record.
29. The data availability statement is missing.
30. Table S1and S2 – please use a different symbol to denote data from Cruz et al. as * is used elsewhere in the table
31. Alves 2003 – this link is broken and I could not find the article at the website.

**References**

Kaufman, D., McKay, N., Routson, C., Erb, M., Dätwyler, C., Sommer, P.S., Heiri, O., Davis, B., 2020. Holocene global mean surface temperature, a multi-method reconstruction approach. Sci Data 7, 201. https://doi.org/10.1038/s41597-020-0530-7

Treble, P.C., Baker, A., Abram, N.J., Hellstrom, J.C., Crawford, J., Gagan, M.K., Borsato, A., Griffiths, A.D., Bajo, P., Markowska, M., Priestley, S.C., Hankin, S., Paterson, D., 2022. Ubiquitous karst hydrological control on speleothem oxygen isotope variability in a global study. Commun Earth Environ 3, 1–10. https://doi.org/10.1038/s43247-022-00347-3

---

## Community Comment (CC1)

Comments on "Spatiotemporal ITCZ dynamics during the last three millennia in Northeastern Brazil and related impacts in modern human history."

Authors: Giselle Utida, Francisco William Cruz, Mathias Vuille, Angela Ampuero, Valdir F. Novello, Jelena Maksic, Gilvan Sampaio, Hai Cheng, Haiwei Zhang, Fabio Ramos Dias de Andrade, and R. Lawrence Edwards

This is an interesting study that uses speleothem $\delta^{18}O$ and $\delta^{13}C$ records to characterize the nuanced behavior of the ITCZ/tropical rain belt and its impact on the regional hydroclimate (i.e., precipitation variability) of Nordeste and eastern Amazona during the late Holocene. The main objective of this study is to improve the interpretation of late Holocene ITCZ dynamics in the South American tropics, which may help to better our understanding of past SASM variability. Additionally, their interpretation of RN $\delta^{18}O$ as a recorder of extreme dry events during the last 500 years has archeological and societal implications. This manuscript presents several thought-provoking and novel ideas pertaining to Atlantic and Pacific impacts on ITCZ-related precipitation during the late Holocene, which have the potential to reconcile paleoclimate records from Nordeste and Amazonia. Overall, this study also has the potential to be an excellent contribution to the field of South American paleoclimatology. However, I find that the manuscript (in its present state) has several major issues, which require further consideration, detail, and development before it should be accepted for publication. As such, I would recommend major revisions of the manuscript before final acceptance.

**Major issues:**

   1. **I am concerned that the AMV reconstruction presented in figure 5 (also referenced in the main text) is misleading. Specifically:**

Figure 5 (and lines 451–454): It is true that the presented AMV time series and the RN composite $\delta^{18}O$ time series look similar, but it is unclear what the authors are plotting. The green time series in figure 5 (shown below, top figure) does not look like the AMV reconstruction from Lapointe et al. (2020) (shown below, bottom figure)—raw data from https://www.ncei.noaa.gov/access/paleo-search/study/31353. The full range of values from the Lapointe dataset is 21.7–22.7, while the reconstruction shown in figure 5 only appears to be from 21.95 to 22.40.

[Figure]

Perhaps the authors plotted a different reconstruction of the AMV and used the wrong citation? Or perhaps it is the reconstruction from Lapointe et al. (2020) but downsampled (if so, the authors need to make this clear in the methods or supplementary information)?

2. **The authors do not sufficiently explain the mechanisms driving the anti-phased behavior observed between the RN composite and Paraíso Cave δ¹⁸O records. Specifically:**

Lines 436–440: It is unclear what is meant by "a zonal behavior of precipitation shifts in the ITCZ domain." Are the authors proposing that RN and Paraíso are in-phase from 250–1100 CE, anti-phased at ~1100 CE, back in-phase from 1100–1500 CE, and then anti-phased again from 1500–1750 CE? The authors should provide more explanation for this behavior.

Additionally, the authors state that "even though the Paraíso and Cariaco sites are located in different hemispheres, the observed in-phase climate relationship during the LIA suggests that their isotopic signatures were both sensitive to the same rainfall changes over northern South America." The Cariaco record is not an isotope-based record. Rather, it is a bulk titanium % record. The wording of this sentence should be changed accordingly.

Lines 446–451: Here, the authors discuss the AMV and ITCZ displacement during a warm AMV. However, the authors have not defined what a warm AMV is, albeit the reader could find out in the cited studies. I recommend the authors specifically define the AMV in detail, and make clear what is meant by a warm vs cold AMV.

Lines 461–463: The authors state, "Our analysis corroborates with this and points to increasing precipitation over N-NEB and decreasing precipitation over eastern Amazon, between 1500–1750 CE, when both AMV and PDV are in cold phase (Fig 4)." There is no reference to the PDV in figure 4, nor has the PDV been described/defined yet at this point in the text. No PDV reconstructions are provided in any of the figures, and the provided AMV reconstruction is in figure 5, not figure 4. Last millennium SST gradients from Steinman et al. (2022) are provided in figure 5, but they are not PDV or AMV reconstructions. I recommend either including a PDV reconstruction in one of the figures, or to remove this text from the manuscript.

Lines 463–465: The authors state, "This sign reversal is assigned to perturbations of the regional Walker cell's produced by teleconnection between the Atlantic and Pacific (Kayano et al., 2022, He et al., 2021)." I find this explanation to be vague, and recommend that the authors provide a clearer and more detailed explanation for the sign reversal. What does "perturbations of the regional Walker cell's" mean exactly? What teleconnections are the authors referring to, and what are the mechanisms driving the aforementioned perturbations?

3. **The conclusion and abstract both discuss ITCZ dynamics forced by the AMV and PDV, including position, intensity, and width. However, in the main text, the authors do not sufficiently explain which dynamical aspect of the ITCZ responds to different AMV/PDV phases, nor do they explain any mechanism(s) behind the AMV/PDV forcing. Specifically:**

Lines 570–577: In this paragraph, the authors suggest that during the last millennia, ITCZ dynamics cannot be explained solely by north-south ITCZ migrations or one single forcing mechanism. They propose a zonally non-uniform behavior of the ITCZ during times when the RN

record is anti-phased with the Paraíso cave record—forced by the interactions between the AMV and PDV modes that changed the regional Walker cell position and ITCZ intensity/width.

However, the authors never really attributed the anti-phased behavior between N-NEB and eastern Amazonia to the differential AMV/PDV phases. They discussed observed precipitation anomalies during overlapping periods of AMV and PDV phases in the modern, and suggested that it could be responsible for the observed anti-phased behavior. However, they never directly compared the speleothem time series with AMV and PDV reconstructions. Nor did the authors propose a detailed mechanism for how different AMV/PDV phases impact ITCZ width/intensity, despite changes in ITCZ width/intensity also being mentioned in the abstract (lines 46–50). In addition, the authors did not really describe when the ITCZ may have expanded/contracted or became weaker/stronger (aside from stating that this may have happened when the RN composite record and Paraíso are anti-phased). Ultimately, they never describe mechanism(s) for 1) how different AMV/PDV phases impact ITCZ dynamics, 2) how changes in ITCZ width/intensity may cause the observed anti-phased behavior, and 3) how the regional Walker cell position is forced by different AMV/PDV phases. I recommend that the authors provide more detail to this part of the Conclusions and Discussion sections overall, and propose/explain specific mechanisms that can reconcile the observed hydroclimate variability in N-NEB and eastern Amazonia.

Additional note: The authors should be extremely clear when generally discussing ITCZ width/intensity. What exactly do the authors mean by ITCZ width? Is it the width of the actual band of deep convection? Width of the seasonal range of the ITCZ? These terms should be explicitly defined early in the manuscript. Some papers that may be useful to reference include Donohoe et al. (2013), Atwood et al. (2020), Byrne and Schneider (2016), and Roberts et al. (2017).

**Additional comments and concerns:**
Lines 89–92: The authors cite Lechleitner et al. (2019), but I believe the correct citation is Lechleitner et al. (2017). Additionally, another relevant citation that may be relevant and could be included here is Asmerom et al. (2020) published in Science.

Lines 95–102: The authors call out the SASM and the ITCZ here as focus points of recent studies on tropical South American precipitation, but have not mentioned the South Atlantic Convergence Zone (SACZ). While not explicitly relevant to their findings, the SACZ should at least be mentioned here because of its important relationship with the SASM and ITCZ, and because it has been the topic of several recent paleoclimate and modern precipitation studies (Novello et al., 2018; Nielsen et al., 2019; Zilli et al., 2019; Wong et al., 2021).

Figure 1: It may help the reader to include annotations in the figure, including labeling the core SASM domain, ITCZ location, SACZ, etc. Additionally, while I understand the choice to include austral autumn precipitation climatology (when N-NEB receives most of its precipitation), it may be worthwhile to include panels with precipitation climatology for the austral winter and spring (either added to figure 1 or included in the supplement). This would allow for the reader to visually assess the spatiotemporal dynamics of the ITCZ, SASM, and SACZ, and how precipitation varies at sites 1–4 during the different seasons.

Lines 165–174: Figure S1 receives a lot of attention in this paragraph, and should probably be included as a main text figure. Alternatively, it could be incorporated into an existing main figure.

Figure 2: Readers who are green-red colorblind will not be able to see the small green dots (that denote the location of the GNIP stations) in any of the panels. I recommend changing the color to black and potentially increasing the size of the dots.

Lines 362–363: It gets confusing when the authors use both before present (BP) dates and before common era/common era (BCE/CE) dates. Additionally, ky has not been defined before this point, so the authors should spell it out before using the abbreviation.

Figure 3: Same red–green issue as mentioned in Figure 2.

Figure 4: It would be extremely helpful for the authors to include vertical bars when referencing specific time periods in the text. Such periods include the LIA, MCA, Bond 2 event, etc. Additionally, the authors reference trends resulting from insolation forcing in the paragraph starting at line 417. The authors should consider including a time series of solar insolation.

Also, the $\delta D$ record from Boqueirão Lake is relative to VSMOW, not VPDB (Utida et al., 2019). This appears to be a typo and should be changed accordingly.

Lines 389–392: The authors state that from 1060 to 480 BCE, there was increased precipitation in N-NEB as suggested by negative $\delta^{18}O$ anomalies. But it is unclear what the authors mean by 'increased precipitation'. During this time, there is multidecadal variability in the RN composite $\delta^{18}O$ record, but no clear/obvious trend between 1060 and 480 BCE. Perhaps the authors meant that there was increased precipitation *relative* to another part of the record. I would recommend clearing this up.

Lines 408–409: The authors reference the $\delta D$ record from Boqueirão Lake, and the same record is shown in figure 4. However, the authors describe the record as a "$\delta D$ lipids" record. Lipids are a broad group of molecules which include waxes, glycerides, terpenoids, tetrapyrrole pigments, etc. The authors should be more specific, and should reference the record as a leaf wax $\delta D$ record of *n*-$C_{28}$ alkanoic acids from Boqueirão Lake sediments (hereinafter referred to as $\delta D$ lipids).

Lines 495–497: The authors focus their discussion of extreme dry events recorded in the TRA5 $\delta^{18}O$ record between 1500 and 1850 CE. However, it is unclear why the authors do not discuss dry events/distinct $\delta^{18}O$ peaks after 1850 CE, despite their record extending into the 21st century. Is it because the TRA5 speleothem chronology is not as precise during this time?

Lines 518–523 and Figure 6: The authors reference several historical droughts that had severe societal/socioeconomic consequences. It may be helpful to annotate figure 6 to highlight the most severe droughts referenced in the text. The number/letter labeling in figure 6 makes it hard to discern the severity of the droughts by looking at the figure alone.

Lines 533: The authors should provide more detail here. Which Governor are the authors referring to? Governor of what/where?

Figure 6: Why focus on just TRA5? TRA7 and FN1 appear to cover the same period as TRA5. Is TRA5 the only speleothem that records the extreme drought events? Do TRA7 or FN1 record any of the same drought events? If they do not, why would only one speleothem record these drought events and not the others?

It may be helpful to include the age uncertainty in the right panel of the figure under the heading "TRA5". For example, 1546 ± XX. Especially because this figure focuses on only the last 500 years, it would allow the reader to critically compare the speleothem dates to the historical drought dates listed in the column labeled "Historical."

Additionally, I am curious if there is an available archeological record(s) or something similar that could be plotted with the TRA5 $\delta^{18}O$ record. Especially since the authors discuss the societal implications of the extreme droughts in relation to human population and welfare, it would be useful for the reader to visualize the impact through comparison with the speleothem record.

Line 565–567: The authors state, "The N-NEB record presents a trend toward drier conditions as is also being observed in the Diva de Maura Cave in S-NEB, interpreted as an ITCZ withdrawal and SASM weakening, respectively." It is unclear what the authors mean by "ITCZ withdrawal," especially since the authors highlighted the dynamical behavior of the ITCZ earlier in the paper. Is it a withdrawal via mean ITCZ displacement? Contraction or weakening of the ITCZ? More detail here would be helpful for the reader.

**References:**

Asmerom, Y., Baldini, J. U. L., Prufer, K. M., Polyak, V. J., Ridley, H. E., Aquino, V. V., Baldini, L. M., Breitenbach, S. F. M., Macpherson, C. G., and Kennett, D. J.: Intertropical convergence zone variability in the Neotropics during the Common Era, Sci. Adv., 6, eaax3644, https://doi.org/10.1126/sciadv.aax3644, 2020.

Atwood, A. R., Donohoe, A., Battisti, D. S., Liu, X., and Pausata, F. S. R.: Robust Longitudinally Variable Responses of the ITCZ to a Myriad of Climate Forcings, Geophysical Research Letters, 47, https://doi.org/10.1029/2020GL088833, 2020.

Byrne, M. P. and Schneider, T.: Energetic Constraints on the Width of the Intertropical Convergence Zone, Journal of Climate, 29, 4709–4721, https://doi.org/10.1175/JCLI-D-15-0767.1, 2016.

Donohoe, A., Marshall, J., Ferreira, D., and Mcgee, D.: The Relationship between ITCZ Location and Cross-Equatorial Atmospheric Heat Transport: From the Seasonal Cycle to the Last Glacial Maximum, Journal of Climate, 26, 3597–3618, https://doi.org/10.1175/JCLI-D-12-00467.1, 2013.

He, Z., Dai, A., and Vuille, M.: The joint impacts of Atlantic and Pacific multidecadal variability on South American precipitation and temperature, Journal of Climate, 1–55, https://doi.org/10.1175/JCLI-D-21-0081.1, 2021.

Kayano, M. T., Cerón, W. L., Andreoli, R. V., Souza, R. A. F., Avila-Diaz, A., Zuluaga, C. F., and Carvalho, L. M. V.: Does the El Niño-Southern Oscillation Affect the Combined Impact of the Atlantic Multidecadal Oscillation and Pacific Decadal Oscillation on the Precipitation and Surface Air Temperature Variability over South America?, Atmosphere, 13, 231, https://doi.org/10.3390/atmos13020231, 2022.

Lapointe, F., Bradley, R. S., Francus, P., Balascio, N. L., Abbott, M. B., Stoner, J. S., St-Onge, G., De Coninck, A., and Labarre, T.: Annually resolved Atlantic sea surface temperature variability over the past 2,900 y, Proc. Natl. Acad. Sci. U.S.A., 117, 27171–27178, https://doi.org/10.1073/pnas.2014166117, 2020.

Lechleitner, F. A., Breitenbach, S. F. M., Rehfeld, K., Ridley, H. E., Asmerom, Y., Prufer, K. M., Marwan, N., Goswami, B., Kennett, D. J., Aquino, V. V., Polyak, V., Haug, G. H., Eglinton, T. I., and Baldini, J. U. L.: Tropical rainfall over the last two millennia: evidence for a low-latitude hydrologic seesaw, Sci Rep, 7, 45809, https://doi.org/10.1038/srep45809, 2017.

Nielsen, D. M., Belém, A. L., Marton, E., and Cataldi, M.: Dynamics-based regression models for the South Atlantic Convergence Zone, Clim Dyn, 52, 5527–5553, https://doi.org/10.1007/s00382-018-4460-4, 2019.

Novello, V. F., Cruz, F. W., Moquet, J. S., Vuille, M., De Paula, M. S., Nunes, D., Edwards, R. L., Cheng, H., Karmann, I., Utida, G., Stríkis, N. M., and Campos, J. L. P. S.: Two Millennia of South Atlantic Convergence Zone Variability Reconstructed From Isotopic Proxies, Geophys. Res. Lett., 45, 5045–5051, https://doi.org/10.1029/2017GL076838, 2018.

Roberts, W. H. G., Valdes, P. J., and Singarayer, J. S.: Can energy fluxes be used to interpret glacial/interglacial precipitation changes in the tropics?, Geophys. Res. Lett., 44, 6373–6382, https://doi.org/10.1002/2017GL073103, 2017.

Steinman, B. A., Stansell, N. D., Mann, M. E., Cooke, C. A., Abbott, M. B., Vuille, M., Bird, B. W., Lachniet, M. S., and Fernandez, A.: Interhemispheric antiphasing of neotropical precipitation during the past millennium, Proceedings of the National Academy of Sciences, 119, e2120015119, 2022.

Utida, G., Cruz, F. W., Etourneau, J., Bouloubassi, I., Schefuß, E., Vuille, M., Novello, V. F., Prado, L. F., Sifeddine, A., Klein, V., Zular, A., Viana, J. C. C., and Turcq, B.: Tropical South Atlantic influence on Northeastern Brazil precipitation and ITCZ displacement during the past 2300 years, Sci Rep, 9, 1698, https://doi.org/10.1038/s41598-018-38003-6, 2019.

Wong, M. L., Wang, X., Latrubesse, E. M., He, S., and Bayer, M.: Variations in the South Atlantic Convergence Zone over the mid-to-late Holocene inferred from speleothem δ18O in central Brazil, Quaternary Science Reviews, 270, 107178, https://doi.org/10.1016/j.quascirev.2021.107178, 2021.

Zilli, M. T., Carvalho, L. M. V., and Lintner, B. R.: The poleward shift of South Atlantic Convergence Zone in recent decades, Clim Dyn, 52, 2545–2563, https://doi.org/10.1007/s00382-018-4277-1, 2019.

---

## Author Comment (AC2)

*We are grateful for the reviewer's comments on manuscript cp-2023-2. We addressed the reviewer's comments below in italicized text.*

RC2: 'Comment on cp-2023-2', Anonymous Referee #2, 21 Mar 2023

**Review „Spatiotemporal ITCZ dynamics during the last three millennia in Northeastern Brazil and related impacts in modern human history" by Utida et al.**

I have read with great interest the discussion paper by Utida et al. The authors analyze spatiotemporal ITCZ dynamics during the last three millennia in NE Brazil (NEB), and claim to relate their inferences to modern human history. The study presents partially replicated speleothem proxy records from two caves in NE Brazil, and provide an overview of past (hydro)climate trends and variability in the greater region of NEB of the southern margin of the ITCZ.

This new data set is sound and I have no doubt about the quality of the applied methods and presented data. In principle, the scientific significance is valid, since this dataset complements the northern South American speleothem record in high resolution. However, I have some concerns about the structure and clarity of the manuscript, which I feel needs some improvements before final publication.

**Main comments:**

- The structure:

I find most of the conclusions concerning ITCZ dynamics intriguing and interesting. However, I found the manuscript sometimes hard to read and some parts of the discussion are not easy to follow. For example, the section in L388ff has a rather unclear structure. The first half seems to be organized chronologically along the time of record, and describes the observed trends. But before this discussion is finished, the discussion jumps to comparing relationships between proxies, and refers to sections of the record which have not been described yet. Later on, the discussion also jumps from describing potential processes back to certain events and forth to other aspects again. I feel like the whole discussion should be carefully restructured and streamlined to build the arguments better on each other, and to provide the reader a common thread throughout the manuscript to prepare and justify your conclusions properly. I suggest to choose a consistent, logic structure, such as building up the discussion more strictly chronologically along your record, and also discuss trends first and events later separately?! Another possibility would be to bring the proxy interpretation first, and then compare to other records and discuss the forcings and consequences… There are several possibilities, but please do not mix it all up…

*Thank you for drawing attention to the structure of the manuscript. We will restructure the manuscript according to all reviewer suggestions. Most of the above comments related to structure are addressed in the comments below and we will do our best to improve the general structure of the manuscript after a final revision including all suggestions. As far as the paragraph in L38ff is concerned: we rewrote it and it is presented as part of the comments below.*

I also strongly suggest to put special effort in elaborating how the two parts of the discussion (i.e., the paleo-record description, and the discussion of historical droughts) actually build on each other, and better justify why both aspects need to be discussed in one paper. In the current version of the manuscript these appear more as two separated stories.

*Thank you for drawing attention to the connection between these paleo-records and historical records. The paleo-precipitation record from Northeast Brazil is important to understand the modern climate and to put it in a long-term historical context. Hence, there is really no separation between the two, just a continuing precipitation history over time, indeed. No observed or reconstructed precipitation record exists for the period prior to and 1850 CE in NE Brazil. The only available information is the historical record of droughts. Hence the speleothem record allows us to put these droughts in a longer-term context and provide a broader spatiotemporal assessment. As far as the possibility to discuss the historical droughts in*

*another paper is concerned, we believe that we still lack sufficient data for a second paper. The current analysis should really be viewed as a first attempt to compare paleo-precipitation and historical records in this region.*

- U/Th Results description

I miss a proper description of the U/Th results in the main text. This should e.g., comprise U and Th concentrations, uncertainties, Th contamination, description of inversions, etc, … (check Dutton et al. 2017 as a guideline to report U-series data). This is also important due to the presence of both calcite and aragonite, where we would expect an influence on the ages if recrystallization occurred! In addition, a statement concerning the final uncertainties of the age-depth model is essential, also regarding the several outliers. This is particularly relevant when reporting absolute ages for extreme events! From the so presented age models, it is not at all clear if the dating supports an annual precision of a single drought event, or the unequivocal allocation to an event reported in the historic record.

*The description of the methods and U/Th results has been revised and will be included in the manuscript according to the text below. The methods were revised to be in accordance with Dutton et al. (2017) suggestions for U/Th series publications.*

*Section 3: Materials and Methods*

*Chronological studies on speleothems were based on U-Th geochronology performed at the Laboratories of the Department of Earth and Environmental Sciences, College of Science and Engineering, University of Minnesota (USA), and at the Isotope Laboratory of the Institute of Global Environmental Change, Xi'an Jiaotong University (China), according to Cheng et al. (2013). Subsamples of ~100 mg were obtained in clear layers, close to the growth axis trying to keep a maximum thickness of 1.5 mm, 10 mm wide and no more than 3 mm depth. The powder samples were dissolved in 14 N $HNO_3$ and spiked with a mixed solution of known $^{233}U$ (0.78646 ± 0.0002 pmol/g) and $^{229}Th$ (0.21686 ± 0.0001 pmol/g) concentration. Th and U were co-precipitated with FeCl and separated with Spectra/Gel® Ion Exchange 1x8 resin column with 6N HCl and super clear water, respectively. Th and U were counted in an inductively coupled plasma-mass spectrometer (MC-ICP-MS Thermo-Finnigan NEPTUNE PLUS) and the results were calculated in a standard spreadsheet based on Edwards et al. (1987) and Richards and Dorale (2003) using the isotopic ratios measured, machine parameters and corrections factors to eliminate effects of contamination by detrital Th to finally obtain the age of each sample. The decay constants used are: $\lambda_{238}$ 1.55125 x $10^{-10}$ (Jaffey et al., 1971), $\lambda_{234}$ 2.82206 x $10^{-6}$ and $\lambda_{230}$ = 9.1705 x $10^{-6}$ (Cheng et al., 2013). Corrected $^{230}Th$ ages assume the initial $^{230}Th/^{232}Th$ atomic ratio of 4.4 ± 2.2 x $10^{-6}$. Those are the values for a material at secular equilibrium, with the bulk earth $^{232}Th/^{238}U$ value of 3.8 (McDonough and Sun, 1995). The ages are reported in BP (Before Present defined as the year 1950 A.D.) and converted to Common Years (CE). Age uncertainties are 2 σ.*

*Results and discussions*

*The results and discussions below regarding $^{232}Th$ contamination and calcite x aragonite crystallization will be included in the appropriate section of the manuscript, according to the reviewer's suggestion.*

*The high values of $^{232}Th$ and low $^{230}Th/^{232}Th$ ratio suggest incorporation of detrital Th transported by the seepage solution to the speleothems, which lead to a higher uncertainty of the age values. Recrystallization of aragonite into calcite might also reduce the U content and given older age for carbonates (Lachniet et al., 2012). These are the main reasons for age inversions along speleothems from Northeast Brazil. Therefore, we analyzed a large number of U/Th ages to improve the age model and reduce the errors associated with detrital Th and recrystallization.*

*FN1 is partially composed of calcite between the depths of 83 and 128 mm (Table S3), and top and base are composed of aragonite. Overall this stalagmite presents low U concentration and high $^{232}Th$ amounts. We considered the association of low $^{230}Th/^{232}Th$ and low U content the most important factor*

*affecting the age errors and inversions in the FN1 stalagmite. In contrast the FN2 stalagmite has a more precise chronology due to the predominant aragonite composition, with high $^{238}U$ content and higher $^{230}Th/^{232}Th$ ratio than FN1. The ages from the FN1 stalagmite are all in chronological order and contain low errors and were therefore all kept in the age model.*

*The TRA5 stalagmite is entirely composed of calcite, but the $^{238}U$ content is relatively high compared to other stalagmites, which improves the confidence in its age results. However, the high $^{232}Th$ content of samples from the top of TRA5 affects the age results over the last 200 years. The other two inversions in TRA5 (71 and 104 mm, Table S2) might also be a result of $^{232}Th$ contamination resulting in increased errors.*

*Most of the TRA7 stalagmite used in our composite is composed of calcite (from top to 130 BCE). According to age results produced by Utida et al. (2020), most of the ages are in chronological order and the inversions seem to not have a direct relationship with $^{238}U$, and the high $^{232}Th$ content is similar to other ages from TRA7.*

*The age uncertainties caused by high $^{232}Th$ concentration and calcite recrystallization in stalagmites might affect the age model. However the strong coherence between the $\delta^{18}O$ curves from different stalagmites argues in favor of the good quality of our chronology. This is evident when FN2, which is composed 100% of aragonite, is compared with other samples. There is a different amplitude range in its $\delta^{18}O$ values, but when the curve is superposed on other $\delta^{18}O$ records the variability is similar. This amplitude range is corrected when the $\delta^{18}O$ results are submitted to the ISCAM composite construction, since it normalizes the results.*

*Historical records and age model uncertainties*

*The errors of our age model for TRA5 are around ± 30 years (95% confidence interval) and we are thus aware that this uncertainty complicates the attribution to a single three-year long event. There exist no precipitation reconstructions or observations from this region between 1500 and 1850 CE, aside from these historical drought records. We thus consider our speleothem-based record as a first attempt to reconstruct precipitation in Northeast Brazil that would allow a comparison with historical droughts. If our speleothem records regional hydroclimate, it should retain a signal of the most intense droughts over NEB that are known to have struck the region based on the available historical literature of Brazil. The historical droughts we discuss in the paper, and we identify in our record, are the longest drought events in Northeast Brazil that occurred within the zone of influence of the ITCZ, and are thus probably the most likely to be recorded by stalagmites. Note that despite dating uncertainties of our record, the $\delta^{18}O$ peak of each drought event recorded, is consistent with the historical record of Lima and Magalhães (2018). Furthermore, the period between 1620 and 1717 CE is devoid of any abrupt drought events in the TRA5 stalagmite, which is again consistent with the historical records. Lima and Magalhães (2018) registered only 3 short drought events within this period of almost 100 years. It is also important to mention that Lima and Magalhães (2018) report all drought events in NEB and do not indicate their location. As discussed above northern and southern NEB are influenced by different climatic systems, the ITCZ and SASM, respectively, and this can explain, in part, the differences between historical and stalagmite records of Rio Grande do Norte.*

I have some more general comments to the style of the writing and presentation, which I summarize here. Please find specific locations related to the following points in my minor comments along the text:

- Across the manuscript I found repetitive statements, but also rather irrelevant information. This makes the reader lose focus, so I suggest to try to shorten/streamline the text in general.

- In many figures, some aspects are hardly visible. Please improve accessibility, e.g., text sizes, increase size of markers of locations, use colors that are better visible.

- Sometimes past and present tense is mixed, please check language style.

*We are in the process of performing a complete revision of the text in order to improve the language quality and the conciseness of the text. The figures are updated and can be seen below. They were updated*

*for text and markers sizes, as well as adapted for color blind readers. Certainly, the points mentioned will help us produce a higher quality manuscript.*

**Minor comments:**

L49 weakening

> *The word spelling will be corrected.*

L62-63: Is there a reference for this statement?

> *The references are the same as those mentioned after this statement. We will add the references Marengo and Bernasconi (2015) and Lima and Magalhães (2018) to this sentence.*

L91: I think the Lechleitner Paper is from 2017.

> *The year of publication will be corrected to 2017.*

L129-131 is this relevant?

> *We believe so. But we combined the two sentences into one, and we clarified the meaning of the text.*
>
> *"The caves were developed in the Cretaceous carbonate rocks of the Jandaíra Formation, Potiguar Basin, close to the Apodi River valley in a region of exposed karst pavements (Pessoa-Neto, 2003; Melo et al., 2016; Silva et al., 2017)."*

L138: Any idea why the cave temperature is considerably lower than the annual mean temperature? Is this relevant for your data?

> *The annual mean temperature was taken from a climate station in the city, kilometers from the cave. Temperatures in cities tend to be higher than in pristine environments (urban heat-island effect) such as those where the caves are located. This information is not directly relevant for the interpretation of our results, but nonetheless helpful for those who want to better understand the climatology of the region.*

L148: I feel like most of this section is rather results than material/methods description?

> *We agree with the reviewer that this part of the text is better suited in the results than the methods or Regional settings sections. We will adjust this section accordingly.*

L149ff: There is a lot of discussion of the different sectors within NEB, it may be helpful for the discussion and the readers to indicate those in a figure?

> *The spatial correlations of Figure 2 are used to define the northern and southern NEB climatologically. The new version of this Figure includes the labels "Northern NEB" and "Southern NEB" in graphs a) and b). Please, see the revised version of Figure 2.*

L164: How is "most significant" defined?

> *The "most significant" years of El Niño in NEB are those that most drastically impacted the precipitation amount. We changed the text to clarify the statement. Please see the revised text below.*

*"…we excluded the 39 El Niño - Southern Oscillation (ENSO) years that most drastically changed the precipitation amount in NEB, following the methodology of Araújo et al. (2013)."*

L174: "is primarily the result of a shorter rainy season". This is not quite what is described above. There you write that the rainy season has the same length but is weaker?

L175: "The anomalous length…" See previous comment, according to your own results, this is only for the wetter years.

*We rewrote the paragraph mentioned in comments L174 and L175 to clarify these aspects. Please see below.*

*"The results (Fig. S1) reveal that in the majority of years (normal years - interquartile range) the rainy season persists from February to April, with precipitation varying from 100 to 180 mm/month, and minor contributions occurring in January and May (50-70 mm/month). During the drier years (lower quartile), February has a reduced precipitation amount, similar to the amount in January during normal years, as described above. The maximum precipitation of 90 mm/month occurs between March and April. For wetter years (upper quartile), the rainy season starts in January with more than 100 mm/month and lasts until May with almost 150 mm/month, reaching values higher than 250 mm around March. These data show that wetter years are characterized by increased precipitation amounts and a longer rainy season starting in January and ending in May, while the precipitation deficit during drought years is a result of decreased precipitation amount and a shorter rainy season, with a peak in precipitation between March and April. The anomalous length of the rainy season during dry and wet years is attributed to variations in the meridional SST gradient in the tropical Atlantic that results in a shift of the ITCZ to the north or south of its climatological position (e.g., Andreoli et al., 2011; Marengo and Bernasconi, 2015; Alvalá et al., 2019)."*

L189ff: If this is relevant for the discussion later, I feel like the authors should clearly define the difference between ITCZ related rainfall in NEB, and SASM related rainfall in S-NEB. Some reader may not be able to recall the exact difference at once…

*As mentioned in comment L149ff, the spatial correlations of Figure 2 are used to define the northern and southern NEB climatologically. The new version of this figure indicates the "Northern NEB" and "Southern NEB" in order to call attention to the differences. Please see the new version of Figure 2.*

L229-230: This information is not relevant for this study.

*The information will be removed.*

L282 to significantly reduce

*The error will be corrected.*

L327: Avoid repetition of the method description.

*We will remove the sentence of lines 327-329.*

L343: Why is a composite record only constructed for $\delta^{18}$O? what about the early phase 2-3k BCE? It is shown in the figure, but I didn't find a statement why it is shown but not discussed

*$\delta^{18}$O is a proxy for regionally integrated climate and circulation upstream, with the main signal related to atmospheric processes, while $\delta^{13}$C values are more diverse and more site-specific, related to the seepage solution pathway and spots of vegetation above the cave. Therefore, a composite for the $\delta^{13}$C will give a more heterogenic mosaic that may not be related with the regional conditions. For the early part of*

*our record, we therefore decided to remove it from the main text. As it is not being discussed, as suggested by the first reviewer, we will move Figure 3, with this early part of our record, to the Supplemental Material as Figure S5.*

L389: if the age axis is correct, the oldest period of the RN composite is around 1.5 to 1k BCE?

L390: I see rather positive anomalies between 1 and 0.5 BCE…

L391ff: Confusing section, please clarify. In the previous sentence you state, there is soil erosion, here you state that did not contribute much… Now what?

L397: more negative as compared to what? to me, the $\delta^{18}O$ values are rather higher than in the earlier part of the record… please clarify.

L399ff: This paragraph is hard to follow. Please don't start compare/discussing sections of the record that haven't been mentioned before… (here the LIA suddenly pops up)

> *Thank you for calling attention to this paragraph. We combine our reply to comments L389 - L397 below. We rewrote the paragraph as shown below, paying attention to all suggestions made by the reviewer.*

> *"The oldest period covered by the RN Composite, from 1200 to 500 BCE, is characterized by successive dry and wet multidecadal periods, with increased precipitation in N-NEB from 1060 to 750 BCE and from 460 to 290 BCE, as suggested by the negative departures seen in the $\delta^{18}O$ values. During this last period, there is also a tendency from lower to higher $\delta^{13}C$ values, suggesting progressive surface soil erosion related to rainfall variability (Fig. 4), as interpreted by Utida et al. (2020). This period ends up in a stable interval from 300 BCE to 0 CE with $\delta^{13}C$ values close to the bedrock signature at about -1‰ to +1‰, indicating a lack of soil above the cave. After an abrupt reduction of $\delta^{13}C$, the values decrease to approximately -2‰ between 200 CE and 1500 CE. From 1500 CE to the present, negative values of $\delta^{13}C$ is responding to wet climatic conditions as indicated by lower $\delta^{18}O$ values. The more negative $\delta^{13}C$ during this period can be related to denser vegetation that favored both soil production and stability above the cave."*

L408: On the millennial scale, yes… since you also mention shorter timescales earlier, I would clarify this here…

> *We adapted the text to clarify this statement. Please see below.*

> *"During the last 2500 years, the RN Composite shows similar characteristics as the lower-resolution δD lipids record obtained in Boqueirão Lake sediments"*

L413: Unclear, why is this?

L414 do you mean "latter"?

L415: Very vague statement, please specify. Also, how well are the lake sediments dated and is that comparable to your chronology?

> *The lake sediments were dated with the 14C method, which has larger errors than the U/Th method used for stalagmites. Furthermore, the age model of Boqueirão Lake was constructed with fewer ages compared to stalagmite chronology. We rewrote the sentence to simplify it and answer the comments from L413 to L415. Please, see below.*

> *"This inconsistency might be related to different chronological controls between lake and stalagmite records and possibly also by the location of Boqueirão Lake that is more strongly affected by the ITCZ and as it is located in the eastern coastal sector of NEB (Zular et al., 2018; Utida et al., 2019)."*

L420 Maybe indicate the insolation curve in the Figure?

*We have included the insolation curve in Figure 3 and 4 of the manuscript. This Figure in question already contains a lot of information and adding even more would make it difficult to read. We will instead add the insolation curve to the Figure S5 in the Supplement as shown below.*

[Figure]

*Figure S5 – Rio Grande do Norte stalagmite isotope record. (a) U/Th ages for RN stalagmites. (b) Raw data of $\delta^{13}C$. (c) Oxygen isotope results corrected for calcite-aragonite fractionation ($\delta^{18}O_{C-A}$), according to weight proportion of mineralogical results. (d) $\delta^{18}O$ RN Composite constructed using stalagmite records from NEB (black line). Grey lines denote the age model confidence interval of 99%. (e) February insolation curve at 10°S.*

L421 persistently (?)

*The word will be corrected.*

L426ff: This is an interesting conclusion which is however barely discussed beforehand. The discussion here rather ends quite abruptly. I feel this could be more elaborated, because it seems to relate to the statement in the abstract, that you can make inferences on spatio-temporal ITCZ variability?

*We rewrote the whole paragraph to call attention to the differences between N- and S-NEB and reinforce our conclusions. Please, check it below.*

*"It is important to note that the RN record exhibits a climatic signal that is distinctly different from the DV2 speleothem record from Diva de Maura Cave in S-NEB (Novello et al., 2012). The general trend toward more positive values, as a result of insolation forcing, occurs from 150 to 1500 CE in the RN Composite, but from 600 to 1900 CE in the DV2 sample (Cruz et al., 2009; Novello et al., 2012). This trend is a result of the persistent dry conditions in the entire NEB region following the 4.2 ky BP event. However, the DV2 record does not document the same multidecadal and centennial-scale climate variability as recorded*

*in the RN speleothem record, nor the less dry interval from 600 to 1060 CE seen in the RN Composite (Fig 3). As demonstrated by the spatial correlation maps between $\delta^{18}O$ values and regional precipitation (Fig. 2), the S-NEB and N-NEB regions are influenced by distinct rainfall regimes whose peaks of precipitation arise during the summer monsoon season and the autumn ITCZ, respectively. Our data provide evidence for a spatial and temporal distinction of NEB climate patterns in the past that can be interpreted as differences in seasonality during the last millennia. Furthermore, contemporaneous dry or wet events in both N-NEB and S-NEB suggest the occurrence of larger regional climate changes with higher environmental impacts."*

L432: very vague and unclear which characteristics are meant

L429ff (the whole section) difficult to follow here, you jump from describing a trend to single events, and then to processes again - not clear where this leads to? please provide the reader with some kind of guidance in between, maybe in form of a summary and/or statement which observation will be tested/explained now…

L437: unclear what information your record adds to this aspect, and how this relates to the discussion?

*We rewrote the paragraph to adjust it according to the reviewer's suggestions. Please see revised version below.*

*"When comparing N-NEB and eastern Amazon conditions, it is evident that the RN Composite shares some similarities with the Paraíso stalagmite record (Wang et al., 2017), due to the contribution of ITCZ precipitation in both places. But there are also important differences (Fig 4). The RN Composite shows lower δ18O values between 500 and 1000 CE, compared to the earlier period, while Paraiso shows decreasing values around the same period, suggesting a slight increase in precipitation in both areas. From 1160 to 1500 CE, abrupt increases in $\delta^{18}O$ values are seen in both records, which indicates abrupt and prolonged drought conditions due to a northward ITCZ migration. However, around 1100 CE, and the period from 1500 to 1750 CE, Paraiso is antiphased with the RN Composite and in phase with the Cariaco Basin (Haug et al., 2001), which is inconsistent with the notion of an ITCZ-induced regional precipitation change. Instead, a zonally-oriented precipitation change within the ITCZ domain over Brazil is required to explain the anti-phased behavior between precipitation in N-NEB and the eastern Amazon, and similarities between Cariaco and the eastern Amazon."*

L441: now the discussion jumps again back in time to another event… I would bring this example later to showcase a potential relationship to Atlantic temperatures…?

*We will adjust the entire manuscript according to suggestions and comments of the reviewers. Certainly the discussion about Bond events can be combined with the Atlantic temperature discussion and its relationship with the ITCZ. It will become clear where the best position for this paragraph is once we finalize the revision of the manuscript.*

L447: I suggest to turn the argument around - the idea is that ITCZ displacements are forced by temperatures, so we check if there is a relationship of our record to AMV?

L448ff This sentence seems incomplete

*We rewrote this paragraph in order to clarify our ITCZ displacement hypothesis related to meridional temperature gradients in the Atlantic. Please see the revised text below.*

*"We investigate the potential relationship between $\delta^{18}O$ values in our RN speleothems and an ITCZ displacement toward the warmer hemisphere to explain paleoclimate variability observed in N-NEB. In order to test this hypothesis, the RN Composite was compared with a reconstruction of Atlantic Multidecadal Variability (AMV) (Lapointe et al., 2020) (Fig. 4). Some studies suggest that the warm phase of the AMV forces the mean ITCZ to shift to the north of its climatological position, thereby causing a reduction in NEB*

*rainfall (Knight et al., 2006, Levine et al., 2018), while a recent study suggests that the warm phase of the AMV would cause a weakening of the ITCZ from February to July (Maksic et al., 2022)."*

L452: Have you checked other records of AMV / Atlantic SSTs that allow to check if the Lapointe record is representative for the entire basin during these times or not?

*Lapointe et al. (2020) present a record that is in good agreement with other temperature records from the North Atlantic and with other AMV reconstructions (Mann et al., 2009; Cunningham et al., 2013; Miettinen et al., 2015; Reynolds et al., 2016; Wang, et al., 2017; Spooner et al., 2020) and also with records from the Cariaco Basin (Black et al., 1999), which suggest that their AMV reconstruction is reliable and indicative of a large-scale teleconnection with the tropics.*

L462ff As far as I understand the plot, there is no PDV record in the plot, so how do you infer a cold phase of PDV during that time? I guess you refer rater to Fig 5, but still I suggest to explain which record / curve you are referring to here exactly and what they are showing? Is the Pacific SST gradient a measure of PDV? this curve shows centennial scale variability, but not at all decadal?

*The Figure presents only the AMV. The discussion about the relationship between AMV and PDV was only based on an observed precipitation analysis. We made some adjustments in the paragraph to clarify this aspect. Please see the revised text below.*

*"According to Kayano et al. (2020, 2022), during the last century, dry conditions over N-NEB and the eastern Amazon are present when AMV and Pacific Decadal Variability (PDV) are in both in their warm phase, or when the AMV is in a cold phase and the PDV in its warm phase. On the other hand, when AMV and PDV are both in their cold phase, precipitation over the Amazon is anti-phased with NEB, resulting in decreased precipitation over the Amazon and increased precipitation over NEB. This zonally aligned precipitation signal over eastern tropical S. America is the result of joint perturbations of both the regional Walker and Hadley Cell's, produced by teleconnection between the Atlantic and Pacific (Kayano et al., 2022, He et al., 2021). These conditions can explain in part our results, however during the decoupling of our record with AMV (between 1500 and 1750 CE), increasing precipitation over N-NEB and decreasing precipitation over the eastern Amazon can be better explained by the positive gradients both in Atlantic and Pacific Oceans forcing a south ITCZ migration (Fig. 4)."*

L494ff: what does the $\delta^{13}$C record tell in this time? extreme events could be also visible there, the record looks quite "spikey"

*The $\delta^{13}$C and $\delta^{18}$O records of TRA5 show similar characteristics during this time. These data can be interpreted in the same way as the rest of our record, indicating increased (decreased) precipitation ($\delta^{18}$O) and soil production (erosion), combined with a decrease (increase) in Prior Calcite Precipitation (PCP) at the epikarst ($\delta^{13}$C). All mentioned processes drive oxygen and carbon isotope variability in the same direction (Novello et al., 2021).*

L495: How do the other speleothem records compare during this time? I understand that they have lower resolutions, but to support your point and strengthen your arguments (e.g. concerning age model uncertainties, etc) a zoom into the comparison of the different proxy records might be helpful? Also, how would ISCAM move the TRA7 record with respect to the original agemodel? This gives also a hint regarding dating uncertainty...

*Unfortunately, the TRA7 record is not suitable for this kind of comparison because of its lower resolution. The deposition rates (DR) of TRA7 and TRA5 are different, 0.18mm/yr and 0.33 mm/yr. We tried to sample TRA7 at the maximum possible resolution to achieve such a comparison. Considering the uncertainties of the age model, some peaks are too smoothed and the $\delta^{18}$O data are not suitable for comparison. We believe that including such a comparison would not aid in our interpretation. The TRA7*

*ISCAM age model does not significantly change compared to the COPRA model, since both use the linear method. We did not plot them together in the Figure S4 because the superposition would not be visible. We show here a figure to demonstrate the similarity of both TRA7 age models and to clarify this question.*

[Figure]

L497ff: statements like this require a proper report of dating and age model uncertainties. From visual inspection there are some ages which have quite high uncertainties, which could limit the fidelity of such a record to absolutely date extreme events with annual precision! It could be also short-term hiatuses, that last longer than a single year…? I understand that the TRA7 age model is part of another paper, but then please still give a statement here, because this is relevant for your conclusions. It is also not clearly visible from the plot of the age model in the supplement.

*We will improve the methods, results and discussion sections when referring to the stalagmite ages. We also improved the figures in the supplemental material to better describe the age models and uncertainties of our record and we modified the text in this paragraph. Please see revised text below.*

*"In NEB, the low water availability has been one of the major challenges faced by its people during the last centuries (Marengo and Bernasconi, 2015; Marengo et al., 2021; Lima and Magalhães, 2018). On the other hand, the last 500 years were the wettest of the last two millennia, according to our results (Fig. 3). Superimposed on these long-term negative $\delta^{18}O$ anomalies, distinct peaks are recorded in the TRA5 $\delta^{18}O$ record from 1500 to 1850 CE (Fig 5). These drought events are visible in this record thanks to its higher deposition rate (faster growth) and thus higher temporal resolution of the $\delta^{18}O$ record when compared to other stalagmites used in our study. Although the age model errors of TRA5 are larger and could limit our ability to attribute $\delta^{18}O$ peaks to specific single-year events, it still allows for a comparison between these abrupt events with historical records to demonstrate the long-term context of abrupt drought events in modern human history."*

L538ff: how many droughts are not recorded in your stalagmite record? the reference is not accessible, so please provide a clear statement, or, better, a plot/histogram of all droughts reported by the other study in Fig 6

*The historical record of Lima and Magalhães (2018) (Graph 1 in the original paper) mentions drought events compiled from different historical letters and books from all of Northeast Brazil (NEB). Hence, some of these events might be located in the southern and/or northern part of Northeast Brazil. Our data record a smaller number of events than are listed in the historical data, probably recording primarily the most intense events that affected all of NEB, or ITCZ changes that affected only the northern portion of NEB. According to our correlation maps, southern and northern NEB have different precipitation sources and seasonality. Therefore, the TRA5 stalagmites do not record all events mentioned in the compilation of Lima and Magalhães (2018).*

*The paper can be accessed using the link below. The link will be updated in the reference section of the revised manuscript.*

*https://seer.cgee.org.br/parcerias_estrategicas/article/view/896/814*

[Figure]

*Graph 1 – Historical drought events in Northeast Brazil. Extracted from Lima and Magalhães (2018).*

L551: Discussion ends quite abruptly, following from your section 5.1 one would at least expect a hypothesis of a forcing mechanism of the drought occurrence?

*We will include a concluding paragraph at the end of section 5.1 suggesting a hypothesis related to our main conclusion. Please see the revised text below.*

*"We suggest that progressive changes in the mean ITCZ position over the course of the last 500 years might be responsible for historical droughts that affected the seasonality of N-NEB and caused abrupt and strong drought events. No preferred periodicity of these events is apparent in our record. Additional drought-sensitive high-resolution records will be required to improve our understanding of these historical droughts events in NEB."*

How is the drought frequency related to what you found out from your record of the past 2.5ka? I suggest to elaborate this a little bit more…

*Our stalagmite and RN Composite records contain variability at multidecadal and interdecadal frequencies. However, the wavelet analysis did not show a temporally continuous signal at a preferred wavelength. We therefore chose not to discuss this aspect in greater detail.*

**Figures**

Figure 1: Locations hardly visible, please increase the size of the text and the stars. Also No. 5 is barely visible, please choose other colors.

*We updated all figures to improve the font size of text and symbols. Please see the revised Figures 1 and 5 below. Figure 5 was updated and changed to Figure 4.*

[Figure]

*Figure 1 – Location and precipitation climatology of study sites during the austral summer (DJF - December to February) and autumn (MAM - March to May). Color shading indicates percentage of the annual precipitation total that is received during either DJF or MAM and highlights the extent of (a) the SASM over the continent and (b) the ITCZ over the ocean. Precipitation data is from the Global Precipitation Measurement (GPM) mission, with averages calculated over the period 2001–2020. 1) Trapiá and Furna Nova Cave (this study), 2) Boqueirão Lake (Utida et al., 2019), 3) Diva de Maura Cave (Novello et al., 2012), 4) Paraíso Cave (Wang et al., 2017), 5) Cariaco Basin (Haug et al., 2001). GNIP stations: A) Fortaleza, B) Brasília, C) Manaus.*

[Figure]

*Figure 4 - δ18O RN Composite compared with (a) Atlantic Multidecadal Variability (Lapointe et al., 2020) and (b) Pacific and Atlantic Sea Surface Temperature gradients calculated (z-score) according to Steinman et al. (2022). Atlantic: 2σ range of 1,000 realizations of the Atlantic meridional SST gradient (north – south). Pacific: median of 1,000 realizations of the Pacific zonal SST gradient (west – east).*

**Figure 2:** Increase symbols for locations. Please improve visibility in general. Caption should be streamlined, "precipitation amount" is mentioned twice in the first sentence (L199-201). Correlation maps is repeated in L199 and L204. GNIP is repeated in L200 and L207. No need to repeat all information to all caves again, it is also ok to refer to the previous figure...

*We updated all figures for size and to render them suitable for color-blind readers. Please see the revised Figure 2 and caption below.*

[Figure]

*Figure 2 – Monthly mean observed precipitation amount collected at ANA and $\delta^{18}O$ values for GNIP stations (IAEA-WMO, 2021) (black dots) and correlation maps between gridded precipitation and $\delta^{18}O$ anomalies from the same stations (black dots) for: (a) Northern NEB, Fortaleza and Pedra das Abelhas stations (star 1), (b) Southern NEB, Brasília and Andaraí stations (star 3), c) Eastern Amazon, Manaus and Belterra stations (star 4). The maps show the spatial correlation between $\delta^{18}O$ anomalies at GNIP stations and GPCC gridded precipitation anomalies based on the period 1961-1990 for December to February (DJF) and March to May (MAM) for Fortaleza, Brasília and Manaus stations (Ziese et al., 2018). The $\delta^{18}O$ values (left y axis) and precipitation (right y axis) for each station were obtained from the GNIP IAEA/WMO database. Stars indicate the site locations: 1) Trapiá Cave, Furna Nova Cave and Pedra das Abelhas ANA Station (reference period 1910-2019), 2) Boqueirão Lake (Utida et al., 2019), 3) Diva de Maura Cave (Novello et al., 2012) and Andaraí ANA Station (reference period 1960-1986), 4) Paraíso Cave (Wang et al., 2017) and Belterra ANA Station (reference period 1975-2007), 5) Cariaco Basin (Haug et al., 2001).*

Figure 3: Please check if colors are color-blind friendly (red and green mixed…?)

*We updated all figures to render them suitable for color blind people and we checked them using the website that simulates color blindness, as suggested by the journal. Please see the updated version of Figure 3 in this comment. Please note that we merged Figure 3 and 4 because of the overlapping data. The complete TRA7 record is in the supplemental material. Please see Figure S7 below.*

[Figure]

*Figure 3 – Rio Grande do Norte stalagmite isotope records and comparisons with other records from South America. a) U/Th ages from each stalagmite studied. b) Raw data of δ¹³C. c) Oxygen isotope results corrected for calcite-aragonite fractionation (δ¹⁸O_C-A), according to weight proportion of mineralogical results. d) δ¹⁸O RN Composite constructed using stalagmite records from NEB (black line). Grey shaded area denotes 99% confidence interval of age model. e) Boqueirão Lake δD record (Utida et al., 2019). f) DV2 δ¹⁸O*

*speleothem record from Diva de Maura cave, southern NEB (Novello et al., 2012). g) PAR01 and PAR03 δ¹⁸O records from Paraíso cave stalagmites, eastern Amazon (Wang et al., 2017).  h) Ti record of Cariaco Basin (Haug et al., 2001).*

[Figure]

*Figure S7 – Oxygen isotope records and age model results calculated by ISCAM for individual stalagmites and Composite. The normalization of the data is made by ISCAM (Fohlmeister, 2012).*

Also, why is the early phase of TRA7 between 3 and 2k not included in the composite?

*This part of the TRA7 stalagmite was not included in the ISCAM composite, because this interval was not the focus of our discussion. Even though it is new data, most of its interpretation is related to the 4.2 ky BP event and was described previously by Cruz et al. (2009) and Utida et al. (2020). Therefore, and following the suggestion of Reviewer 1, the Figure 3 was merged with Figure 4, and this older part of TRA7 was included in the Supplementary Material.*

**Supplementary material**

Tables S1, S2, S3: Please check decimal and 1000s delimiter, there are different styles used (comma and points mixed, sometimes comma as 1000s delimiter, sometimes not). Also "delta"234U instead of d234U.

*Thank you for mentioning the lack of harmonization. All data will be delimited consistently by using periods. The delta notation was also corrected in Tables S1, S2 and S3.*

Figure S4: Any ideas for the outliers, e.g., in TRA7 or FN1? Also, why is the age model of FN1 systematically older than the stalagmite ages? Also, why do you show ISCAM uncertainties, but COPRA average age model? Why not show ISCAM and COPRA in comparison?

*The outliers for TRA7 and FN1 were discussed above. We will include a more complete description of the U/Th ages when we submit our revision. The outliers can be explained by the [232]Th content and*

$^{230}$Th/$^{232}$Th *results. Please, see our detailed response to this question in the third paragraph of this RC2 response, when discussing U/Th dating results.*

*We decided to show COPRA age models, because the age model of ISCAM failed to produce reasonable extrapolations for the first and last millimeters of the stalagmites or to bridge intervals where we had identified a possible hiatus such as in the sample FN1. COPRA produces an independent linear age model allowing us to evaluate them without changes as made by ISCAM. However, both age model methods use a linear interpolation and produce very similar results. The plot with two time series does not show any significant differences between them; hence the choice of age model does not affect our interpretation. Please see figure below.*

*We also revisited the age models and the caption of Figure S4. The systematically older ages for FN2 were the result of a plotting error. We also corrected the text concerning the age model errors in the caption. This was not an ISCAM age model error, but a COPRA age model error. We corrected the graph and caption and present the revised version below (Figure S4).*

[Figure]

*Figure S4 – Age models for each stalagmite from Rio Grande do Norte. Age models were calculated using COPRA (Breitenbach et al., 2012) through a set of 2.000 Monte Carlo simulations. The COPRA age model was produced for each sample and covers the entire stalagmite. Squares and horizontal bars: age results with error bars. Red line: COPRA average age model. Grey line: age model errors considering 95% confidence interval.*

*References*

*Alvalá, R.C.S., Cunha, A.P.M.A., Briton, S.S.B., Seluchi, M.E., Marengo, J.A., Moraes, O.L.L., Carvalho, M.A.: Drought monitoring in the Brazilian Semiarid region, An. Acad. Bras. Ciênc., 91 (1), e20170209, https://doi.org/10.1590/0001-3765201720170209, 2019.*

*Andreoli, R. F. S., de Souza, R.A.F., Kayano, M.T., Candido, L.A.: Seasonal anomalous rainfall in the central and eastern Amazon and associated anomalous oceanic and atmospheric patterns, Int. J. Climatol., 32, 1193–1205, https://doi.org/10.1002/joc.2345, 2011.*

Black, D.E., Peterson, L.C., Overpeck, J.T., Kaplan, A., Evans, M.N., Kashgarian, M.: Eight centuries of North Atlantic Ocean atmosphere variability, Science 286, 1709–1713, https://doi.org/10.1126/science.286.5445.1709, 1999.

Breitenbach, S.F.M., Rehfeld, K., Goswami, B., Baldini, J.U.L., Ridley, H. E., Kennett, D. J., Prufer, K.M., Aquino, V.V., Asmerom, Y., Polyak, V.J., Cheng, H., Kurths, J., Marwan, N.: COnstructing Proxy Records from Age models (COPRA), Clim. Past, 8, 1765–1779, https://doi.org/10.5194/cp-8-1765-2012, 2012.

Cheng, H., Edwards, R.L., Shen, C-C., Polyak, V.J., Asmerom, Y., Woodhead, J., Hellstrom, J., Wang, Y., Kong, X., Spötl, C., Wang, X., Alexander Jr. E.C.: Improvements in $^{230}$Th dating, $^{230}$Th and $^{234}$U half-life values and U-Th isotopic measurements by multi-collector inductively coupled plasma mass spectrometry, Earth Planet. Sci. Lett., 371-372, 82-91, https://doi.org/10.1016/j.epsl.2013.04.006, 2013.

Cunningham , L.K., Austin, W.EN., Knudsen, K.L. et al.: Reconstructions of surface ocean conditions from the northeast Atlantic and Nordic seas during the last millennium, Holocene 23, 921–935, https://doi.org/10.1177/0959683613479677, 2013.

Edwards, R. L., Cheng, H., Wasserburg, J.: 238U- 234U-230Th- 232Th systematics and the precise measurement of time over the past 500,000 years, Earth Planet. Sci. Lett., 81, 175-192, https://doi.org/10.1016/0012-821X(87)90154-3, 1987.

Haug, G., Hughen, K.A., Sigman, D.M., Peterson, L.C., Röhl, U.: Southward migration of the Intertropical Convergence Zone through the Holocene, Science, 293, 5533, 1304-1308, https://doi.org/10.1126/science.1059725, 2001.

He, Z., Dai, A., Vuille, M.: The joint impacts of Atlantic and Pacific multidecadal variability on South American precipitation and temperature. J. Climate, 34(19), 7959-7981. https://doi.org/10.1175/JCLI-D-21-0081.1, 2021.

Jaffey, A.H., Flynn, K.F., Glendenin, L.E., Bentley, W.C., Essling, A.M., Precision measurement of half-lives and specific activities of 235U and 238U. Phys. Rev. C 4, 1889-1906, https://doi.org/10.1103/PhysRevC.4.1889, 1971.

Knight, J.R., Folland, C.K., Scaife, A.A.: Climate impacts of the Atlantic Multidecadal Oscillation, Geophys. Res. Lett., 33, L17706, https://doi.org/10.1029/2006GL026242, 2006.

Kayano, M.T., Andreoli, R.V., de Souza, R.A.: Pacific and Atlantic multidecadal variability relations to the El Niño events and their effects on the South American rainfall, Int. J. Clim., 40(4), 2183-2200, https://doi.org/10.1002/joc.6326, 2020.

Kayano, M.T., Cerón, W.L., Andreoli, R.V., Souza, R.A.F., Avila-Diaz, A., Zuluaga, C.F., Carvalho, L.M.V.: Does the El Niño-Southern Oscillation Affect the Combined Impact of the Atlantic Multidecadal Oscillation and Pacific Decadal Oscillation on the Precipitation and Surface Air Temperature Variability over South America?, Atmos., 13, 231, https://doi.org/10.3390/atmos13020231, 2022.Lachniet, M.S., Bernal, J.P., Asmerom, Y., Polyak, V.: Uranium loss and aragonite calcite age discordance in a calcitized aragonite stalagmite. Quat. Geochron., 14, 26-37, http://dx.doi.org/10.1016/j.quageo.2012.08.003, 2012.

Lapointe, F., Bradley, R.S., Francus, P., Balascio, N.L., Abbott, M.B., Stoner, J.S., St-Onge, G., De Coninck, A., Labarre, T.: Annually resolved Atlantic Sea surface temperature variability over the past 2,900 y, Proc. Natl. Acad. Sci., 117, 44, 27171–27178, https://doi.org/10.1073/pnas.2014166117, 2020.

Levine, A.F.Z., Frierson, D.M.W., McPhaden, M.J.: AMO Forcing of Multidecadal Pacific ITCZ Variability, J. Clim., 31, 5749–5764, https://doi.org/10.1175/JCLI-D-17-0810.1, 2018.

Lima, J.R., Magalhães, A.R.: Secas no Nordeste: registros históricos das catástrofes econômicas e humanas do século 16 ao século 21, Parcer. Estratég., 23, 46, 191-212, 2018. Available at: https://seer.cgee.org.br/parcerias_estrategicas/article/view/896/814.

Maksic J., Shimizu, M.H., Kayano, M.T., Chiessi, C.M., Prange, M., Sampaio, G.: Influence of the Atlantic Multidecadal Oscillation on South American Atmosphere Dynamics and Precipitation, Atmos., 13, 11, 1778, https://doi.org/10.3390/atmos13111778, 2022.

Mann, M.E., Zhang, Z., Rutherford, S., Bradley, R.S., Hughes, M.K., Shindell, D., Ammann, C., Faluvegi, G., Ni, F.: Global-scale signatures and dynamical origins of the Little Ice Age and Medieval Climate Anomaly, Science 326, 1256–1260, https://doi.org/10.1126/science.1177303, 2009.

Marengo, J.A., Bernasconi, M.: Regional differences in aridity/drought conditions over Northeast Brazil: present state and future projections, Clim. Chang., 129, 103-115, https://doi.org/10.1007/s10584-014-1310-1, 2015.

Marengo, J.A., Galdos, M.V., Challinor, A., Cunha, A.P., Marin, F.R., Vianna, M.S., Alvala, R.C.S., Alves, L.M., Moraes, O.L., Bender, F.: Drought in Northeast Brazil: A review of agricultural and policy adaptation options for food security, Clim. Resil. Sustain., 1, 17, https://doi.org/10.1002/cli2.17, 2021.

Miettinen, A., Divine, D. V., Husum, K., Koç, N., Jennings, A.: Exceptional ocean surface conditions on the SE Greenland shelf during the medieval climate anomaly, Paleoceanogr., 30, 1657–1674, https://doi.org/10.1002/2015PA002849, 2015.

Novello, V.F., Cruz, F.W., Vuille, M., Campos, J.L.P.S., Stríkis, N.M., Apaéstegui, J., Moquet, J.S., Azevedo, V., Ampuero, A., Utida, G., Wang, X., Paula-Santos, G.M., Jaqueto, P., Pessenda, L.C.R., Breecker, D.O., Karmann, I.: Investigating d13C values in stalagmites from tropical South America for the last two millennia, Quat. Sci. Rev., 255, 106822, https://doi.org/10.1016/j.quascirev.2021.106822, 2021.

Reynolds, D.J., Scourse, J.D., Halloran, P.R., Nederbragt, A.J., Wanamaker, A.D., Butler, P.G., Richardson, C. A., Heinemeier, J., Eiríksson, J., Knudsen, K.L. , Hall, I. R.: Annually resolved North Atlantic marine climate over the last millennium, Nat. Commun. 7, 13502, https://doi.org/10.1038/ncomms13502, 2016.

Richards, D., Dorale, J.: Uranium-series chronology and environmental applications of speleothems, Rev. Mineral., 52, 407-460, https://doi.org/10.2113/0520407, 2003.

Spooner, P.T., Thornalley, D.J.R., Oppo, D.W., Fox, A.D., Radionovskaya, S., Rose, N.L., Mallett, R., Cooper, E., Roberts, J.M.: Exceptional 20th century ocean circulation in the northeast Atlantic, Geophys. Res. Lett. 47, e2020GL087577, https://doi.org/10.1029/2020GL087577, 2020.

Utida, G., Cruz, F.W., Etourneau, J., Bouloubassi, I., Schefuß, E., Vuille, M., Novello, V., Prado, L.F., Sifeddine, A., Klein, V., Zular, A., Viana, J.C.C., Turcq, B.: Tropical South Atlantic influence on Northeastern Brazil precipitation and ITCZ displacement during the past 2300 years, Sci. Rep., 9, 1698, https://doi.org/10.1038/s41598-018-38003-6, 2019.

Wang, J., Yang, B., Ljungqvist, F.C., Luterbacher, J., Osborn, T.J., Briffa, K.R., Zorita, E.: Internal and external forcing of multidecadal Atlantic climate variability over the past 1,200 years, Nat. Geosci. 10, 512–517, https://doi.org/10.1038/ngeo2962, 2017.

Ziese, M., Rauthe-Schöch, A., Becker, A., Finger, P., Meyer-Christoffer, A., Schneider, U.: GPCC Full Data Daily Version 2018 at 1.0°: Daily Land-Surface Precipitation from Rain-Gauges built on GTS-based and Historic Data [datset], https://doi.org/10.5676/DWD_GPCC/FD_D_V2018_100, 2018.

---

## Author Comment (AC3)

*We are grateful for the community comments on manuscript cp-2023-2. We addressed the community's comments below in italicized text.*

CC1

Comments on "Spatiotemporal ITCZ dynamics during the last three millennia in Northeastern Brazil and related impacts in modern human history."

Authors: Giselle Utida, Francisco William Cruz, Mathias Vuille, Angela Ampuero, Valdir F. Novello, Jelena Maksic, Gilvan Sampaio, Hai Cheng, Haiwei Zhang, Fabio Ramos Dias de Andrade, and R. Lawrence Edwards

This is an interesting study that uses speleothem $\delta18O$ and $\delta13C$ records to characterize the nuanced behavior of the ITCZ/tropical rain belt and its impact on the regional hydroclimate (i.e., precipitation variability) of Nordeste and eastern Amazona during the late Holocene. The main objective of this study is to improve the interpretation of late Holocene ITCZ dynamics in the South American tropics, which may help to better our understanding of past SASM variability. Additionally, their interpretation of RN $\delta18O$ as a recorder of extreme dry events during the last 500 years has archeological and societal implications. This manuscript presents several thought-provoking and novel ideas pertaining to Atlantic and Pacific impacts on ITCZ-related precipitation during the late Holocene, which have the potential to reconcile paleoclimate records from Nordeste and Amazonia. Overall, this study also has the potential to be an excellent contribution to the field of South American paleoclimatology. However, I find that the manuscript (in its present state) has several major issues, which require further consideration, detail, and development before it should be accepted for publication. As such, I would recommend major revisions of the manuscript before final acceptance.

**Major issues:**

**1. I am concerned that the AMV reconstruction presented in figure 5 (also referenced in the main text) is misleading. Specifically:**

Figure 5 (and lines 451–454): It is true that the presented AMV time series and the RN composite $\delta18O$ time series look similar, but it is unclear what the authors are plotting. The green time series in figure 5 (shown below, top figure) does not look like the AMV reconstruction from Lapointe et al. (2020) (shown below, bottom figure)—raw data from https://www.ncei.noaa.gov/access/paleo-search/study/31353. The full range of values from the Lapointe dataset is 21.7–22.7, while the reconstruction shown in figure 5 only appears to be from 21.95 to 22.40.

Perhaps the authors plotted a different reconstruction of the AMV and used the wrong citation? Or perhaps it is the reconstruction from Lapointe et al. (2020) but downsampled (if so, the authors need to make this clear in the methods or supplementary information)?

*Thank you for your comment and question. We addressed this point in the response to reviewer`s comments (RC1, comment 19). We incidentally plotted the AMV curve (Lapointe et al., 2020) backwards in the original manuscript. We have corrected the figure and the discussion based on the AMV. We invite you to read our response to the reviewers as it should help clarify your question.*

**2. The authors do not sufficiently explain the mechanisms driving the anti-phased behavior observed between the RN composite and Paraíso Cave δ18O records. Specifically:**

Lines 436–440: It is unclear what is meant by "a zonal behavior of precipitation shifts in the ITCZ domain." Are the authors proposing that RN and Paraíso are in-phase from 250–1100 CE, anti-phased at ~1100 CE, back in-phase from 1100–1500 CE, and then anti-phased again from 1500–1750 CE? The authors should provide more explanation for this behavior.

Additionally, the authors state that "even though the Paraíso and Cariaco sites are located in different hemispheres, the observed in-phase climate relationship during the LIA suggests that their isotopic signatures were both sensitive to the same rainfall changes over northern South America." The Cariaco record is not an isotope-based record. Rather, it is a bulk titanium % record. The wording of this sentence should be changed accordingly.

*The Paraíso record cannot be interpreted in the same way as the RN record that predominantly receives rainfall originating from the ITCZ, while the Paraíso Cave is located at the margin of two different systems, the ITCZ and the South American Summer Monsoon (SASM), as described in our Climatology section (Figure 2). The location of Paraiso at the very edge of the SASM region likely explains why during certain intervals it varies in-phase and during others out of phase with the RN record. As shown by Orrison et al. (2022) during the last millennia the Paraiso record tends to be out of phase with the core monsoon region as a result of Bolivian-High-Nordeste Low intensification. However, a slight zonal shift of this leading mode of monsoon variability would change this relationship, as the Paraiso record would become part of the monsoon system, leaving it antiphased with the subsidence region over NE Brazil, where the RN record is located. Hence, the location of Paraiso at the node of this dipole, renders its response very sensitive to slight changes in the monsoon core. Furthermore, the zonal precipitation gradient between northeastern Brazil and the eastern-central Amazon is highly sensitive to changes in Pacific and Atlantic SST on multidecadal timescales. As shown by He et al. (2021), during the monsoon season (DJF), the zonal precipitation gradient response to Pacific SST variability completely reverses in this region, depending on the state of the Atlantic (see Figure 7 in He et al., 2021) and this change is transmitted via a perturbed Walker circulation (see their Figure 9). We now reference this mechanism in the revised paper, but discussing in great depth the joint interactions between Pacific and Atlantic and how they perturb Hadley and Walker circulation, respectively, is beyond the scope of this paper. We refer the interested reader to He et al. (2021) instead.*

*We have also revised the text in order to clarify that Cariaco is not an isotopic record.*

*We have rewritten this paragraph to adjust the discussion about the RN Composite the Paraíso and Cariaco records, respectively, according to suggestions we received from RC1 and 2. Please see our revised version below.*

*"When comparing N-NEB and eastern Amazon conditions, it is evident that the RN Composite shares some similarities with the Paraiso stalagmite record (Wang et al., 2017), due to the contribution of ITCZ precipitation in both places. But there are also important differences (Fig. 4). The RN Composite shows lower $\delta^{18}O$ values between 500 and 1000 CE, compared to the earlier period, while Paraiso shows decreasing values around the same period, suggesting a slight increase in precipitation in both areas. From 1160 to 1500 CE, abrupt increases in $\delta^{18}O$ values are seen in both records, which indicates abrupt and prolonged drought conditions due to a northward ITCZ migration. However, around 1100 CE, and the period from 1500 to 1750 CE, Paraiso is antiphased with the RN Composite and in phase with the Cariaco Basin (Haug et al., 2001), which is inconsistent with the notion of an ITCZ-induced regional precipitation change.*

*Instead, a zonally-oriented precipitation change within the ITCZ domain over Brazil is required to explain the anti-phased behavior between precipitation in N-NEB and the eastern Amazon, and similarities between Cariaco and the eastern Amazon."*

Lines 446–451: Here, the authors discuss the AMV and ITCZ displacement during a warm AMV. However, the authors have not defined what a warm AMV is, albeit the reader could find out in the cited studies. I recommend the authors specifically define the AMV in detail, and make clear what is meant by a warm vs cold AMV.

*We will clarify in the revised manuscript how warm and cold AMV are defined.*

Lines 461–463: The authors state, "Our analysis corroborates with this and points to increasing precipitation over N-NEB and decreasing precipitation over eastern Amazon, between 1500–1750 CE, when both AMV and PDV are in cold phase (Fig 4)." There is no reference to the PDV in figure 4, nor has the PDV been described/defined yet at this point in the text. No PDV reconstructions are provided in any of the figures, and the provided AMV reconstruction is in figure 5, not figure 4. Last millennium SST gradients from Steinman et al. (2022) are provided in figure 5, but they are not PDV or AMV reconstructions. I recommend either including a PDV reconstruction in one of the figures, or to remove this text from the manuscript.

Lines 463–465: The authors state, "This sign reversal is assigned to perturbations of the regional Walker cell's produced by teleconnection between the Atlantic and Pacific (Kayano et al., 2022, He et al., 2021)." I find this explanation to be vague, and recommend that the authors provide a clearer and more detailed explanation for the sign reversal. What does "perturbations of the regional Walker cell's" mean exactly? What teleconnections are the authors referring to, and what are the mechanisms driving the aforementioned perturbations?

*The Figure presents only the AMV. The discussion about the relationship between AMV and PDV was only based on an observed precipitation analysis. We made some adjustments in the paragraph to clarify the aspects mentioned in the last two comments. Please see the revised text below.*

*"According to Kayano et al. (2020, 2022), during the last century, dry conditions over N-NEB and the eastern Amazon are present when AMV and Pacific Decadal Variability (PDV) are in both in their warm phase, or when the AMV is in a cold phase and the PDV in its warm phase. On the other hand, when AMV and PDV are both in their cold phase, precipitation over the Amazon is anti-phased with NEB, resulting in decreased precipitation over the Amazon and increased precipitation over NEB. This zonally aligned precipitation signal over eastern tropical S. America is the result of joint perturbations of both the regional Walker and Hadley Cell's, produced by teleconnection between the Atlantic and Pacific (Kayano et al., 2022, He et al., 2021). These conditions can explain in part our results, however during the decoupling of our record with AMV (between 1500 and 1750 CE), increasing precipitation over N-NEB and decreasing precipitation over the eastern Amazon can be better explained by the positive gradients both in Atlantic and Pacific Oceans forcing a south ITCZ migration (Fig. 4)."*

**3. The conclusion and abstract both discuss ITCZ dynamics forced by the AMV and PDV, including position, intensity, and width. However, in the main text, the authors do not sufficiently explain which dynamical aspect of the ITCZ responds to different AMV/PDV phases, nor do they explain any mechanism(s) behind the AMV/PDV forcing. Specifically:**

Lines 570–577: In this paragraph, the authors suggest that during the last millennia, ITCZ dynamics cannot be explained solely by north-south ITCZ migrations or one single forcing mechanism. They propose a zonally non-uniform behavior of the ITCZ during times when the RN 4 record is anti-phased with the Paraíso cave record—forced by the interactions between the AMV and PDV modes that changed the regional Walker cell position and ITCZ intensity/width.

However, the authors never really attributed the anti-phased behavior between N-NEB and eastern Amazonia to the differential AMV/PDV phases. They discussed observed precipitation anomalies during overlapping periods of AMV and PDV phases in the modern, and suggested that it could be responsible for the observed anti-phased behavior. However, they never directly compared the speleothem time series with AMV and PDV reconstructions. Nor did the authors propose a detailed mechanism for how different AMV/PDV phases impact ITCZ width/intensity, despite changes in ITCZ width/intensity also being mentioned in the abstract (lines 46–50). In addition, the authors did not really describe when the ITCZ may have expanded/contracted or became weaker/stronger (aside from stating that this may have happened when the RN composite record and Paraíso are anti-phased). Ultimately, they never describe mechanism(s) for 1) how different AMV/PDV phases impact ITCZ dynamics, 2) how changes in ITCZ width/intensity may cause the observed anti-phased behavior, and 3) how the regional Walker cell position is forced by different AMV/PDV phases. I recommend that the authors provide more detail to this part of the Conclusions and Discussion sections overall, and propose/explain specific mechanisms that can reconcile the observed hydroclimate variability in N-NEB and eastern Amazonia.

We have responded to this comment above.

Additional note: The authors should be extremely clear when generally discussing ITCZ width/intensity. What exactly do the authors mean by ITCZ width? Is it the width of the actual band of deep convection? Width of the seasonal range of the ITCZ? These terms should be explicitly defined early in the manuscript. Some papers that may be useful to reference include Donohoe et al. (2013), Atwood et al. (2020), Byrne and Schneider (2016), and Roberts et al. (2017).

*The ITCZ definition adopted is the one referring to it as the modern tropical rain belt of maximum precipitation and the ITCZ position is defined according to Schneider et al. (2014). The position is mentioned in line 160 of the manuscript, when we define the locality of our study site and its relationship with the ITCZ. We will add to this definition by including the ITCZ position during the boreal winter over the Atlantic (2° N). We will change the term "ITCZ width" by "ITCZ length". We were referring to the duration of the ITCZ over N-NEB, from March to May, during its southernmost extent, but we did not intend to imply a specific ITCZ dimension. We will rephrase how we refer to the ITCZ's southernmost expansion in MAM to avoid confusion.*

**Additional comments and concerns:**

Lines 89–92: The authors cite Lechleitner et al. (2019), but I believe the correct citation is Lechleitner et al. (2017). Additionally, another relevant citation that may be relevant and could be included here is Asmerom et al. (2020) published in Science.

*The citation will be corrected in the text. We will consider including other references as appropriate in the manuscript.*

Lines 95–102: The authors call out the SASM and the ITCZ here as focus points of recent studies on tropical South American precipitation, but have not mentioned the South Atlantic Convergence Zone (SACZ). While not explicitly relevant to their findings, the SACZ should at least be mentioned here because of its important relationship with the SASM and ITCZ, and because it has been the topic of several recent paleoclimate and modern precipitation studies (Novello et al., 2018; Nielsen et al., 2019; Zilli et al., 2019; Wong et al., 2021).

*We will include a brief discussion of the SACZ in the Introduction section, although the SACZ is not directly responsible for the precipitation observed at our study sites.*

Figure 1: It may help the reader to include annotations in the figure, including labeling the core SASM domain, ITCZ location, SACZ, etc. Additionally, while I understand the choice to include austral autumn precipitation climatology (when N-NEB receives most of its precipitation), it may be worthwhile to include panels with precipitation climatology for the austral winter and spring (either added to figure 1 or included in the supplement). This would allow for the reader to visually assess the spatiotemporal dynamics of the ITCZ, SASM, and SACZ, and how precipitation varies at sites 1–4 during the different seasons. 5

*We will consider including the annotations of ITCZ, SASM and SACZ in Figure 1. However, including fractional precipitation panels for JJA and SON does not add much relevant information for our region as precipitation at this time of year is low (see panels below). We therefore prefer to focus on the key rainy seasons DJF and MAM.*

[Figure]

Lines 165–174: Figure S1 receives a lot of attention in this paragraph, and should probably be included as a main text figure. Alternatively, it could be incorporated into an existing main figure.

*We will change this section and include the text related to the climatology of the region and Figure 2 in the results section instead, following to RC2's suggestion. Certainly, Figure S1 can be included in the main text.*

Figure 2: Readers who are green-red colorblind will not be able to see the small green dots (that denote the location of the GNIP stations) in any of the panels. I recommend changing the color to black and potentially increasing the size of the dots.

*The journal editorial team already mentioned that we had to adapt the figure for color blind readers during the revision stage. All figures in the manuscript are now adapted accordingly.*

Lines 362–363: It gets confusing when the authors use both before present (BP) dates and before common era/common era (BCE/CE) dates. Additionally, ky has not been defined before this point, so the authors should spell it out before using the abbreviation.

*The nomenclature of the time periods will be standardized.*

Figure 3: Same red–green issue as mentioned in Figure 2.

*The journal editorial team already mentioned that we had to adapt the figure for color blind readers during the revision stage. All figures in the manuscript are now adapted accordingly.*

Figure 4: It would be extremely helpful for the authors to include vertical bars when referencing specific time periods in the text. Such periods include the LIA, MCA, Bond 2 event, etc. Additionally, the authors reference trends resulting from insolation forcing in the paragraph starting at line 417.

The authors should consider including a time series of solar insolation.

*We have included the insolation curve and also vertical bars in Figure 3 and 4 of the manuscript. This Figure in question already contains a lot of information and adding even more would make it difficult to read. We will instead add the insolation curve to the Figure S5 in the Supplement as shown below.*

Also, the δD record from Boqueirão Lake is relative to VSMOW, not VPDB (Utida et al., 2019). This appears to be a typo and should be changed accordingly.

*We have corrected this typo. Thank you for drawing attention to it.*

Lines 389–392: The authors state that from 1060 to 480 BCE, there was increased precipitation in N-NEB as suggested by negative δ18O anomalies. But it is unclear what the authors mean by 'increased precipitation'. During this time, there is multidecadal variability in the RN composite δ18O record, but no clear/obvious trend between 1060 and 480 BCE. Perhaps the authors meant that there was increased precipitation *relative* to another part of the record. I would recommend clearing this up.

*We have rewritten this paragraph to clarify our statement. Please see below.*

*"The oldest period covered by the RN Composite, from 1200 to 500 BCE, is characterized by successive dry and wet multidecadal periods, with increased precipitation in N-NEB from 1060 to 750 BCE and from 460 to 290 BCE, as suggested by the negative departures seen in the $\delta^{18}O$ values. During this last period, there is also a tendency from lower to higher $\delta^{13}C$ values, suggesting progressive surface soil erosion*

*related to rainfall variability (Fig. 4), as interpreted by Utida et al. (2020). This period ends in a stable interval, lasting from 300 BCE to 0 CE, with $\delta^{13}C$ values close to the bedrock signature at about -1‰ to +1‰, indicating a lack of soil above the cave. After an abrupt reduction of $\delta^{13}C$, the values decrease to approximately -2‰ between 200 CE and 1500 CE. From 1500 CE to the present, negative values of $\delta^{13}C$ represent wet climatic conditions as indicated by lower $\delta^{18}O$ values. The more negative $\delta^{13}C$ during this period can be related to denser vegetation that favored both soil production and stability above the cave."*

Lines 408–409: The authors reference the δD record from Boqueirão Lake, and the same record is shown in figure 4. However, the authors describe the record as a "δD lipids" record. Lipids are a broad group of molecules which include waxes, glycerides, terpenoids, tetrapyrrole pigments, etc. The authors should be more specific, and should reference the record as a leaf wax δD record of *n*-C28 alkanoic acids from Boqueirão Lake sediments (hereinafter referred to as δD lipids).

*We will include the description "n-C28 alkanoic acid obtained in leaf waxes" when first discussing the δD record from Boqueirão Lake (Utida et al., 2019).*

Lines 495–497: The authors focus their discussion of extreme dry events recorded in the TRA5 δ18O record between 1500 and 1850 CE. However, it is unclear why the authors do not discuss dry events/distinct δ18O peaks after 1850 CE, despite their record extending into the 21st century. Is it because the TRA5 speleothem chronology is not as precise during this time?

*The TRA5 chronology during the last 150 years is indeed not precise enough to discuss historical events. Since we will improve our discussion of age models, according to RC's2 suggestion, the TRA5 chronology will also be better explained and we will clarify this question in the updated version of the manuscript.*

Lines 518–523 and Figure 6: The authors reference several historical droughts that had severe societal/socioeconomic consequences. It may be helpful to annotate figure 6 to highlight the most severe droughts referenced in the text. The number/letter labeling in figure 6 makes it hard to discern the severity of the droughts by looking at the figure alone. 6

*According to the historical records, the most significant drought events registered in our stalagmite are related to points 4, 6 and 7. They will be highlighted in Figure 6 and mentioned in the caption.*

Lines 533: The authors should provide more detail here. Which Governor are the authors referring to? Governor of what/where?

*The Governor mentioned here is the Brazilian Governor. In that period, Brazil was a colony of Portugal and there was a local government. We will specify this in the text as "Brazilian Governor".*

Figure 6: Why focus on just TRA5? TRA7 and FN1 appear to cover the same period as TRA5. Is TRA5 the only speleothem that records the extreme drought events? Do TRA7 or FN1 record any of the same drought events? If they do not, why would only one speleothem record these drought events and not the others?

*Thank you for your comments and question. We addressed this point in the reviewer's comments. We invite you to please read our response in those files and hope they will help clarify your question.*

It may be helpful to include the age uncertainty in the right panel of the figure under the heading "TRA5". For example, 1546 ± XX. Especially because this figure focuses on only the last 500 years, it would allow the reader to critically compare the speleothem dates to the historical drought dates listed in the column labeled "Historical."

*Thank you for your comments and question. We addressed this point in detail in the RC2's comments considering the U/Th ages and age model. We invite you to please read our response in those files and hope they will help clarify your question.*

Additionally, I am curious if there is an available archeological record(s) or something similar that could be plotted with the TRA5 δ18O record. Especially since the authors discuss the societal implications of the extreme droughts in relation to human population and welfare, it would be useful for the reader to visualize the impact through comparison with the speleothem record.

*The Brazilian archeological records were discussed and compared with stalagmite data during the Holocene by Utida et al. (2020). However, these data basically describe the total population size during random intervals (https://memoria.ibge.gov.br/historia-do-ibge/historico-dos-censos/dados-historicos-dos-censos-demograficos.html) and they are not helpful to discuss episodic extreme events. Furthermore, considering the lack of demographic data in Brazil, from 1500 to 1870 CE, such a comparison with the stalagmite record, unfortunately, is not feasible.*

Line 565–567: The authors state, "The N-NEB record presents a trend toward drier conditions as is also being observed in the Diva de Maura Cave in S-NEB, interpreted as an ITCZ withdrawal and SASM weakening, respectively." It is unclear what the authors mean by "ITCZ withdrawal," especially since the authors highlighted the dynamical behavior of the ITCZ earlier in the paper. Is it a withdrawal via mean ITCZ displacement? Contraction or weakening of the ITCZ? More detail here would be helpful for the reader.

'Withdrawal' of the ITCZ was meant to indicate that it's mean position moved northward. We will clarify this in the revised version of the manuscript.

[Figure]

*Figure S5 – Rio Grande do Norte stalagmite isotope record. (a) U/Th ages for RN stalagmites. (b) Raw data of $\delta^{13}C$. (c) Oxygen isotope results corrected for calcite-aragonite fractionation ($\delta^{18}O_{C-A}$), according to weight proportion of mineralogical results. (d) $\delta^{18}O$ RN Composite constructed using stalagmite records from NEB (black line). Grey lines denote the age model confidence interval of 99%. (e) February insolation curve at 10°S.*

**References:**

Asmerom, Y., Baldini, J. U. L., Prufer, K. M., Polyak, V. J., Ridley, H. E., Aquino, V. V., Baldini, L. M., Breitenbach, S. F. M., Macpherson, C. G., and Kennett, D. J.: Intertropical convergence zone variability in the Neotropics during the Common Era, Sci. Adv., 6, eaax3644, https://doi.org/10.1126/sciadv.aax3644, 2020.

Atwood, A. R., Donohoe, A., Battisti, D. S., Liu, X., and Pausata, F. S. R.: Robust Longitudinally Variable Responses of the ITCZ to a Myriad of Climate Forcings, Geophysical Research Letters, 47, https://doi.org/10.1029/2020GL088833, 2020.

Byrne, M. P. and Schneider, T.: Energetic Constraints on the Width of the Intertropical Convergence Zone, Journal of Climate, 29, 4709–4721, https://doi.org/10.1175/JCLI-D-15-0767.1, 2016.

Donohoe, A., Marshall, J., Ferreira, D., and Mcgee, D.: The Relationship between ITCZ Location and Cross-Equatorial Atmospheric Heat Transport: From the Seasonal Cycle to the Last Glacial Maximum, Journal of Climate, 26, 3597–3618, https://doi.org/10.1175/JCLI-D-12-00467.1, 2013.

He, Z., Dai, A., and Vuille, M.: The joint impacts of Atlantic and Pacific multidecadal variability on South American precipitation and temperature, Journal of Climate, 1–55, https://doi.org/10.1175/JCLI-D-21-0081.1, 2021.

Kayano, M. T., Cerón, W. L., Andreoli, R. V., Souza, R. A. F., Avila-Diaz, A., Zuluaga, C. F., and Carvalho, L. M. V.: Does the El Niño-Southern Oscillation Affect the Combined Impact of the Atlantic Multidecadal Oscillation and Pacific Decadal Oscillation on the Precipitation and Surface Air Temperature Variability over South America?, Atmosphere, 13, 231, https://doi.org/10.3390/atmos13020231, 2022.

Lapointe, F., Bradley, R. S., Francus, P., Balascio, N. L., Abbott, M. B., Stoner, J. S., St-Onge, G., De Coninck, A., and Labarre, T.: Annually resolved Atlantic sea surface temperature variability over the past 2,900 y, Proc. Natl. Acad. Sci. U.S.A., 117, 27171–27178, https://doi.org/10.1073/pnas.2014166117, 2020.

Lechleitner, F. A., Breitenbach, S. F. M., Rehfeld, K., Ridley, H. E., Asmerom, Y., Prufer, K. M., Marwan, N., Goswami, B., Kennett, D. J., Aquino, V. V., Polyak, V., Haug, G. H., Eglinton, T. I., and Baldini, J. U. L.: Tropical rainfall over the last two millennia: evidence for a low-latitude hydrologic seesaw, Sci Rep, 7, 45809, https://doi.org/10.1038/srep45809, 2017.

Nielsen, D. M., Belém, A. L., Marton, E., and Cataldi, M.: Dynamics-based regression models for the South Atlantic Convergence Zone, Clim Dyn, 52, 5527–5553, https://doi.org/10.1007/s00382-018-4460-4, 2019.

Novello, V. F., Cruz, F. W., Moquet, J. S., Vuille, M., De Paula, M. S., Nunes, D., Edwards, R. L., Cheng, H., Karmann, I., Utida, G., Stríkis, N. M., and Campos, J. L. P. S.: Two Millennia of South Atlantic Convergence Zone Variability Reconstructed From Isotopic Proxies, Geophys. Res. Lett., 45, 5045–5051, https://doi.org/10.1029/2017GL076838, 2018.

Roberts, W. H. G., Valdes, P. J., and Singarayer, J. S.: Can energy fluxes be used to interpret glacial/interglacial precipitation changes in the tropics?, Geophys. Res. Lett., 44, 6373–6382, https://doi.org/10.1002/2017GL073103, 2017.

Steinman, B. A., Stansell, N. D., Mann, M. E., Cooke, C. A., Abbott, M. B., Vuille, M., Bird, B. W., Lachniet, M. S., and Fernandez, A.: Interhemispheric antiphasing of neotropical precipitation during the past millennium, Proceedings of the National Academy of Sciences, 119, e2120015119, 2022. 8

Utida, G., Cruz, F. W., Etourneau, J., Bouloubassi, I., Schefuß, E., Vuille, M., Novello, V. F., Prado, L. F., Sifeddine, A., Klein, V., Zular, A., Viana, J. C. C., and Turcq, B.: Tropical South Atlantic influence on Northeastern Brazil precipitation and ITCZ displacement during the past 2300 years, Sci Rep, 9, 1698, https://doi.org/10.1038/s41598-018-38003-6, 2019.

Wong, M. L., Wang, X., Latrubesse, E. M., He, S., and Bayer, M.: Variations in the South Atlantic Convergence Zone over the mid-to-late Holocene inferred from speleothem δ18O in central Brazil, Quaternary Science Reviews, 270, 107178, https://doi.org/10.1016/j.quascirev.2021.107178, 2021.

Zilli, M. T., Carvalho, L. M. V., and Lintner, B. R.: The poleward shift of South Atlantic Convergence Zone in recent decades, Clim Dyn, 52, 2545–2563, https://doi.org/10.1007/s00382-018-4277-1, 2019.

**References**

*Atwood, A. R., Donohoe, A., Battisti, D.S., Liu, X., & Pausata, F. S. R.: Robust longitudinally variable responses of the ITCZ to a myriad of climate forcings. Geophys. Res. Lett., 47, e2020GL088833. https://doi.org/10.1029/2020GL088833R, 2020.*

*Cruz, F.W., Vuille, M., Burns, S.J., Wang. X., Cheng, H., Werner, M., Edwards, R.L., Karman, I., Auler, A.S., Nguyen, H.: Orbitally driven east-west antiphasing of South American precipitation, Nat. Geosci., 2, 210-214, https://doi.org/10.1038/ngeo444, 2009.*

*He, Z., Dai, A., Vuille, M.: The joint impacts of Atlantic and Pacific multidecadal variability on South American precipitation and temperature. J. Climate, 34(19), 7959-7981. https://doi.org/10.1175/JCLI-D-21-0081.1, 2021.*

*Kayano, M.T., Andreoli, R.V., de Souza, R.A.: Pacific and Atlantic multidecadal variability relations to the El Niño events and their effects on the South American rainfall, Int. J. Climatol., 40(4), 2183-2200, https://doi.org/10.1002/joc.6326, 2020.*

*Kayano, M.T., Cerón, W.L., Andreoli, R.V., Souza, R.A.F., Avila-Diaz, A., Zuluaga, C.F., Carvalho, L.M.V.: Does the El Niño-Southern Oscillation Affect the Combined Impact of the Atlantic Multidecadal Oscillation and Pacific Decadal Oscillation on the Precipitation and Surface Air Temperature Variability over South America?, Atmos., 13, 231, https://doi.org/10.3390/atmos13020231, 2022.*

*Lapointe, F., Bradley, R.S., Francus, P., Balascio, N.L., Abbott, M.B., Stoner, J.S., St-Onge, G., De Coninck, A., Labarre, T.: Annually resolved Atlantic Sea surface temperature variability over the past 2,900 y, Proc. Natl. Acad. Sci., 117, 44, 27171–27178, https://doi.org/10.1073/pnas.2014166117, 2020.*

*Orrison, R., Vuille, M., Smerdon, J. E., Apaéstegui, J., Azevedo, V., Campos, J. L. P. S., Cruz, F. W., Della Libera, M. E., and Stríkis, N. M.: South American Summer Monsoon variability over the last millennium in paleoclimate records and isotope-enabled climate models, Clim. Past, 18, 2045–2062, https://doi.org/10.5194/cp-18-2045-2022, 2022.*

*Schneider, T., Bischoff, T., Haug, G.H.: Migrations and dynamics of the intertropical convergence zone, Nature, 513, 7516, 45-53, https://doi.org/10.1038/nature13636, 2014.*

*Utida, G., Cruz, F.W., Etourneau, J., Bouloubassi, I., Schefuß, E., Vuille, M., Novello, V., Prado, L.F., Sifeddine, A., Klein, V., Zular, A., Viana, J.C.C., Turcq, B.: Tropical South Atlantic influence on Northeastern Brazil precipitation and ITCZ displacement during the past 2300 years, Sci. Rep., 9, 1698, https://doi.org/10.1038/s41598-018-38003-6, 2019.*

Utida, G., Cruz, F.W., Santos., R.V., Sawakuchi, A.O., Wang, H., Pessenda, L.C.R., Novello, V.F., Vuille, M., Strauss, A.M., Borella, A.C., Stríkis, N.M., Guedes, C.C.F., De Andrade, F.D., Zhang., H., Cheng, H., Edwards, R.L.: Climate changes in Northeastern Brazil from deglacial to Meghalayan periods and related environmental impacts, Quat. Sci. Rev., 250, 106655, https://doi.org/10.1016/j.quascirev.2020.106655, 2020.

Wang, J., Yang, B., Ljungqvist, F.C., Luterbacher, J., Osborn, T.J., Briffa, K.R., Zorita, E.: Internal and external forcing of multidecadal Atlantic climate variability over the past 1,200 years, Nat. Geosci. 10, 512–517, https://doi.org/10.1038/ngeo2962, 2017.

---

## Author Response (AR2)

*Point-by-point response to reviewers' comments*

*We are grateful to the editor for accepting our revised version of manuscript cp-2023-2. We indicate changes to the new version, according to the reviewers comments, below in grey text. The italicized text refers to our previously posted replies. The new version of the manuscript has been restructured to make it clearer and more complete. The main manuscript now includes six figures and the Supplement contains six figures and four tables. Other small changes have been applied throughout the manuscript in order to adapt or correct some misleading words or sentences.*
* * *
RC1: 'Comment on cp-2023-2', Anonymous Referee #1, 18 Mar 2023

"Spatiotemporal ITCZ dynamics during the last three millennia in northeastern Brazil and related impacts in modern history" presents a new composite speleothem δ18O record (using new data and previously published data) as well as a new δ13C record used to characterise precipitation and vegetation/soil cover over northeast Brazil for the Late Holocene. The authors make clear links to the necessity for this research in South America, and frame it within the context of the increased proportion of the Brazillians who experience water scarcity in modern times. By analysing samples taken from sites at the southernmost extent of the ITCZ, they are able to link periods of changed precipitation to the movement of the ITCZ.

Strengths

This is relevant research with tangible outcomes for policy. Combining multiple stalagmite proxies can overcome some of the drawbacks encountered by single-proxy studies. It is great to see the continued use of already-published data, supplemented by new data. I really enjoyed the links between the proxy record and historical climate events – finding historical climate information is non-trivial, well done to the authors for their persistence. The introduction and study set-up is good.

Weaknesses

The main weakness of the manuscript is that there is no consideration of the impact of hydrological processes on speleothem δ18O, the primary proxy of the study. Treble et al. (2022) showed in a global analysis of coeval calcite and dripwater samples that karst hydrology exerts a control on speleothem δ18O, and that the variability of δ18Oc can exceed that which can be attributed to rainfall δ18O. In the absence of cave monitoring data in the paper, the authors should add some discussion of how the karst processes at each site impact their results (or could impact their results) and how the composite handles this variability. The introduction/literature review should do also do a more thorough job of what controls δ18O in NEB. The RN composite appears to only have uncertainty in the time domain, while other composites (e.g. Kaufman et al., 2020) include uncertainty in the composited proxy value.

*Thank you for your comments. Certainly your suggestions will help us improve our manuscript in order to produce a high-quality paper.*

*We will expand our discussion of hydrological controls on $\delta^{18}O$ in stalagmites in the Introduction and Discussion sections. Unfortunately, a monitoring program cannot be successfully implemented in the studied caves because modern dripwater in these caves is very rare and intermittent, preventing an adequate monitoring program.*

The discussion about hydrological controls on $\delta^{18}O$ in stalagmites is included in lines 114-126 in the Introduction section of the new version of the manuscript.

*The hydrological processes controlling speleothem $\delta^{18}O$ will be folded into a more exhaustive literature review, as suggested. According to Treble et al. (2022), the variability of the global $\delta^{18}O$ values for speleothems originating from the same cave is ~ 0.37‰, which can attributed to karst fractionation effects. Changes in $\delta^{18}O$ of rainfall that exceed this value, are therefore, in general recorded as a climate signal in stalagmites. While some time intervals in our stalagmites from the same cave are bellow this limit, the*

*overall $\delta^{18}O$ variability in our record is much larger than 0.37‰, and we thus interpret these changes in $\delta^{18}O$ as a result of rainfall changes precipitation. Furthermore, the $\delta^{18}O$ variability recorded throughout the period analyzed, is similar for stalagmites from the same cave and between the two studied caves, further reinforcing the notion that these records can be interpreted in a paleoclimatic context. The compositing procedure has a minimal impact on the variance of the $\delta^{18}O$ record since the ISCAM procedure normalizes $\delta^{18}O$ data before combining them. As discussed further below, after normalization, the difference between stalagmite records is significantly reduced.*

The discussion about hydrological controls and mineralogical effects in the $\delta^{18}O$ in stalagmites and in the composite record is developed in lines 484-495 in the Discussion section of the new version of the manuscript.

*As far as uncertainties of the composite record are concerned, we will include revised text as listed below in the Results section (after line 349) and add a new Figure to the supplemental material (Figure S7). As discussed in Kaufman et al. (2020), there does not exist one preferred standard procedure to calculate proxy errors when a composite is produced. Unfortunately, the ISCAM program (Fohlmeister, 2012) does not return a proxy error as part of the output. It rearranges the proxies to obtain the best calculated age and then calculates the average of the proxy data after normalizing them. As outlined in the Methods section, our record includes only two overlapping stalagmites per period, as the top and base of the FN1 and FN2 stalagmites were not suitable to be used in the composite, respectively. Hence the proxy error can be quantified as the difference between the two $\delta^{18}O$ records at any point in the time. We created a new Figure showing the ISCAM-calculated ages for each stalagmite, plotted together with the final composite. We will include this Figure in the Supplement to clarify the uncertainties related to our $\delta^{18}O$ records. The figure below is already adapted for all color-blind readers, including the monochromatic view.*

*"The composite calculation rearranges the proxies in order to obtain the optimal calculated age and then calculates the average of the proxy data after normalizing the records. The RN record only contains overlapping segments between two stalagmites per period. Hence the RN composite proxy error can be quantified as the difference between the $\delta^{18}O$ of the stalagmites combined for any given point in time (Figure S7). The largest error occurs between 1460 and 1700 CE, when the maximum and minimum values of FN1 and TRA7 are 2.25 ‰ and -0.40 ‰, respectively. This is a period when FN1 registers a dry interval that is not clearly seen in TRA7. The period extending from 1370 to 1460 CE, is characterized by an anti-phased signal between FN1 and TRA7, and hence the RN Composite shows a smoothed signal during this time."*

*However, please note that the high-density of precise ages with errors of approximately 22 years in our stalagmite records, combined with similar variability between different stalagmites from the same and different caves, provide robust evidence that our isotope composite records regional climate and environmental parameters.*

We included a new paragraph in the Discussion section ("5.1. U/Th chronology and RN Composite" – Lines 438-481) to clarify all comments regarding chronology and its impacts on the RN Composite. We also adapted the oldest Figure 3, now Figure S4, and created a new figure (Figure S6) to show differences between $\delta^{18}O$ results before and after the corrections and composite production were applied.

[Figure]

*Figure S7 – Oxygen isotope and age model results calculated by ISCAM for stalagmites and Composite. The normalization of the data is performed by ISCAM (Fohlmeister, 2012).*

Figure S6 – Oxygen isotope and age model results calculated by ISCAM for stalagmites and composite. The normalization of the data is performed by ISCAM (Fohlmeister, 2012).

The above figure is included in the Supplement as Figure S6.

Specific comments and questions

1. Figure 1

Please shade either the land or the ocean to differentiate them. Please choose an accessible colour palette – the rainbow colour palette is not useful for colour blind readers.

*Thank you for bringing this issue to our attention. The journal editorial team already mentioned that we had to adapt the figure for color blind readers during the revision stage. Shading the land helped to differentiate it from the oceanic area. The color palette of Figure 1 is now more accessible. Please see the respective figure and caption below.*

[Figure]

*Figure 1 – Location and precipitation climatology of study sites during the austral summer (DJF – December to February) and autumn (MAM – March to May). Color shading indicates percentage of the annual precipitation total that is received during either DJF or MAM and highlights the extent of (a) the SASM over the continent and (b) the ITCZ over the ocean. Precipitation data is from the Global Precipitation Measurement (GPM) mission, with averages calculated over the period 2001–2020. 1) Trapiá and Furna Nova Cave (this study), 2) Boqueirão Lake (Utida et al., 2019), 3) Diva de Maura Cave (Novello et al., 2012), 4) Paraíso Cave (Wang et al., 2017), 5) Cariaco Basin (Haug et al., 2001). GNIP stations: A) Fortaleza, B) Brasília, C) Manaus.*

The new version of Figure 1 is in Line 137.

2. Line 163: please clarify whether you analysed the precipitation data as annual (or hydrological year), monthly, or daily totals.

*The data for Fortaleza, Brasília and Belterra ANA stations were analyzed on a monthly timescale. The reference period for calculating GPCC anomalies is 1961-1990. Anomalies are obtained by removing the long-term average, calculated over the reference period, from the monthly observed values. We clarified this in the text and in the caption of Figure 2.*

*"In N-NEB, we analyzed **monthly** precipitation data from Pedra das Abelhas Station – RN (Fig. 2a), from 1911 to 2015 (n=103)."*

We clarified this comment in the text (3. Materials and Methods, lines 190-192) and in the caption of Figure 3, referred as Figure 2 in the previous version.

3. Figure 2

*Figure 2 has been changed as discussed below. The revised Figure is also shown below this discussion.*

Please change green dots to another colour (black?). Please also change the green line in the top panel to a different colour.

*The green color of Figure 2 has been changed to black.*

Consider changing the red-blue colour palette – in maps this palette is often used to show temperature variability, and so I find it slightly misleading here.

*Thank you for pointing this out to us. We changed the color palette and also made additional substantive changes to the Figure to address all comments. Please see the revised Figure 2 and the associated Figure caption below.*

Please change the legend in the top panel to 'Site precipitation – GNIP' and 'Site precipitation – ANA' to be consistent with 'Site δ18O – GNIP'.

*The site description has been changed as suggested. Please see the revised Figure 2 below.*

The caption suggests that the correlation map correlates observed precipitation against observed δ18O – suggested rephrase: "Figure 2 – monthly mean observed precipitation amount for ANA stations and δ18O values for GNIP stations (IAEA-WMO, 2021) (green dots), with correlation maps between gridded precipitation anomalies and GNIP δ18O anomalies…." And then carry on from (a) with the rest of your caption, while also adding (star 1) at line 201 for Pedra das Abelhas station.

Please clarify what correlation was used.

*The caption of Figure 2 was modified according to suggestions, and the green dots were changed to black. Please see the figure and caption below. In Figure 2 we used the Pearson's correlation to produce the spatial correlation maps. This information was also included in the figure caption.*

The difference between GNIP rainfall amount and ANA rainfall amount is really large between Fortaleza and Pedras de Abelhas. These sites are so close, have you double checked that that is correct?

*The sites are close to each other indeed. However, this small distance is sufficient to slightly change the precipitation amount at these sites. Fortaleza Station is closest to the coast, and precipitation from the ITCZ is more intense than at the Pedra das Abelhas Station, which is located 88 km further inland, and thus just marginally influenced by the ITCZ. We plotted the GNIP stations' position in Figure 1 to clarify this aspect. Please see the revised Figure 1 above. Although, there are differences in precipitation amount, the precipitation trend is similar.*

[Figure]

*Figure 2 – Monthly mean observed precipitation amount collected at ANA and $\delta^{18}O$ values for GNIP stations (IAEA-WMO, 2021) (black dots) and correlation maps between gridded precipitation and $\delta^{18}O$ anomalies from the same stations (black dots) for: (a) Northern NEB, Fortaleza and Pedra das Abelhas stations (star 1), (b) Southern NEB, Brasília and Andaraí stations (star 3), c) Eastern Amazon, Manaus and Belterra stations (star 4). The maps show the spatial correlation between $\delta^{18}O$ anomalies at GNIP stations and GPCC gridded precipitation anomalies based on the period 1961-1990 for December to February (DJF) and March to May (MAM) for Fortaleza, Brasília and Manaus stations (Ziese et al., 2018). The $\delta^{18}O$ values (left y axis) and precipitation (right y axis) for each station were obtained from the GNIP IAEA/WMO database. Stars indicate the site locations: 1) Trapiá Cave, Furna Nova Cave and Pedra das Abelhas ANA Station (reference period 1910-2019), 2) Boqueirão Lake (Utida et al., 2019), 3) Diva de Maura Cave (Novello et al., 2012) and Andaraí ANA Station (reference period 1960-1986), 4) Paraíso Cave (Wang et al., 2017) and Belterra ANA Station (reference period 1975-2007), 5) Cariaco Basin (Haug et al., 2001).*

Figure 2 has been changed as discussed above and the new version is now referred to as Figure 3 (Line 351).

4. Line 184: add reference to Fig 2.

*Thank you for mentioning this. Figure 2 will be mentioned in the line 184 of the original manuscript.*

The discussion of Figure 2, now listed as Figure 3, starts on Line 320.

5. Line 190: add ref to Fig 2C

*Thank you for mentioning this. Figure 2a and 2c will be mentioned in line 190 as showing a negative spatial correlation in Northern NEB.*

The discussion of Figure 2, now listed as Figure 3, starts on Line 328.

6. Line 208: why 1960 – 2016 as a reference period?  The WMO uses 1961-1990 for long-term monitoring, or the 3 decades prior to the most recent year ending in 0 (e.g. 1991 – 2020) for short term changes. Could you please justify your choice or change to a standard ref. period.

*The reference period will be changed from 1960-2016 to 1961 to 1990, whenever possible, as suggested by the WMO. However, in some cases this is not possible due to missing data. We therefore included in the caption of Figure 2 the reference period analyzed for each ANA station whenever it is different from the standard period.*

The reference period is mentioned in the caption of Figure 2, now listed as Figure 3, and also in the Material and Methods section (Lines 196-208), where we describe the method used to produce the correlation maps.

7. Line 216: Figure 2C.

*The correct Figure will be listed in the revised text.*

The correct Figure is listed in the revised text (Line 343 and 344).

8. Line 272: typo, please correct to 'would not affect'

*Thank you for pointing out this typo. It will be corrected.*

The correction is made in Line 273.

9. The δ18O data are of different resolutions – can you please clarify how the iscam handles differently-sampled data

*The calculations made by the ISCAM (Fohlmeister, 2012) provide an interpolation of each dataset to the same resolution before merging them. Therefore we can use the original datasets containing the depths and corresponding proxy result at different resolutions in order to produce this unique record.*

We did not apply any changes to the main text, since this is a methodological aspect of the ISCAM procedure and the subject is covered by Fohlmeister (2012).

10. Line 331: please change 'first 1800 years' to 'the period spanning 1940 CE to 130 BCE' for less ambiguity.

*This part of the sentence was  replaced by "the period spanning 130 BCE to 1940 CE" in order to be consistent with always citing the oldest age first.*

The correction is made in Line 403.

More detail is needed about the C-A correction and how it was calculated (this could go in the Supplement. Could you please add the initial mean and corrected mean δ18O values for each interval to your Table S3. Something like the below?

*We use the aragonite-calcite fractionation offset described by Zhang et al. (2014) obtained for stalagmites from China. We used equation 1 below to consider the proportion between calcite and original*

*aragonite for each stalagmite interval of RN stalagmites, according to Table S3. We included the mean $\delta^{18}O$ for each interval before and after C-A correction in Table S3. Please see the Table below.*

$$\Delta^{18}O_{C-A\ corr} = \frac{sample\ calcite\ \%}{100\%\ original\ aragonite} \ x\ calcite\ fractionation\ offset$$

*Table S3 – Speleothem intervals according to texture and mineral weight proportion (wt). Texture description: A – crystals with mosaic and columnar fabrics; B – interbedded needle-like crystals. *Obtained by Utida et al. (2020). C-A: calcite-aragonite correction*

| | | | Speleothem Mineralogy | | | | |
|---|---|---|---|---|---|---|---|
| | | | | | | $\delta^{18}O$ mean (‰ VPDB) | |
| Sample | Interval (mm) | Age (yr BCE/CE) | Texture | Aragonite (wt %) | Calcite (wt %) | before C-A correction | after C-A correction |
| TRA5 | 30-54 | 1855 to 1745 CE | A | 0.0 | **100.0** | -3.50 | -2.65 |
| | 54-87 | 1745 to 1640 CE | A | 0.0 | **100.0** | -3.56 | -2.71 |
| | 87-108 | 1640 to 1565 CE | A | 0.0 | **100.0** | -3.58 | -2.73 |
| | 108-178 | 1565 to 1490 CE | A | 0.0 | **100.0** | -3.40 | -2.55 |
| TRA7* | 0-173 | 1940 CE to 130 BCE | A | 0.0 | **100.0** | -2.80 | -1.95 |
| | 173-215 | 130 to 290 BCE | B | **99.0** | 1.0 | -2.14 | -2.13 |
| | 215-270 | 290 to 3000 BCE | B | **87.1** | 12.9 | -3.12 | -3.01 |
| FN1* | 0-27 | 1790 to 1170 CE | B | **85.2** | 14.9 | -2.14 | -2.01 |
| | 27-83 | 1170 to 610 CE | B | **90.6** | 9.4 | -2.87 | -2.78 |
| | 83-128 | 610 to 80 CE | A | 0.0 | **100.0** | -1.87 | -1.03 |
| | 128-202 | 80 CE to 1730 BCE | B | **94.5** | 5.5 | -2.54 | -2.49 |
| FN2 | 6-31 | 189 to 660 BCE | B | **94.7** | 5.3 | -1.20 | -1.15 |
| | 31-56 | 660 to 960 BCE | B | **94.8** | 5.2 | -1.56 | -1.52 |
| | 56-63 | 960 to 1005 BCE | B | **94.8** | 5.2 | -2.03 | -1.99 |
| | 63-95 | 1005 to 1265 BCE | B | **93.4** | 6.6 | -1.94 | -1.88 |

The text cited above relates to C-A correction and how it was calculated. It is included in the Supplement. The new version of Table S2 is now referred to as Table S3.

11. Can you please move Figure 3 earlier in the manuscript.

*The figure will be moved to the location where it is first mentioned in the text.*

The figure is moved to the Supplement, now referred as Figure S4, according to reviewer's suggestion number 14.

12. Line 362-368: I suggest you reword this to demphasise the 4.2 ka event (which your record mostly postdates). Something like "A generally drier climate prevailed in NEB after the 4.2 ky BP (Before Present) event in the Mid-Holocene (ref). This led to the development of the Caatinga, a sparse vegetation cover which has persisted in NEB to the present (ref). These drier conditions …."

*We will reword the sentence as suggested.*

We reworded the sentence as suggested; now it is in Lines 504-507.

13. Line 368-9: it is unclear if this is statement 'more negative δ13C values in stalagmites are associated with…' refers to NEB samples or is a general statement. If general, please add impact of temperature and PCP (see Fohlmeister et al. 2020), and perhaps relocate this to the literature review.

*In this statement, the more negative δ¹³C refers to the stalagmite samples from the same caves. We modified the text to clarify this. Please see the revised sentence below.*

*"When erosion events remove most of the soil cover, there is an increase in the carbon contribution from local bedrock (mean δ¹³C of 0.5 ‰), which leads to higher δ¹³C values in the NEB stalagmites from RN. On the other hand, more negative δ¹³C values in stalagmites are associated with increased soil coverage and soil production (Utida et al., 2020)."*

The new version of this paragraph is positioned in Lines 508-513.

14. Figure 3

As for other figures, please change the colour scheme.

Please make the lines in the legend thicker so that the colours are easier to see.

Please update the 99% confidence interval to a shaded band – the two cyan lines are hard to see (assuming there are 2? In some places it seems like the black line is outside of the bounds of the 99% confidence interval? E.g. see ~1100 CE).

The U-Th data should have a label (i.e. a) to be consistent with the other data presented here.

Can this figure be combine with Figure 4? There is a lot of overlap.

15. Figure 4

As for Figure 3 re. colour palette, composite, and U-Th data.

Are the older TRA7 δ13C data needed – suggest removing them if they are not referred to in the paper.

*We have combined the answers for the above two questions and comments (14 and 15):*

*The Figures 3 and 4 were combined and the older part of TRA7 was removed from the main text, and the complete TRA7 data in the original Figure 3 was moved to the Supplement (Figure S5). We do not discuss in detail the older interval of TRA7 because it has no significant variability that is worth discussing in comparison with the other records we are presenting. The two curves representing the 99% confidence interval for the RN Composite were updated with grey color and enlarged for easier viewing. Two periods in the RN Composite age model confidence interval show a large range of variability, around 350 BCE and the base of the Composite around 1200 BCE. However, this does not affect our main interpretation. Please, see the updated version of Figure 3 below.*

The figure was adapted and moved to the Supplement. It is now referred to as Figure S4, according to reviewer's suggestion.

[Figure]

*Figure 3 – Rio Grande do Norte stalagmite isotope records and comparisons with other records from South America. A) U/Th ages from each stalagmite studied. B) Raw data of $\delta^{13}C$. c) Oxygen isotope results corrected for calcite-aragonite fractionation ($\delta^{18}O_{C-A}$), according to weight proportion of mineralogical results. D) $\delta^{18}O$ RN Composite constructed using stalagmite records from NEB (black line). Grey shaded area*

*denotes the 99% confidence interval of the age model. E) Boqueirão Lake δD record (Utida et al., 2019). F) DV2 δ¹⁸O speleothem record from Diva de Maura cave, 11outhern NEB (Novello et al., 2012). G) PAR01 and PAR03 δ¹⁸O records from Paraíso cave stalagmites, eastern Amazon (Wang et al., 2017).  H) Ti record of Cariaco Basin (Haug et al., 2001).*

Figure 3 is discussed as Figure 4 in the new version of the manuscript, starting in line 407.

Have you quantified the difference in δ13C between samples? From ~1500 CE onwards they don't appear to covary closely.

*The reviewer is correct – there are indeed some differences between the TRA7 and TRA5 $\delta^{13}C$ records that can be explained by different time resolutions between these samples. Therefore, the last 500 years were interpreted only based on TRA5. Furthermore, we did not discussed $\delta^{13}C$ during the last 500 years because the soil signal might be affected by anthropogenic impacts. Although the area above Trapiá cave probably was not occupied by settlements, the local communities have been exploring the carbonate rocks above the cave, since the exposed karst is easy to remove, and collected wood for local use, which could impact the soil $\delta^{13}C$ signal.*

No change is necessary regarding the comment above.

16. Line 417: can you please expand on why DV2 and the RN record differ? "The general trend towards more positive values" – please add over what time period this trend occurs, as I don't think it persists over the whole records.

*The text was expanded according to your suggestion. Please, see the modifications we made below.*

*"It is important to note that the RN record exhibits a climatic signal that is distinctly different from the from DV2 speleothem record from Diva de Maura Cave in S-NEB (Novello et al., 2012). Although both regions are affected by the same mesoscale atmospheric circulation, the RN site receives its precipitation directly from the ITCZ. At the S-NEB site, on the other hand the primary source of precipitation is associated with the monsoon, as it is located too far inland to be affected directly by the ITCZ. The general trend toward more positive values, as a result from insolation forcing, occurs from 150 to 1500 CE in the RN Composite, but from 600 to 1900 CE in the DV2 sample (Cruz et al., 2009; Novello et al., 2012). This trend is a result of the persistent dry conditions in the entire NEB region following the 4.2 ky BP event. However, the DV2 record does not document the same multidecadal and centennial-scale climate variability as recorded in the RN speleothem record, nor the less dry interval from 600 to 1060 CE seen in the RN Composite (Fig 3)."*

The text was expanded according to suggestions in Lines 547-569.

17. Line 421: please change 4.2 ka BP, or whatever convention you choose and be consistent throughout.

*Both mentions will be corrected to 4.2 ky BP.*

The new version only includes one instance discussing the 4.2 ky BP event, in Line 504.

18. Line 452: please explain why you think AMV and RN decoupled after ~0 CE.

*The original graph was plotted backwards in the manuscript, which affected the relationship between the AMV and RN. We corrected this error and rewrote the paragraph, now discussing the corrected relationship between the RN Composite and the AMV. In this new version of Figure 5, the decoupling between the RN Composite and the AMV reconstruction occurs between 1400 and 1500 CE. We do not have a definite answer as to why this decoupling occurs, but it might be related to differences in age models and data range. Both reconstructions come with their own sets of uncertainties that can affect the relationship. The fact that the RN Composite and the AMV reconstruction diverge most prominently during the Current*

*Warm Period might indicate that external (i.e. greenhouse gas) forcing might affect the relationship between the two records. An alternative explanation is that Pacific multidecadal variability modulated this relationship, since the state of the Pacific can affect the relationship between the AMV and Nordeste rainfall (He et al., 2021). However, while assessing these non-stationarities in the relationship is important and has to be investigated in more detail in future work, it is somewhat beyond the scope of this paper. The text will be corrected in the manuscript from lines 446 to 454. The revised Figure 5 and the revised text are shown below.*

*"There is a relationship between the $\delta^{18}O$ values in our RN speleothems and the ITCZ displacement toward the warmer hemisphere which helps explain paleoclimate variability observed in N-NEB. In order to reinforce this idea, the RN Composite was compared with Atlantic Multidecadal Variability (AMV) (Lapointe et al., 2020) (Fig 4). Some studies suggest that the warm phase of the AMV forces the mean ITCZ to shift to the north of its climatological position, causing a reduction in NEB rainfall (Knight et al., 2006, Levine et al., 2018, He et al., 2021), while a recent study suggests that warm phase AMV would cause a weakening of the ITCZ from February to July (Maksic et al., 2022). The driest periods from 750 to 500 BCE, 200 to 580 CE and 1100 to 1400 CE occurred during long periods of relatively warm AMV anomalies, considering the average temperature of 22.19°C for the period, which would force a northward ITCZ displacement or an ITCZ weakening. In both cases the result would be reduced precipitation over NEB. Although there is a decoupling between our results and the AMV between 1400 and 1500 CE, these differences might be related to age model uncertainties affecting the chronologies of the RN Composite and the AMV record. Opposite conditions between RN Composite and the AMV can also be observed during the Current Warm Period and require further investigation."*

We present an adapted version of this paragraph in Lines 584-600, following other suggestions made, especially the one related to the reorganization of the text.

19. Figure 5

I think you have accidentally plotted the Lapointe AMV backwards.

*Thank you for making us aware of this mistake in Figure 5, which has now become Figure 4. We corrected the graph and present the new version below, adapted for font size and suitable for readers with color blindness.*

[Figure]

*Figure 4 - δ18O RN Composite compared with (a) Atlantic Multidecadal Variability (Lapointe et al., 2020) and (b) Pacific and Atlantic Sea Surface Temperature gradients calculated (z-score) according to Steinman et al. (2022). Atlantic: 2σ range of 1,000 realizations of the Atlantic meridional SST gradient (north – south). Pacific: median of 1,000 realizations of the Pacific zonal SST gradient (west – east).*

Figure 4, now referred as Figure 5, has been adapted. It is presented in Line 607.

20. Line 503: please move Figure 6 up to about here.

*The figure will be moved to where it is first being discussed in the text. The Figures was updated and has now become Figure 5.*

The figure is in Line 607, where it is first being discussed in the text.

21. Line 520: please capitalise 'Indigenous'

*The word will be capitalized.*

The word is capitalized (Line 697).

22. Line 521 – "Entire Indigenous tribes died of starvation as a consequence of this drought and a related smallpox epidemic" – this suggests the smallpox outbreak was caused by the drought – is that correct? Suggest rewording to "Entire Indigenous tribes died of starvation as a consequence of this drought and a concurrent smallpox epidemic"

*Thank you for this important comment. The correction is absolutely necessary and the text will be changed to: "Entire Indigenous tribes died of starvation as a consequence of this drought and a concurrent smallpox (variola) epidemic".*

The corrected text is included in Lines 697-699.

23. Line 529: what is the age error at 1770 CE – adding the uncertainty might bolster your point that this event is the 1776-1778 drought

24. Line 535: as per above please add age uncertainty.

*For both comments 23 and 24 above, we will include a discussion about age model errors for TRA5. This is similar to the comment made by the second reviewer and we will also include a description of the U/Th ages to better explain the age results and age models. Please also see the answers we provide for Reviewer 2 for further details.*

*The errors of our age model for TRA5 are around ± 30 years (95% confidence interval) and we are thus aware that this uncertainty complicates the attribution to a single three-year long event. There exist no precipitation reconstructions or observations from this region between 1500 and 1850 CE, aside from these historical drought records. We thus consider our speleothem-based record as a first attempt to reconstruct precipitation in Northeast Brazil that would allow a comparison with historical droughts. If our speleothem records regional hydroclimate, it should retain a signal of the most intense droughts over NEB that are known to have struck the region based on the available historical literature of Brazil. The historical droughts we discuss in the paper, and we identify in our record, are the longest drought events in Northeast Brazil that occurred within the zone of influence of the ITCZ, and are thus probably the most likely to be recorded by stalagmites. Note that despite dating uncertainties of our record, the $\delta^{18}O$ peak of each drought event recorded, is consistent with the historical record of Lima and Magalhães (2018). Furthermore, the period between 1620 and 1717 CE is devoid of any abrupt drought events in the TRA5 stalagmite, which is again consistent with the historical records. Lima and Magalhães (2018) registered only 3 short drought events within this period of almost 100 years. It is also important to mention that Lima and Magalhães (2018) report all drought events in NEB and do not indicate their location. As discussed above northern and southern NEB are influenced by different climatic systems, the ITCZ and SASM, respectively, and this can explain, in part, the differences between historical and stalagmite records of Rio Grande do Norte.*

A revised text regarding the subject of this comment is included in Lines 666-673.

25. Line 544: suggest reword to "Although the TRA5 speleothem chronology precision is reduced during the last ~150 years…"

*The sentence will be changed according to the suggestion.*

The sentence has been changed according to the suggestion in Lines 717-718.

26. Figure 6: as for earlier figs, add a, b… label for U-Th data

*The figures were updated and the suggested modifications were made. Figure 6 is now Figure 5. Please, see the revised version of the Figure below.*

[Figure]

*Figure 5 – TRA5 record and equivalent historical record. (a) U/Th age is represented by black dots and horizontal lines indicate age uncertainty. (b) $\delta^{18}O_{C-A}$ record, numbers represent the peak of a drought event. A - Few drought events interval from 1620 to 1717 CE. B - 1940s to 1970s period. (c) the occurrence of historical drought years compiled from Lima and Magalhães (2018).*

The new Figure 6 is presented in Line 658.

27. Line 567: "these data suggest a trend toward increased aridity over NEB from 3000 BP to present…" Please be consistent with use of BP vs BCE. At line 495 you say the last 500 years were the wettest of the last 2 millenia, which contradicts the above statement.

*Thank you for calling attention to this erroneous statement. We reworded this sentence. It is now consistent with our interpretations. Please see the revised sentence below.*

*"The N-NEB record presents a trend toward drier conditions from 1000 BCE to 1500 CE as is also being observed in the Diva de Maura Cave in S-NEB, interpreted as an ITCZ withdrawal and SASM weakening, respectively. Although the two records are influenced by distinctly different climate systems with different precipitation seasonality, ITCZ and SASM dynamics are known to be closely linked (Vuille et al., 2012)."*

The newly revised paragraph is included in Lines 746-750.

28. Line 572: "drought period between 1500 and 1750" – Is this referring to the drought events in TRA5? The wording suggests it is linked to the RN composite, which shows abrupt change at~1500 CE to wetter conditions. Could you please clarify. Throughout, I suggest you make sure you are consistent with naming conventions between samples and between the composite record and the individual samples. Perhaps consider adding sub-headings to differentiate the longer composite record and the more recent drought record.

*In order to clarify the sentence mentioned, we will change the word "period" in line 572 to "events" ("the drought events between 1500 and 1750 CE"). We also will change the name of section 5.2 to "The TRA5 $\delta^{18}O$ stalagmite as a recorder of extreme dry events". The composite is already called RN Composite in the section Materials and Methods. A thorough review of the manuscript will be performed to clarify any other misleading nomenclature.*

The modification described above is included on Lines 754 and the Section "5.3. TRA5 $\delta^{18}O$ stalagmite and the extreme drought events" starts at Line 647.

29. The data availability statement is missing.

*The following data availability statement will be included at the end of the manuscript.*

"Data availability

The dataset generated as part of this study will be available in the PANGAEA website."

The data availability is mentioned in Lines 796-798.

30. Table S1and S2 – please use a different symbol to denote data from Cruz et al. as * is used elsewhere in the table

*The asterisk symbol will be replaced by 1 superscript in Table S1. In Table 2, the information "Data obtained by Cruz et al. (2009)" is not necessary and will be removed.*

The Tables are adapted accordantly.

31. Alves 2003 – this link is broken and I could not find the article at the website.

*The link will be corrected in the manuscript and you can also check here.*

[revised manuscript text omitted]

I have read with great interest the discussion paper by Utida et al. The authors analyze spatiotemporal ITCZ dynamics during the last three millennia in NE Brazil (NEB), and claim to relate their inferences to modern human history. The study presents partially replicated speleothem proxy records from two caves in NE Brazil, and provide an overview of past (hydro)climate trends and variability in the greater region of NEB of the southern margin of the ITCZ.

This new data set is sound and I have no doubt about the quality of the applied methods and presented data. In principle, the scientific significance is valid, since this dataset complements the northern South American speleothem record in high resolution. However, I have some concerns about the structure and clarity of the manuscript, which I feel needs some improvements before final publication.

**Main comments:**

- The structure:

I find most of the conclusions concerning ITCZ dynamics intriguing and interesting. However, I found the manuscript sometimes hard to read and some parts of the discussion are not easy to follow. For example, the section in L388ff has a rather unclear structure. The first half seems to be organized chronologically along the time of record, and describes the observed trends. But before this discussion is finished, the discussion jumps to comparing relationships between proxies, and refers to sections of the record which have not been described yet. Later on, the discussion also jumps from describing potential processes back to certain events and forth to other aspects again. I feel like the whole discussion should be carefully restructured and streamlined to build the arguments better on each other, and to provide the reader a common thread throughout the manuscript to prepare and justify your conclusions properly. I suggest to choose a consistent, logic structure, such as building up the discussion more strictly chronologically along your record, and also discuss trends first and events later separately?! Another possibility would be to bring the proxy interpretation first, and then compare to other records and discuss the forcings and consequences… There are several possibilities, but please do not mix it all up…

*Thank you for drawing attention to the structure of the manuscript. We will restructure the manuscript according to all reviewer suggestions. Most of the above comments related to structure are addressed in the comments below and we will do our best to improve the general structure of the manuscript after a final revision including all suggestions. As far as the paragraph in L38ff is concerned: we rewrote it and it is presented as part of the comments below.*

We restructured the Discussion section according to reviewer's comments in order to organize the ideas chronologically and to better support our conclusions. The section is divided into three subsections: "5.1. U/Th chronology and RN Composite", "5.2. Paleoclimate interpretation" and "5.3. TRA5 $\delta^{18}$O stalagmite and the extreme drought events". In the section about the paleoclimate interpretation, the discussion follows basically three topics: 1) hydrogeological fractionation processes that could affect the oxygen isotopes in RN and data interpretation; 2) the RN Composite compared with other records in South America and possibly ITCZ and SASM positions and 3) the RN Composite compared with possible forcing mechanisms (the whole record forced by AMV, the last 100 years forced by Atlantic and Pacific SST and the mechanisms that could explain the zonal ITCZ behavior).

I also strongly suggest to put special effort in elaborating how the two parts of the discussion (i.e., the paleo-record description, and the discussion of historical droughts) actually build on each other, and better justify why both aspects need to be discussed in one paper. In the current version of the manuscript these appear more as two separated stories.

*Thank you for drawing attention to the connection between these paleo-records and historical records. The paleo-precipitation record from Northeast Brazil is important to understand the modern climate and to put it in a long-term historical context. Hence, there is really no separation between the two,*

*just a continuing precipitation history over time, indeed. No observed or reconstructed precipitation record exists for the period prior to and 1850 CE in NE Brazil. The only available information is the historical record of droughts. Hence the speleothem record allows us to put these droughts in a longer-term context and provide a broader spatiotemporal assessment. As far as the possibility to discuss the historical droughts in another paper is concerned, we believe that we still lack sufficient data for a second paper. The current analysis should really be viewed as a first attempt to compare paleo-precipitation and historical records in this region.*

We restructured the first paragraph of the Section about the last 500 years to include the TRA5 results in the main story. In the second paragraph we discuss the limitations of the TRA5 age model and justify our approach based on evidence from historical records.

- U/Th Results description

I miss a proper description of the U/Th results in the main text. This should e.g., comprise U and Th concentrations, uncertainties, Th contamination, description of inversions, etc, … (check Dutton et al. 2017 as a guideline to report U-series data). This is also important due to the presence of both calcite and aragonite, where we would expect an influence on the ages if recrystallization occurred! In addition, a statement concerning the final uncertainties of the age-depth model is essential, also regarding the several outliers. This is particularly relevant when reporting absolute ages for extreme events! From the so presented age models, it is not at all clear if the dating supports an annual precision of a single drought event, or the unequivocal allocation to an event reported in the historic record.

*The description of the methods and U/Th results has been revised and will be included in the manuscript according to the text below. The methods were revised to be in accordance with Dutton et al. (2017) suggestions for U/Th series publications.*

*Section 3: Materials and Methods*

*Chronological studies on speleothems were based on U-Th geochronology performed at the Laboratories of the Department of Earth and Environmental Sciences, College of Science and Engineering, University of Minnesota (USA), and at the Isotope Laboratory of the Institute of Global Environmental Change, Xi'an Jiaotong University (China), according to Cheng et al. (2013). Subsamples of ~100 mg were obtained in clear layers, close to the growth axis trying to keep a maximum thickness of 1.5 mm, 10 mm wide and no more than 3 mm depth. The powder samples were dissolved in 14 N $HNO_3$ and spiked with a mixed solution of known $^{233}U$ (0.78646 ± 0.0002 pmol/g) and $^{229}Th$ (0.21686 ± 0.0001 pmol/g) concentration. Th and U were co-precipitated with FeCl and separated with Spectra/Gel® Ion Exchange 1x8 resin column with 6N HCl and super clear water, respectively. Th and U were counted in an inductively coupled plasma-mass spectrometer (MC-ICP-MS Thermo-Finnigan NEPTUNE PLUS) and the results were calculated in a standard spreadsheet based on Edwards et al. (1987) and Richards and Dorale (2003) using the isotopic ratios measured, machine parameters and corrections factors to eliminate effects of contamination by detrital Th to finally obtain the age of each sample. The decay constants used are: $\lambda_{238}$ 1.55125 x $10^{-10}$ (Jaffey et al., 1971), $\lambda_{234}$ 2.82206 x $10^{-6}$ and $\lambda_{230}$ = 9.1705 x $10^{-6}$ (Cheng et al., 2013). Corrected $^{230}Th$ ages assume the initial $^{230}Th/^{232}Th$ atomic ratio of 4.4 ± 2.2 x $10^{-6}$. Those are the values for a material at secular equilibrium, with the bulk earth $^{232}Th/^{238}U$ value of 3.8 (McDonough and Sun, 1995). The ages are reported in BP (Before Present defined as the year 1950 A.D.) and converted to Common Years (CE). Age uncertainties are 2 σ.*

*Results and discussions*

*The results and discussions below regarding $^{232}Th$ contamination and calcite x aragonite crystallization will be included in the appropriate section of the manuscript, according to the reviewer's suggestion.*

*The high values of $^{232}Th$ and low $^{230}Th/^{232}Th$ ratio suggest incorporation of detrital Th transported by the seepage solution to the speleothems, which lead to a higher uncertainty of the age values.*

*Recrystallization of aragonite into calcite might also reduce the U content and given older age for carbonates (Lachniet et al., 2012). These are the main reasons for age inversions along speleothems from Northeast Brazil. Therefore, we analyzed a large number of U/Th ages to improve the age model and reduce the errors associated with detrital Th and recrystallization.*

*FN1 is partially composed of calcite between the depths of 83 and 128 mm (Table S3), and top and base are composed of aragonite. Overall this stalagmite presents low U concentration and high $^{232}$Th amounts. We considered the association of low $^{230}$Th/$^{232}$Th and low U content the most important factor affecting the age errors and inversions in the FN1 stalagmite. In contrast the FN2 stalagmite has a more precise chronology due to the predominant aragonite composition, with high $^{238}$U content and higher $^{230}$Th/$^{232}$Th ratio than FN1. The ages from the FN1 stalagmite are all in chronological order and contain low errors and were therefore all kept in the age model.*

*The TRA5 stalagmite is entirely composed of calcite, but the $^{238}$U content is relatively high compared to other stalagmites, which improves the confidence in its age results. However, the high $^{232}$Th content of samples from the top of TRA5 affects the age results over the last 200 years. The other two inversions in TRA5 (71 and 104 mm, Table S2) might also be a result of $^{232}$Th contamination resulting in increased errors.*

*Most of the TRA7 stalagmite used in our composite is composed of calcite (from top to 130 BCE). According to age results produced by Utida et al. (2020), most of the ages are in chronological order and the inversions seem to not have a direct relationship with $^{238}$U, and the high $^{232}$Th content is similar to other ages from TRA7.*

*The age uncertainties caused by high $^{232}$Th concentration and calcite recrystallization in stalagmites might affect the age model. However the strong coherence between the $\delta^{18}O$ curves from different stalagmites argues in favor of the good quality of our chronology. This is evident when FN2, which is composed 100% of aragonite, is compared with other samples. There is a different amplitude range in its $\delta^{18}O$ values, but when the curve is superposed on other $\delta^{18}O$ records the variability is similar. This amplitude range is corrected when the $\delta^{18}O$ results are submitted to the ISCAM composite construction, since it normalizes the results.*

*Historical records and age model uncertainties*

*The errors of our age model for TRA5 are around ± 30 years (95% confidence interval) and we are thus aware that this uncertainty complicates the attribution to a single three-year long event. There exist no precipitation reconstructions or observations from this region between 1500 and 1850 CE, aside from these historical drought records. We thus consider our speleothem-based record as a first attempt to reconstruct precipitation in Northeast Brazil that would allow a comparison with historical droughts. If our speleothem records regional hydroclimate, it should retain a signal of the most intense droughts over NEB that are known to have struck the region based on the available historical literature of Brazil. The historical droughts we discuss in the paper, and we identify in our record, are the longest drought events in Northeast Brazil that occurred within the zone of influence of the ITCZ, and are thus probably the most likely to be recorded by stalagmites. Note that despite dating uncertainties of our record, the $\delta^{18}O$ peak of each drought event recorded, is consistent with the historical record of Lima and Magalhães (2018). Furthermore, the period between 1620 and 1717 CE is devoid of any abrupt drought events in the TRA5 stalagmite, which is again consistent with the historical records. Lima and Magalhães (2018) registered only 3 short drought events within this period of almost 100 years. It is also important to mention that Lima and Magalhães (2018) report all drought events in NEB and do not indicate their location. As discussed above northern and southern NEB are influenced by different climatic systems, the ITCZ and SASM, respectively, and this can explain, in part, the differences between historical and stalagmite records of Rio Grande do Norte.*

A more detailed explanation about the U/Th method is included in the Material and Methods section (Lines 213-235), as well as the U/Th description in the Results section (Lines 368-383). Furthermore, in the Discussion section we include a comprehensive discussion regarding $^{232}$Th contamination and calcite x aragonite crystallization (Lines 439-481).

I have some more general comments to the style of the writing and presentation, which I summarize here. Please find specific locations related to the following points in my minor comments along the text:

- Across the manuscript I found repetitive statements, but also rather irrelevant information. This makes the reader lose focus, so I suggest to try to shorten/streamline the text in general.

- In many figures, some aspects are hardly visible. Please improve accessibility, e.g., text sizes, increase size of markers of locations, use colors that are better visible.

- Sometimes past and present tense is mixed, please check language style.

*We are in the process of performing a complete revision of the text in order to improve the language quality and the conciseness of the text. The figures are updated and can be seen below. They were updated for text and markers sizes, as well as adapted for color blind readers. Certainly, the points mentioned will help us produce a higher quality manuscript.*

Although the text is not shorter, because of some additional sections included, we restructured the text to be more concise. All figures are updated for text and markers sizes, as well as adapted for color blind readers.

**Minor comments:**

L49 weakening

*The word spelling will be corrected.*

The word spelling has been corrected (Line 50).

L62-63: Is there a reference for this statement?

*The references are the same as those mentioned after this statement. We will add the references Marengo and Bernasconi (2015) and Lima and Magalhães (2018) to this sentence.*

The reference is included in Lines 62-63.

L91: I think the Lechleitner Paper is from 2017.

*The year of publication will be corrected to 2017.*

The reference has been corrected.

L129-131 is this relevant?

*We believe so. But we combined the two sentences into one, and we clarified the meaning of the text.*

*"The caves were developed in the Cretaceous carbonate rocks of the Jandaíra Formation, Potiguar Basin, close to the Apodi River valley in a region of exposed karst pavements (Pessoa-Neto, 2003; Melo et al., 2016; Silva et al., 2017)."*

*The sentence is presented in Lines 151-154.*

L138: Any idea why the cave temperature is considerably lower than the annual mean temperature? Is this relevant for your data?

*The annual mean temperature was taken from a climate station in the city, kilometers from the cave. Temperatures in cities tend to be higher than in pristine environments (urban heat-island effect) such*

*as those where the caves are located. This information is not directly relevant for the interpretation of our results, but nonetheless helpful for those who want to better understand the climatology of the region.*

No change is necessary regarding the comment above.

L148: I feel like most of this section is rather results than material/methods description?

*We agree with the reviewer that this part of the text is better suited in the results than the methods or Regional settings sections. We will adjust this section accordingly.*

We moved the ANA and GNIP data to a brief paragraph in the Material section (Lines 190-212) and the text about correlation maps and precipitation to the Results section named "4.1. Modern climatology and δ$^{18}$O rainfall distribution" (Lines 302-366).

L149ff: There is a lot of discussion of the different sectors within NEB, it may be helpful for the discussion and the readers to indicate those in a figure?

*The spatial correlations of Figure 2 are used to define the northern and southern NEB climatologically. The new version of this Figure includes the labels "Northern NEB" and "Southern NEB" in graphs a) and b). Please, see the revised version of Figure 2.*

The new figure is included in the text in Line 351 and referred as Figure 3.

L164: How is "most significant" defined?

*The "most significant" years of El Niño in NEB are those that most drastically impacted the precipitation amount. We changed the text to clarify the statement. Please see the revised text below.*

*"…we excluded the 39 El Niño - Southern Oscillation (ENSO) years that most drastically changed the precipitation amount in NEB, following the methodology of Araújo et al. (2013)."*

The revised text is included in Lines 193-195.

L174: "is primarily the result of a shorter rainy season". This is not quite what is described above. There you write that the rainy season has the same length but is weaker?

L175: "The anomalous length…" See previous comment, according to your own results, this is only for the wetter years.

*We rewrote the paragraph mentioned in comments L174 and L175 to clarify these aspects. Please see below.*

*"The results (Fig. S1) reveal that in the majority of years (normal years - interquartile range) the rainy season persists from February to April, with precipitation varying from 100 to 180 mm/month, and minor contributions occurring in January and May (50-70 mm/month). During the drier years (lower quartile), February has a reduced precipitation amount, similar to the amount in January during normal years, as described above. The maximum precipitation of 90 mm/month occurs between March and April. For wetter years (upper quartile), the rainy season starts in January with more than 100 mm/month and lasts until May with almost 150 mm/month, reaching values higher than 250 mm around March. These data show that wetter years are characterized by increased precipitation amounts and a longer rainy season starting in January and ending in May, while the precipitation deficit during drought years is a result of decreased precipitation amount and a shorter rainy season, with a peak in precipitation between March and April. The anomalous length of the rainy season during dry and wet years is attributed to variations in the meridional SST gradient in the tropical Atlantic that results in a shift of the ITCZ to the north or south of its climatological position (e.g., Andreoli et al., 2011; Marengo and Bernasconi, 2015; Alvalá et al., 2019)."*

The revised text is included in Lines 303-318.

L189ff: If this is relevant for the discussion later, I feel like the authors should clearly define the difference between ITCZ related rainfall in NEB, and SASM related rainfall in S-NEB. Some reader may not be able to recall the exact difference at once…

*As mentioned in comment L149ff, the spatial correlations of Figure 2 are used to define the northern and southern NEB climatologically. The new version of this figure indicates the "Northern NEB" and "Southern NEB" in order to call attention to the differences. Please see the new version of Figure 2.*

The new figure is included in the manuscript in Line 351 and referred to as Figure 3.

L229-230: This information is not relevant for this study.

*The information will be removed.*

The information has been removed.

L282 to significantly reduce

*The error will be corrected.*

The error has been corrected.

L327: Avoid repetition of the method description.

*We will remove the sentence of lines 327-329.*

The information has been removed.

L343: Why is a composite record only constructed for $\delta^{18}$O? what about the early phase 2-3k BCE? It is shown in the figure, but I didn't find a statement why it is shown but not discussed

*$\delta^{18}$O is a proxy for regionally integrated climate and circulation upstream, with the main signal related to atmospheric processes, while $\delta^{13}$C values are more diverse and more site-specific, related to the seepage solution pathway and spots of vegetation above the cave. Therefore, a composite for the $\delta^{13}$C will give a more heterogenic mosaic that may not be related with the regional conditions. For the early part of our record, we therefore decided to remove it from the main text. As it is not being discussed, as suggested by the first reviewer, we will move Figure 3, with this early part of our record, to the Supplemental Material as Figure S5.*

No change is necessary regarding the comment above.

L389: if the age axis is correct, the oldest period of the RN composite is around 1.5 to 1k BCE?

L390: I see rather positive anomalies between 1 and 0.5 BCE…

L391ff: Confusing section, please clarify. In the previous sentence you state, there is soil erosion, here you state that did not contribute much… Now what?

L397: more negative as compared to what? to me, the $\delta^{18}$O values are rather higher than in the earlier part of the record… please clarify.

L399ff: This paragraph is hard to follow. Please don't start compare/discussing sections of the record that haven't been mentioned before… (here the LIA suddenly pops up)

*Thank you for calling attention to this paragraph. We combine our reply to comments L389 - L397 below. We rewrote the paragraph as shown below, paying attention to all suggestions made by the reviewer.*

*"The oldest period covered by the RN Composite, from 1200 to 500 BCE, is characterized by successive dry and wet multidecadal periods, with increased precipitation in N-NEB from 1060 to 750 BCE and from 460 to 290 BCE, as suggested by the negative departures seen in the $\delta^{18}O$ values. During this last period, there is also a tendency from lower to higher $\delta^{13}C$ values, suggesting progressive surface soil erosion related to rainfall variability (Fig. 4), as interpreted by Utida et al. (2020). This period ends up in a stable interval from 300 BCE to 0 CE with $\delta^{13}C$ values close to the bedrock signature at about -1‰ to +1‰, indicating a lack of soil above the cave. After an abrupt reduction of $\delta^{13}C$, the values decrease to approximately -2‰ between 200 CE and 1500 CE. From 1500 CE to the present, negative values of $\delta^{13}C$ is responding to wet climatic conditions as indicated by lower $\delta^{18}O$ values. The more negative $\delta^{13}C$ during this period can be related to denser vegetation that favored both soil production and stability above the cave."*

The revised text is included in Lines 518-536.

L408: On the millennial scale, yes… since you also mention shorter timescales earlier, I would clarify this here…

*We adapted the text to clarify this statement. Please see below.*

*"During the last 2500 years, the RN Composite shows similar characteristics as the lower-resolution $\delta D$ lipids record obtained in Boqueirão Lake sediments"*

The revised text is included in Lines 537-539.

L413: Unclear, why is this?

L414 do you mean "latter"?

L415: Very vague statement, please specify. Also, how well are the lake sediments dated and is that comparable to your chronology?

*The lake sediments were dated with the 14C method, which has larger errors than the U/Th method used for stalagmites. Furthermore, the age model of Boqueirão Lake was constructed with fewer ages compared to stalagmite chronology. We rewrote the sentence to simplify it and answer the comments from L413 to L415. Please, see below.*

*"This inconsistency might be related to different chronological controls between lake and stalagmite records and possibly also by the location of Boqueirão Lake that is more strongly affected by the ITCZ and as it is located in the eastern coastal sector of NEB (Zular et al., 2018; Utida et al., 2019)."*

The revised text is included in Lines 543-546.

L420 Maybe indicate the insolation curve in the Figure?

*We have included the insolation curve in Figure 3 and 4 of the manuscript. This Figure in question already contains a lot of information and adding even more would make it difficult to read. We will instead add the insolation curve to the Figure S5 in the Supplement as shown below.*

[Figure]

*Figure S5 – Rio Grande do Norte stalagmite isotope record. (a) U/Th ages for RN stalagmites. (b) Raw data of $\delta^{13}C$. (c) Oxygen isotope results corrected for calcite-aragonite fractionation ($\delta^{18}O_{C-A}$), according to weight proportion of mineralogical results. (d) $\delta^{18}O$ RN Composite constructed using stalagmite records from NEB (black line). Grey lines denote the age model confidence interval of 99%. (e) February insolation curve at 10°S.*

The insolation curve is included in Figure S4 in the Supplement (Line 24).

L421 persistently (?)

*The word will be corrected.*

The word spelling has been corrected (Line 556).

L426ff: This is an interesting conclusion which is however barely discussed beforehand. The discussion here rather ends quite abruptly. I feel this could be more elaborated, because it seems to relate to the statement in the abstract, that you can make inferences on spatio-temporal ITCZ variability?

*We rewrote the whole paragraph to call attention to the differences between N- and S-NEB and reinforce our conclusions. Please, check it below.*

*"It is important to note that the RN record exhibits a climatic signal that is distinctly different from the DV2 speleothem record from Diva de Maura Cave in S-NEB (Novello et al., 2012). The general trend toward more positive values, as a result of insolation forcing, occurs from 150 to 1500 CE in the RN Composite, but from 600 to 1900 CE in the DV2 sample (Cruz et al., 2009; Novello et al., 2012). This trend is a result of the persistent dry conditions in the entire NEB region following the 4.2 ky BP event. However, the DV2 record does not document the same multidecadal and centennial-scale climate variability as recorded in the RN speleothem record, nor the less dry interval from 600 to 1060 CE seen in the RN Composite (Fig 3). As demonstrated by the spatial correlation maps between $\delta^{18}O$ values and regional precipitation (Fig. 2),*

*the S-NEB and N-NEB regions are influenced by distinct rainfall regimes whose peaks of precipitation arise during the summer monsoon season and the autumn ITCZ, respectively. Our data provide evidence for a spatial and temporal distinction of NEB climate patterns in the past that can be interpreted as differences in seasonality during the last millennia. Furthermore, contemporaneous dry or wet events in both N-NEB and S-NEB suggest the occurrence of larger regional climate changes with higher environmental impacts."*

The text has been expanded according to the suggestion and is included in Lines 547-569.

L432: very vague and unclear which characteristics are meant

L429ff (the whole section) difficult to follow here, you jump from describing a trend to single events, and then to processes again - not clear where this leads to? please provide the reader with some kind of guidance in between, maybe in form of a summary and/or statement which observation will be tested/explained now…

L437: unclear what information your record adds to this aspect, and how this relates to the discussion?

*We rewrote the paragraph to adjust it according to the reviewer's suggestions. Please see revised version below.*

*"When comparing N-NEB and eastern Amazon conditions, it is evident that the RN Composite shares some similarities with the Paraiso stalagmite record (Wang et al., 2017), due to the contribution of ITCZ precipitation in both places. But there are also important differences (Fig 4). The RN Composite shows lower δ18O values between 500 and 1000 CE, compared to the earlier period, while Paraiso shows decreasing values around the same period, suggesting a slight increase in precipitation in both areas. From 1160 to 1500 CE, abrupt increases in $\delta^{18}O$ values are seen in both records, which indicates abrupt and prolonged drought conditions due to a northward ITCZ migration. However, around 1100 CE, and the period from 1500 to 1750 CE, Paraiso is antiphased with the RN Composite and in phase with the Cariaco Basin (Haug et al., 2001), which is inconsistent with the notion of an ITCZ-induced regional precipitation change. Instead, a zonally-oriented precipitation change within the ITCZ domain over Brazil is required to explain the anti-phased behavior between precipitation in N-NEB and the eastern Amazon, and similarities between Cariaco and the eastern Amazon."*

The text has been expanded according to the suggestion and is included in Lines 570-583.

L441: now the discussion jumps again back in time to another event… I would bring this example later to showcase a potential relationship to Atlantic temperatures…?

*We will adjust the entire manuscript according to suggestions and comments of the reviewers. Certainly the discussion about Bond events can be combined with the Atlantic temperature discussion and its relationship with the ITCZ. It will become clear where the best position for this paragraph is once we finalize the revision of the manuscript.*

The Discussion section was restructured. The Bond event is now mentioned right after the discussion of North Atlantic variability (Lines 600-606), which makes the explanation more coherent.

L447: I suggest to turn the argument around - the idea is that ITCZ displacements are forced by temperatures, so we check if there is a relationship of our record to AMV?

L448ff This sentence seems incomplete

*We rewrote this paragraph in order to clarify our ITCZ displacement hypothesis related to meridional temperature gradients in the Atlantic. Please see the revised text below.*

*"We investigate the potential relationship between $\delta^{18}O$ values in our RN speleothems and an ITCZ displacement toward the warmer hemisphere to explain paleoclimate variability observed in N-NEB. In order to test this hypothesis, the RN Composite was compared with a reconstruction of Atlantic Multidecadal*

*Variability (AMV) (Lapointe et al., 2020) (Fig. 4). Some studies suggest that the warm phase of the AMV forces the mean ITCZ to shift to the north of its climatological position, thereby causing a reduction in NEB rainfall (Knight et al., 2006, Levine et al., 2018), while a recent study suggests that the warm phase of the AMV would cause a weakening of the ITCZ from February to July (Maksic et al., 2022)."*

The text has been revised and is now presented in Lines 584-592.

L452: Have you checked other records of AMV / Atlantic SSTs that allow to check if the Lapointe record is representative for the entire basin during these times or not?

*Lapointe et al. (2020) present a record that is in good agreement with other temperature records from the North Atlantic and with other AMV reconstructions (Mann et al., 2009; Cunningham et al., 2013; Miettinen et al., 2015; Reynolds et al., 2016; Wang, et al., 2017; Spooner et al., 2020) and also with records from the Cariaco Basin (Black et al., 1999), which suggest that their AMV reconstruction is reliable and indicative of a large-scale teleconnection with the tropics.*

No change is necessary regarding the comment above.

L462ff As far as I understand the plot, there is no PDV record in the plot, so how do you infer a cold phase of PDV during that time? I guess you refer rater to Fig 5, but still I suggest to explain which record / curve you are referring to here exactly and what they are showing? Is the Pacific SST gradient a measure of PDV? this curve shows centennial scale variability, but not at all decadal?

*The Figure presents only the AMV. The discussion about the relationship between AMV and PDV was only based on an observed precipitation analysis. We made some adjustments in the paragraph to clarify this aspect. Please see the revised text below.*

*"According to Kayano et al. (2020, 2022), during the last century, dry conditions over N-NEB and the eastern Amazon are present when AMV and Pacific Decadal Variability (PDV) are in both in their warm phase, or when the AMV is in a cold phase and the PDV in its warm phase. On the other hand, when AMV and PDV are both in their cold phase, precipitation over the Amazon is anti-phased with NEB, resulting in decreased precipitation over the Amazon and increased precipitation over NEB. This zonally aligned precipitation signal over eastern tropical S. America is the result of joint perturbations of both the regional Walker and Hadley Cell's, produced by teleconnection between the Atlantic and Pacific (Kayano et al., 2022, He et al., 2021). These conditions can explain in part our results, however during the decoupling of our record with AMV (between 1500 and 1750 CE), increasing precipitation over N-NEB and decreasing precipitation over the eastern Amazon can be better explained by the positive gradients both in Atlantic and Pacific Oceans forcing a south ITCZ migration (Fig. 4)."*

The text has been revised and is presented in Lines 634-645.

L494ff: what does the $\delta^{13}$C record tell in this time? extreme events could be also visible there, the record looks quite "spikey"

*The $\delta^{13}$C and $\delta^{18}$O records of TRA5 show similar characteristics during this time. These data can be interpreted in the same way as the rest of our record, indicating increased (decreased) precipitation ($\delta^{18}$O) and soil production (erosion), combined with a decrease (increase) in Prior Calcite Precipitation (PCP) at the epikarst ($\delta^{13}$C). All mentioned processes drive oxygen and carbon isotope variability in the same direction (Novello et al., 2021).*

We did not include any changes related to this comment, as outlined in the answer 15 for RC1: "… *we did not discuss $\delta^{13}$C during the last 500 years because the soil signal might be affected by anthropogenic impacts. Although the area above Trapiá cave probably was not occupied by settlements, the local communities have been exploring the carbonate rocks above the cave, since the exposed karst is easy to remove, and collected wood for local use, which could impact the soil $\delta^{13}$C signal.".*

L495: How do the other speleothem records compare during this time? I understand that they have lower resolutions, but to support your point and strengthen your arguments (e.g. concerning age model uncertainties, etc) a zoom into the comparison of the different proxy records might be helpful? Also, how would ISCAM move the TRA7 record with respect to the original agemodel? This gives also a hint regarding dating uncertainty…

*Unfortunately, the TRA7 record is not suitable for this kind of comparison because of its lower resolution.  The deposition rates (DR) of TRA7 and TRA5 are different, 0.18mm/yr and 0.33 mm/yr. We tried to sample TRA7 at the maximum possible resolution to achieve such a comparison. Considering the uncertainties of the age model, some peaks are too smoothed and the $\delta^{18}O$ data are not suitable for comparison. We believe that including such a comparison would not aid in our interpretation. The TRA7 ISCAM age model does not significantly change compared to the COPRA model, since both use the linear method. We did not plot them together in the Figure S4 because the superposition would not be visible. We show here a figure to demonstrate the similarity of both TRA7 age models and to clarify this question.*

[Figure]

The first paragraph of the topic "5.3. TRA5 $\delta^{18}O$ stalagmite and the extreme drought events" (Lines 648-656) gives an explanation as to why we use only TRA5 to interpret the last 500 years.

L497ff: statements like this require a proper report of dating and age model uncertainties. From visual inspection there are some ages which have quite high uncertainties, which could limit the fidelity of such a record to absolutely date extreme events with annual precision! It could be also short-term hiatuses, that last longer than a single year…? I understand that the TRA7 age model is part of another paper, but then please still give a statement here, because this is relevant for your conclusions. It is also not clearly visible from the plot of the age model in the supplement.

*We will improve the methods, results and discussion sections when referring to the stalagmite ages. We also improved the figures in the supplemental material to better describe the age models and uncertainties of our record and we modified the text in this paragraph. Please see revised text below.*

*"In NEB, the low water availability has been one of the major challenges faced by its people during the last centuries (Marengo and Bernasconi, 2015; Marengo et al., 2021; Lima and Magalhães, 2018). On the other hand, the last 500 years were the wettest of the last two millennia, according to our results (Fig. 3). Superimposed on these long-term negative $\delta^{18}O$ anomalies, distinct peaks are recorded in the TRA5 $\delta^{18}O$ record from 1500 to 1850 CE (Fig 5). These drought events are visible in this record thanks to its higher deposition rate (faster growth) and thus higher temporal resolution of the $\delta^{18}O$ record when compared to other stalagmites used in our study. Although the age model errors of TRA5 are larger and could limit our ability to attribute $\delta^{18}O$ peaks to specific single-year events, it still allows for a comparison between these abrupt events with historical records to demonstrate the long-term context of abrupt drought events in modern human history."*

A more detailed explanation about the U/Th method is included in the Material and Methods section (Lines 213-242), as well as in the U/Th description of the Results section (Lines 368-383).

Furthermore, the Discussion section contains a comprehensive analysis of $^{232}$Th contamination and calcite x aragonite crystallization (Lines 438-481). We also justify the use of TRA5 in the first and second paragraphs of the topic "5.3. TRA5 $\delta^{18}$O stalagmite and the extreme drought events" (Lines 648-673), explaining why we use only TRA5 to interpret the last 500 years and what the limitations of this approach are.

L538ff: how many droughts are not recorded in your stalagmite record? the reference is not accessible, so please provide a clear statement, or, better, a plot/histogram of all droughts reported by the other study in Fig 6

*The historical record of Lima and Magalhães (2018) (Graph 1 in the original paper) mentions drought events compiled from different historical letters and books from all of Northeast Brazil (NEB). Hence, some of these events might be located in the southern and/or northern part of Northeast Brazil. Our data record a smaller number of events than are listed in the historical data, probably recording primarily the most intense events that affected all of NEB, or ITCZ changes that affected only the northern portion of NEB. According to our correlation maps, southern and northern NEB have different precipitation sources and seasonality. Therefore, the TRA5 stalagmites do not record all events mentioned in the compilation of Lima and Magalhães (2018).*

*The paper can be accessed using the link below. The link will be updated in the reference section of the revised manuscript.*

*https://seer.cgee.org.br/parcerias_estrategicas/article/view/896/814*

[Figure]

[Figure]

*Graph 1 – Historical drought events in Northeast Brazil. Extracted from Lima and Magalhães (2018).*

We explain in section "5.3. TRA5 $\delta^{18}$O stalagmite and the extreme drought events" (Lines 725-734) why TRA5 does not record all drought events mentioned by historical records .

L551: Discussion ends quite abruptly, following from your section 5.1 one would at least expect a hypothesis of a forcing mechanism of the drought occurrence?

*We will include a concluding paragraph at the end of section 5.1 suggesting a hypothesis related to our main conclusion. Please see the revised text below.*

*"We suggest that progressive changes in the mean ITCZ position over the course of the last 500 years might be responsible for historical droughts that affected the seasonality of N-NEB and caused abrupt*

*and strong drought events. No preferred periodicity of these events is apparent in our record. Additional drought-sensitive high-resolution records will be required to improve our understanding of these historical droughts events in NEB."*

The text has been revised and it is now presented in Lines 734-738.

How is the drought frequency related to what you found out from your record of the past 2.5ka? I suggest to elaborate this a little bit more…

*Our stalagmite and RN Composite records contain variability at multidecadal and interdecadal frequencies. However, the wavelet analysis did not show a temporally continuous signal at a preferred wavelength. We therefore chose not to discuss this aspect in greater detail.*

No change is necessary regarding the comment above.

**Figures**

Figure 1: Locations hardly visible, please increase the size of the text and the stars. Also No. 5 is barely visible, please choose other colors.

*We updated all figures to improve the font size of text and symbols. Please see the revised Figures 1 and 5 below. Figure 5 was updated and changed to Figure 4.*

[Figure]

*Figure 1 – Location and precipitation climatology of study sites during the austral summer (DJF - December to February) and autumn (MAM - March to May). Color shading indicates percentage of the annual precipitation total that is received during either DJF or MAM and highlights the extent of (a) the SASM over the continent and (b) the ITCZ over the ocean. Precipitation data is from the Global Precipitation Measurement (GPM) mission, with averages calculated over the period 2001–2020. 1) Trapiá and Furna Nova Cave (this study), 2) Boqueirão Lake (Utida et al., 2019), 3) Diva de Maura Cave (Novello et al., 2012), 4) Paraíso Cave (Wang et al., 2017), 5) Cariaco Basin (Haug et al., 2001). GNIP stations: A) Fortaleza, B) Brasília, C) Manaus.*

The new version of Figure 1 is in Line 137.

[Figure]

*Figure 4 - δ18O RN Composite compared with (a) Atlantic Multidecadal Variability (Lapointe et al., 2020) and (b) Pacific and Atlantic Sea Surface Temperature gradients calculated (z-score) according to Steinman et al. (2022). Atlantic: 2σ range of 1,000 realizations of the Atlantic meridional SST gradient (north − south). Pacific: median of 1,000 realizations of the Pacific zonal SST gradient (west − east).*

Figure 4, now referred to as Figure 5, was revised and is now presented in Line 607.

Figure 2: Increase symbols for locations. Please improve visibility in general. Caption should be streamlined, "precipitation amount" is mentioned twice in the first sentence (L199-201). Correlation maps is repeated in L199 and L204. GNIP is repeated in L200 and L207. No need to repeat all information to all caves again, it is also ok to refer to the previous figure…

*We updated all figures for size and to render them suitable for color-blind readers. Please see the revised Figure 2 and caption below.*

[Figure]

*Figure 2 – Monthly mean observed precipitation amount collected at ANA and δ¹⁸O values for GNIP stations (IAEA-WMO, 2021) (black dots) and correlation maps between gridded precipitation and δ¹⁸O anomalies from the same stations (black dots) for: (a) Northern NEB, Fortaleza and Pedra das Abelhas stations (star 1), (b) Southern NEB, Brasília and Andaraí stations (star 3), c) Eastern Amazon, Manaus and Belterra stations (star 4). The maps show the spatial correlation between δ¹⁸O anomalies at GNIP stations and GPCC gridded precipitation anomalies based on the period 1961-1990 for December to February (DJF) and March to May (MAM) for Fortaleza, Brasília and Manaus stations (Ziese et al., 2018). The δ¹⁸O values (left y axis) and precipitation (right y axis) for each station were obtained from the GNIP IAEA/WMO database. Stars indicate the site locations: 1) Trapiá Cave, Furna Nova Cave and Pedra das Abelhas ANA Station (reference period 1910-2019), 2) Boqueirão Lake (Utida et al., 2019), 3) Diva de Maura Cave (Novello et al., 2012) and Andaraí ANA Station (reference period 1960-1986), 4) Paraíso Cave (Wang et al., 2017) and Belterra ANA Station (reference period 1975-2007), 5) Cariaco Basin (Haug et al., 2001).*

Figure 2 has been changed as discussed in the response to both reviewers. The new version is now referred to as Figure 3 (Line 351).

Figure 3: Please check if colors are color-blind friendly (red and green mixed…?)

*We updated all figures to render them suitable for color blind people and we checked them using the website that simulates color blindness, as suggested by the journal. Please see the updated version of Figure 3 in this comment. Please note that we merged Figure 3 and 4 because of the overlapping data. The complete TRA7 record is in the supplemental material. Please see Figure S7 below.*

[Figure]

*Figure 3 – Rio Grande do Norte stalagmite isotope records and comparisons with other records from South America. a) U/Th ages from each stalagmite studied. b) Raw data of $\delta^{13}C$. c) Oxygen isotope results corrected for calcite-aragonite fractionation ($\delta^{18}O_{C-A}$), according to weight proportion of mineralogical results. d) $\delta^{18}O$ RN Composite constructed using stalagmite records from NEB (black line). Grey shaded area denotes 99% confidence interval of age model. e) Boqueirão Lake $\delta D$ record (Utida et al., 2019). f) DV2 $\delta^{18}O$*

*speleothem record from Diva de Maura cave, southern NEB (Novello et al., 2012). g) PAR01 and PAR03 $\delta^{18}O$ records from Paraíso cave stalagmites, eastern Amazon (Wang et al., 2017).  h) Ti record of Cariaco Basin (Haug et al., 2001).*

Figure 3 is referred to as Figure 4 in the new version of the manuscript in Line 407.

[Figure]

*Figure S7 – Oxygen isotope records and age model results calculated by ISCAM for individual stalagmites and Composite. The normalization of the data is made by ISCAM (Fohlmeister, 2012).*

Figure S6 – Oxygen isotope and age model results calculated by ISCAM for stalagmites and composite. The normalization of the data is performed by ISCAM (Fohlmeister, 2012).

The figure above is included in the Supplement as Figure S6.

Also, why is the early phase of TRA7 between 3 and 2k not included in the composite?

*This part of the TRA7 stalagmite was not included in the ISCAM composite, because this interval was not the focus of our discussion. Even though it is new data, most of its interpretation is related to the 4.2 ky BP event and was described previously by Cruz et al. (2009) and Utida et al. (2020). Therefore, and following the suggestion of Reviewer 1, the Figure 3 was merged with Figure 4, and this older part of TRA7 was included in the Supplementary Material.*

The figure referred to above is included as Figure S4 in the Supplement (Line 24).

**Supplementary material**

Tables S1, S2, S3: Please check decimal and 1000s delimiter, there are different styles used (comma and points mixed, sometimes comma as 1000s delimiter, sometimes not). Also "delta"234U instead of d234U.

*Thank you for mentioning the lack of harmonization. All data will be delimited consistently by using periods. The delta notation was also corrected in Tables S1, S2 and S3.*

The Tables have been revised accordingly.

Figure S4: Any ideas for the outliers, e.g., in TRA7 or FN1? Also, why is the age model of FN1 systematically older than the stalagmite ages? Also, why do you show ISCAM uncertainties, but COPRA average age model? Why not show ISCAM and COPRA in comparison?

*The outliers for TRA7 and FN1 were discussed above. We will include a more complete description of the U/Th ages when we submit our revision. The outliers can be explained by the $^{232}$Th content and $^{230}$Th/$^{232}$Th results. Please, see our detailed response to this question in the third paragraph of this RC2 response, when discussing U/Th dating results.*

*We decided to show COPRA age models, because the age model of ISCAM failed to produce reasonable extrapolations for the first and last millimeters of the stalagmites or to bridge intervals where we had identified a possible hiatus such as in the sample FN1. COPRA produces an independent linear age model allowing us to evaluate them without changes as made by ISCAM. However, both age model methods use a linear interpolation and produce very similar results. The plot with two time series does not show any significant differences between them; hence the choice of age model does not affect our interpretation. Please see figure below.*

*We also revisited the age models and the caption of Figure S4. The systematically older ages for FN2 were the result of a plotting error. We also corrected the text concerning the age model errors in the caption. This was not an ISCAM age model error, but a COPRA age model error. We corrected the graph and caption and present the revised version below (Figure S4).*

[Figure]

*Figure S4 – Age models for each stalagmite from Rio Grande do Norte. Age models were calculated using COPRA (Breitenbach et al., 2012) through a set of 2.000 Monte Carlo simulations. The COPRA age model was produced for each sample and covers the entire stalagmite. Squares and horizontal bars: age results with error bars. Red line: COPRA average age model. Grey line: age model errors considering 95% confidence interval.*

A more detailed explanation of the U/Th method is included in the Material and Methods section (Lines 213-235). There is also as description of the U/Th method in the Results section (Lines 368-383) and in the Discussion section we include a comprehensive explanation of $^{232}$Th contamination and calcite x aragonite crystallization (Lines 439-481). The figure referred to above is included as Figure S3 in the Supplement (Line 18).


Figure 5 (and lines 451–454): It is true that the presented AMV time series and the RN composite δ18O time series look similar, but it is unclear what the authors are plotting. The green time series in figure 5 (shown below, top figure) does not look like the AMV reconstruction from Lapointe et al. (2020) (shown below, bottom figure)—raw data from https://www.ncei.noaa.gov/access/paleo-search/study/31353. The full range of values from the Lapointe dataset is 21.7–22.7, while the reconstruction shown in figure 5 only appears to be from 21.95 to 22.40.

Perhaps the authors plotted a different reconstruction of the AMV and used the wrong citation? Or perhaps it is the reconstruction from Lapointe et al. (2020) but downsampled (if so, the authors need to make this clear in the methods or supplementary information)?

*Thank you for your comment and question. We addressed this point in the response to reviewer`s comments (RC1, comment 19). We incidentally plotted the AMV curve (Lapointe et al., 2020) backwards in the original manuscript. We have corrected the figure and the discussion based on the AMV. We invite you to read our response to the reviewers as it should help clarify your question.*

Figure 5 has been revised and is presented in Line 607.

**2. The authors do not sufficiently explain the mechanisms driving the anti-phased behavior observed between the RN composite and Paraíso Cave δ18O records. Specifically:**

Lines 436–440: It is unclear what is meant by "a zonal behavior of precipitation shifts in the ITCZ domain." Are the authors proposing that RN and Paraíso are in-phase from 250–1100 CE, anti-phased at ~1100 CE, back in-phase from 1100–1500 CE, and then anti-phased again from 1500–1750 CE? The authors should provide more explanation for this behavior.

Additionally, the authors state that "even though the Paraíso and Cariaco sites are located in different hemispheres, the observed in-phase climate relationship during the LIA suggests that their isotopic signatures were both sensitive to the same rainfall changes over northern South America." The Cariaco record is not an isotope-based record. Rather, it is a bulk titanium % record. The wording of this sentence should be changed accordingly.

*The Paraíso record cannot be interpreted in the same way as the RN record that predominantly receives rainfall originating from the ITCZ, while the Paraíso Cave is located at the margin of two different systems, the ITCZ and the South American Summer Monsoon (SASM), as described in our Climatology section (Figure 2). The location of Paraiso at the very edge of the SASM region likely explains why during certain intervals it varies in-phase and during others out of phase with the RN record. As shown by Orrison et al. (2022) during the last millennia the Paraiso record tends to be out of phase with the core monsoon region as a result of Bolivian-High-Nordeste Low intensification. However, a slight zonal shift of this leading mode of monsoon variability would change this relationship, as the Paraiso record would become part of the monsoon system, leaving it antiphased with the subsidence region over NE Brazil, where the RN record is located. Hence, the location of Paraiso at the node of this dipole, renders its response very sensitive to slight changes in the monsoon core. Furthermore, the zonal precipitation gradient between northeastern Brazil and the eastern-central Amazon is highly sensitive to changes in Pacific and Atlantic SST on multidecadal timescales. As shown by He et al. (2021), during the monsoon season (DJF), the zonal precipitation gradient response to Pacific SST variability completely reverses in this region, depending on the state of the Atlantic (see Figure 7 in He et al., 2021) and this change is transmitted via a perturbed Walker circulation (see their Figure 9). We now reference this mechanism in the revised paper, but discussing in great depth the joint interactions between Pacific and Atlantic and how they perturb Hadley and Walker circulation, respectively, is beyond the scope of this paper. We refer the interested reader to He et al. (2021) instead.*

*We have also revised the text in order to clarify that Cariaco is not an isotopic record.*

*We have rewritten this paragraph to adjust the discussion about the RN Composite the Paraíso and Cariaco records, respectively, according to suggestions we received from RC1 and 2. Please see our revised version below.*

*"When comparing N-NEB and eastern Amazon conditions, it is evident that the RN Composite shares some similarities with the Paraiso stalagmite record (Wang et al., 2017), due to the contribution of ITCZ precipitation in both places. But there are also important differences (Fig. 4). The RN Composite shows lower δ18O values between 500 and 1000 CE, compared to the earlier period, while Paraiso shows decreasing values around the same period, suggesting a slight increase in precipitation in both areas. From 1160 to 1500 CE, abrupt increases in δ18O values are seen in both records, which indicates abrupt and prolonged drought conditions due to a northward ITCZ migration. However, around 1100 CE, and the period from 1500 to 1750 CE, Paraiso is antiphased with the RN Composite and in phase with the Cariaco Basin (Haug et al., 2001), which is inconsistent with the notion of an ITCZ-induced regional precipitation change.*

*Instead, a zonally-oriented precipitation change within the ITCZ domain over Brazil is required to explain the anti-phased behavior between precipitation in N-NEB and the eastern Amazon, and similarities between Cariaco and the eastern Amazon."*

The text has been revised and expanded according to the reviewer's suggestions. It is included in Lines 570-583.

Lines 446–451: Here, the authors discuss the AMV and ITCZ displacement during a warm AMV. However, the authors have not defined what a warm AMV is, albeit the reader could find out in the cited studies. I recommend the authors specifically define the AMV in detail, and make clear what is meant by a warm vs cold AMV.

*We will clarify in the revised manuscript how warm and cold AMV are defined.*

The text has been revised and is presented in Lines 588-590 and 596.

Lines 461–463: The authors state, "Our analysis corroborates with this and points to increasing precipitation over N-NEB and decreasing precipitation over eastern Amazon, between 1500–1750 CE, when both AMV and PDV are in cold phase (Fig 4)." There is no reference to the PDV in figure 4, nor has the PDV been described/defined yet at this point in the text. No PDV reconstructions are provided in any of the figures, and the provided AMV reconstruction is in figure 5, not figure 4. Last millennium SST gradients from Steinman et al. (2022) are provided in figure 5, but they are not PDV or AMV reconstructions. I recommend either including a PDV reconstruction in one of the figures, or to remove this text from the manuscript.

Lines 463–465: The authors state, "This sign reversal is assigned to perturbations of the regional Walker cell's produced by teleconnection between the Atlantic and Pacific (Kayano et al., 2022, He et al., 2021)." I find this explanation to be vague, and recommend that the authors provide a clearer and more detailed explanation for the sign reversal. What does "perturbations of the regional Walker cell's" mean exactly? What teleconnections are the authors referring to, and what are the mechanisms driving the aforementioned perturbations?

*The Figure presents only the AMV. The discussion about the relationship between AMV and PDV was only based on an observed precipitation analysis. We made some adjustments in the paragraph to clarify the aspects mentioned in the last two comments. Please see the revised text below.*

*"According to Kayano et al. (2020, 2022), during the last century, dry conditions over N-NEB and the eastern Amazon are present when AMV and Pacific Decadal Variability (PDV) are in both in their warm phase, or when the AMV is in a cold phase and the PDV in its warm phase. On the other hand, when AMV and PDV are both in their cold phase, precipitation over the Amazon is anti-phased with NEB, resulting in decreased precipitation over the Amazon and increased precipitation over NEB. This zonally aligned precipitation signal over eastern tropical S. America is the result of joint perturbations of both the regional Walker and Hadley Cell's, produced by teleconnection between the Atlantic and Pacific (Kayano et al., 2022, He et al., 2021). These conditions can explain in part our results, however during the decoupling of our record with AMV (between 1500 and 1750 CE), increasing precipitation over N-NEB and decreasing precipitation over the eastern Amazon can be better explained by the positive gradients both in Atlantic and Pacific Oceans forcing a south ITCZ migration (Fig. 4)."*

The text has been revised and is presented in Lines 634-645.

**3. The conclusion and abstract both discuss ITCZ dynamics forced by the AMV and PDV, including position, intensity, and width. However, in the main text, the authors do not sufficiently explain which dynamical aspect of the ITCZ responds to different AMV/PDV phases, nor do they explain any mechanism(s) behind the AMV/PDV forcing. Specifically:**

Lines 570–577: In this paragraph, the authors suggest that during the last millennia, ITCZ dynamics cannot be explained solely by north-south ITCZ migrations or one single forcing mechanism. They propose a zonally non-uniform behavior of the ITCZ during times when the RN 4 record is anti-phased with the Paraíso cave record—forced by the interactions between the AMV and PDV modes that changed the regional Walker cell position and ITCZ intensity/width.

However, the authors never really attributed the anti-phased behavior between N-NEB and eastern Amazonia to the differential AMV/PDV phases. They discussed observed precipitation anomalies during overlapping periods of AMV and PDV phases in the modern, and suggested that it could be responsible for the observed anti-phased behavior. However, they never directly compared the speleothem time series with AMV and PDV reconstructions. Nor did the authors propose a detailed mechanism for how different AMV/PDV phases impact ITCZ width/intensity, despite changes in ITCZ width/intensity also being mentioned in the abstract (lines 46–50). In addition, the authors did not really describe when the ITCZ may have expanded/contracted or became weaker/stronger (aside from stating that this may have happened when the RN composite record and Paraíso are anti-phased). Ultimately, they never describe mechanism(s) for 1) how different AMV/PDV phases impact ITCZ dynamics, 2) how changes in ITCZ width/intensity may cause the observed anti-phased behavior, and 3) how the regional Walker cell position is forced by different AMV/PDV phases. I recommend that the authors provide more detail to this part of the Conclusions and Discussion sections overall, and propose/explain specific mechanisms that can reconcile the observed hydroclimate variability in N-NEB and eastern Amazonia.

> We have responded to this comment above.

Additional note: The authors should be extremely clear when generally discussing ITCZ width/intensity. What exactly do the authors mean by ITCZ width? Is it the width of the actual band of deep convection? Width of the seasonal range of the ITCZ? These terms should be explicitly defined early in the manuscript. Some papers that may be useful to reference include Donohoe et al. (2013), Atwood et al. (2020), Byrne and Schneider (2016), and Roberts et al. (2017).

> *The ITCZ definition adopted is the one referring to it as the modern tropical rain belt of maximum precipitation and the ITCZ position is defined according to Schneider et al. (2014). The position is mentioned in line 160 of the manuscript, when we define the locality of our study site and its relationship with the ITCZ. We will add to this definition by including the ITCZ position during the boreal winter over the Atlantic (2° N). We will change the term "ITCZ width" by "ITCZ length". We were referring to the duration of the ITCZ over N-NEB, from March to May, during its southernmost extent, but we did not intend to imply a specific ITCZ dimension. We will rephrase how we refer to the ITCZ's southernmost expansion in MAM to avoid confusion.*

> The respective changes related to ITCZ position (Line 183) and ITCZ or rainy season lengths (Line 315 and 750) have been made.

**Additional comments and concerns:**

Lines 89–92: The authors cite Lechleitner et al. (2019), but I believe the correct citation is Lechleitner et al. (2017). Additionally, another relevant citation that may be relevant and could be included here is Asmerom et al. (2020) published in Science.

*The citation will be corrected in the text. We will consider including other references as appropriate in the manuscript.*

The reference has been corrected.

Lines 95–102: The authors call out the SASM and the ITCZ here as focus points of recent studies on tropical South American precipitation, but have not mentioned the South Atlantic Convergence Zone (SACZ). While not explicitly relevant to their findings, the SACZ should at least be mentioned here because of its important relationship with the SASM and ITCZ, and because it has been the topic of several recent paleoclimate and modern precipitation studies (Novello et al., 2018; Nielsen et al., 2019; Zilli et al., 2019; Wong et al., 2021).

*We will include a brief discussion of the SACZ in the Introduction section, although the SACZ is not directly responsible for the precipitation observed at our study sites.*

The SACZ is now being discussed in Lines 99-101 in the revised version of the manuscript.

Figure 1: It may help the reader to include annotations in the figure, including labeling the core SASM domain, ITCZ location, SACZ, etc. Additionally, while I understand the choice to include austral autumn precipitation climatology (when N-NEB receives most of its precipitation), it may be worthwhile to include panels with precipitation climatology for the austral winter and spring (either added to figure 1 or included in the supplement). This would allow for the reader to visually assess the spatiotemporal dynamics of the ITCZ, SASM, and SACZ, and how precipitation varies at sites 1–4 during the different seasons. 5

*We will consider including the annotations of ITCZ, SASM and SACZ in Figure 1. However, including fractional precipitation panels for JJA and SON does not add much relevant information for our region as precipitation at this time of year is low (see panels below). We therefore prefer to focus on the key rainy seasons DJF and MAM.*

*Indications of ITCZ and SACZ locations are included in Figure 1.*

[Figure]

Lines 165–174: Figure S1 receives a lot of attention in this paragraph, and should probably be included as a main text figure. Alternatively, it could be incorporated into an existing main figure.

*We will change this section and include the text related to the climatology of the region and Figure 2 in the results section instead, following to RC2's suggestion. Certainly, Figure S1 can be included in the main text.*

We moved Figure S1 to main text, where is it now included as Figure 2 (Line 338).

Figure 2: Readers who are green-red colorblind will not be able to see the small green dots (that denote the location of the GNIP stations) in any of the panels. I recommend changing the color to black and potentially increasing the size of the dots.

*The journal editorial team already mentioned that we had to adapt the figure for color blind readers during the revision stage. All figures in the manuscript are now adapted accordingly.*

All figures have been adapted for color blindness. The dot size has also been adjusted in the new version.

Lines 362–363: It gets confusing when the authors use both before present (BP) dates and before common era/common era (BCE/CE) dates. Additionally, ky has not been defined before this point, so the authors should spell it out before using the abbreviation.

*The nomenclature of the time periods will be standardized.*

The only instance where we refer to ky BP is with regards to the 4.2 ky BP event, which is known as such in the paleoclimate community and the literature. In order to clarify this aspect, we included a brief description in Lines 504-505.

Figure 3: Same red–green issue as mentioned in Figure 2.

*The journal editorial team already mentioned that we had to adapt the figure for color blind readers during the revision stage. All figures in the manuscript are now adapted accordingly.*

All figures have been adapted for color blindness in the new version.

Figure 4: It would be extremely helpful for the authors to include vertical bars when referencing specific time periods in the text. Such periods include the LIA, MCA, Bond 2 event, etc. Additionally, the authors reference trends resulting from insolation forcing in the paragraph starting at line 417.

The authors should consider including a time series of solar insolation.

*We have included the insolation curve and also vertical bars in Figure 3 and 4 of the manuscript. This Figure in question already contains a lot of information and adding even more would make it difficult to read. We will instead add the insolation curve to the Figure S5 in the Supplement as shown below.*

The mentioned periods are indicated in Figure 4.

Also, the δD record from Boqueirão Lake is relative to VSMOW, not VPDB (Utida et al., 2019). This appears to be a typo and should be changed accordingly.

*We have corrected this typo. Thank you for drawing attention to it.*

This typo has been corrected in Figure 4.

Lines 389–392: The authors state that from 1060 to 480 BCE, there was increased precipitation in N-NEB as suggested by negative δ18O anomalies. But it is unclear what the authors mean by 'increased precipitation'. During this time, there is multidecadal variability in the RN composite δ18O record, but no clear/obvious trend between 1060 and 480 BCE. Perhaps the authors meant that there was increased precipitation *relative* to another part of the record. I would recommend clearing this up.

*We have rewritten this paragraph to clarify our statement. Please see below.*

*"The oldest period covered by the RN Composite, from 1200 to 500 BCE, is characterized by successive dry and wet multidecadal periods, with increased precipitation in N-NEB from 1060 to 750 BCE and from 460 to 290 BCE, as suggested by the negative departures seen in the $\delta^{18}O$ values. During this last period, there is also a tendency from lower to higher $\delta^{13}C$ values, suggesting progressive surface soil erosion related to rainfall variability (Fig. 4), as interpreted by Utida et al. (2020). This period ends in a stable interval, lasting from 300 BCE to 0 CE, with $\delta^{13}C$ values close to the bedrock signature at about -1‰ to +1‰, indicating a lack of soil above the cave. After an abrupt reduction of $\delta^{13}C$, the values decrease to approximately -2‰ between 200 CE and 1500 CE. From 1500 CE to the present, negative values of $\delta^{13}C$ represent wet climatic conditions as indicated by lower $\delta^{18}O$ values. The more negative $\delta^{13}C$ during this period can be related to denser vegetation that favored both soil production and stability above the cave."*

The paragraph has been revised, also taking into consideration other suggestions that were made. It is shown in Lines 518-536.

Lines 408–409: The authors reference the δD record from Boqueirão Lake, and the same record is shown in figure 4. However, the authors describe the record as a "δD lipids" record. Lipids are a broad group of molecules which include waxes, glycerides, terpenoids, tetrapyrrole pigments, etc. The authors should be more specific, and should reference the record as a leaf wax δD record of *n*-C28 alkanoic acids from Boqueirão Lake sediments (hereinafter referred to as δD lipids).

*We will include the description "n-C28 alkanoic acid obtained in leaf waxes" when first discussing the δD record from Boqueirão Lake (Utida et al., 2019).*

The sentence is mentioned in Line 538.

Lines 495–497: The authors focus their discussion of extreme dry events recorded in the TRA5 δ18O record between 1500 and 1850 CE. However, it is unclear why the authors do not discuss dry events/distinct δ18O peaks after 1850 CE, despite their record extending into the 21st century. Is it because the TRA5 speleothem chronology is not as precise during this time?

*The TRA5 chronology during the last 150 years is indeed not precise enough to discuss historical events. Since we will improve our discussion of age models, according to RC's2 suggestion, the TRA5 chronology will also be better explained and we will clarify this question in the updated version of the manuscript.*

A revised text regarding the subject of this comment is included in Lines 717-720. Other changes made regarding U/Th results are also presented in the Result section.

Lines 518–523 and Figure 6: The authors reference several historical droughts that had severe societal/socioeconomic consequences. It may be helpful to annotate figure 6 to highlight the most severe droughts referenced in the text. The number/letter labeling in figure 6 makes it hard to discern the severity of the droughts by looking at the figure alone. 6

*According to the historical records, the most significant drought events registered in our stalagmite are related to points 4, 6 and 7. They will be highlighted in Figure 6 and mentioned in the caption.*

The figure has been revised according to the suggestions.

Lines 533: The authors should provide more detail here. Which Governor are the authors referring to? Governor of what/where?

*The Governor mentioned here is the Brazilian Governor. In that period, Brazil was a colony of Portugal and there was a local government. We will specify this in the text as "Brazilian Governor".*

This question has been clarified in Line 710.

Figure 6: Why focus on just TRA5? TRA7 and FN1 appear to cover the same period as TRA5. Is TRA5 the only speleothem that records the extreme drought events? Do TRA7 or FN1 record any of the same drought events? If they do not, why would only one speleothem record these drought events and not the others?

*Thank you for your comments and question. We addressed this point in the reviewer's comments. We invite you to please read our response in those files and hope they will help clarify your question.*

We justify the use of TRA5 in Lines 651-673. We expanded the discussion regarding U/Th results in order to support all related aspects regarding age model and data resolution.

It may be helpful to include the age uncertainty in the right panel of the figure under the heading "TRA5". For example, 1546 ± XX. Especially because this figure focuses on only the last 500 years, it would allow the reader to critically compare the speleothem dates to the historical drought dates listed in the column labeled "Historical."

*Thank you for your comments and question. We addressed this point in detail in the RC2's comments considering the U/Th ages and age model. We invite you to please read our response in those files and hope they will help clarify your question.*

We restructured the Section about the last 500 years and now discuss the limitations of the TRA5 age model. We also justify our approach based on evidence from historical records.

Additionally, I am curious if there is an available archeological record(s) or something similar that could be plotted with the TRA5 δ18O record. Especially since the authors discuss the societal implications of the extreme droughts in relation to human population and welfare, it would be useful for the reader to visualize the impact through comparison with the speleothem record.

*The Brazilian archeological records were discussed and compared with stalagmite data during the Holocene by Utida et al. (2020). However, these data basically describe the total population size during random intervals (https://memoria.ibge.gov.br/historia-do-ibge/historico-dos-censos/dados-historicos-dos-censos-demograficos.html) and they are not helpful to discuss episodic extreme events. Furthermore, considering the lack of demographic data in Brazil, from 1500 to 1870 CE, such a comparison with the stalagmite record, unfortunately, is not feasible.*

No change is necessary regarding the comment above.

Line 565–567: The authors state, "The N-NEB record presents a trend toward drier conditions as is also being observed in the Diva de Maura Cave in S-NEB, interpreted as an ITCZ withdrawal and SASM weakening, respectively." It is unclear what the authors mean by "ITCZ withdrawal," especially since the authors highlighted the dynamical behavior of the ITCZ earlier in the paper. Is it a withdrawal via mean ITCZ displacement? Contraction or weakening of the ITCZ? More detail here would be helpful for the reader.

*'Withdrawal' of the ITCZ was meant to indicate that it's mean position moved northward. We will clarify this in the revised version of the manuscript.*

We revised the sentence in Lines 746-748,. "withdrawal" has been changed to "contraction" in order to clarify the statement.

[Figure]

*Figure S5 – Rio Grande do Norte stalagmite isotope record. (a) U/Th ages for RN stalagmites. (b) Raw data of $\delta^{13}C$. (c) Oxygen isotope results corrected for calcite-aragonite fractionation ($\delta^{18}O_{C-A}$), according to weight proportion of mineralogical results. (d) $\delta^{18}O$ RN Composite constructed using stalagmite records from NEB (black line). Grey lines denote the age model confidence interval of 99%. (e) February insolation curve at 10°S.*